# High-resolution map of the *Plasmodium falciparum* genome reveals MORC/ApiAP2-mediated links between distant, functionally related genes

Parul Singh[1,6], Jacques Serizay [2,6], Justine Couble[3,6], Maureen D. Cabahug[1,4], Catarina Rosa[1,5], Patty Chen[1], Artur Scherf [1], Romain Koszul [2], Sebastian Baumgarten [3] & Jessica M. Bryant [1] ✉

Genome organization plays an important role in silencing compacted, heterochromatinized genes in the most virulent human malaria parasite, *Plasmodium falciparum*. However, it remains unclear how these genes spatially cluster or whether active genes are also organized in a specific manner. We used Micro-C to achieve near-nucleosome resolution DNA–DNA contact maps, which revealed previously undescribed inter- and intrachromosomal heterochromatic and euchromatic structures in the blood-stage parasite. We observed subtelomeric fold structures that facilitate interactions among heterochromatinized genes involved in antigenic variation. In addition, we identified long-range intra- and interchromosomal interactions among active, stage-specific genes. Both structures are mediated by AP2-P, an ApiAP2 DNA-binding factor, and a putative MORC chromatin remodeler, and functional specificity is achieved via combinatorial binding with other sequence-specific DNA-binding factors. This study provides insight into the organizational machinery used by this medically important eukaryotic parasite to spatially coordinate genes underlying antigenic variation and to co-activate stage-specific genes.

Genome organization within the nucleus is important for transcriptional regulation and genome integrity of eukaryotes[1–4]. Chromatin spatial compartmentalization has been correlated with different transcriptional outcomes. Lamin-associated chromatin domains tend to encompass developmentally regulated or lowly expressed genes and have repressive chromatin signatures[5]. On the other hand, structures such as the nucleolus and nuclear speckles are located towards the interior of the nucleus and are associated with high levels of gene activity[6]. Multiple studies have demonstrated that cohorts of active genes on the same and different chromosomes can physically associate and often overlap with RNA polymerase II-enriched foci in what are called 'transcription factories'[7–9]. Regardless of the term used, spatial association of genes in nuclear bodies is not random but mediated by specific chromatin-associated factors to achieve a boost in transcriptional activity[6].

While most genome organization studies have been in model organisms, non-model organisms such as protozoan pathogens offer fascinating insight into how nuclear architecture and transcription

[1]Institut Pasteur, Université Paris Cité, INSERM U1201, CNRS EMR9195, Biology of Host-Parasite Interactions Unit, Paris, France. [2]Institut Pasteur, Université Paris Cité, CNRS UMR 3525, Unité Régulation Spatiale des Génomes, Paris, France. [3]Institut Pasteur, Université Paris Cité, G5 Parasite RNA Biology, Paris, France. [4]Sorbonne Université, École Doctorale Complexité du Vivant ED515, Paris, France. [5]Present address: Cell Biology of Host – Pathogen Interaction Laboratory, Gulbenkian Institute for Molecular Medicine, Oeiras, Portugal. [6]These authors contributed equally: Parul Singh, Jacques Serizay, Justine Couble. ✉e-mail: jessica.bryant@pasteur.fr

are connected to promote survival in a hostile environment. The life cycle of the most virulent human malaria parasite, *Plasmodium falciparum*, is driven by a complex transcriptional cascade in which each stage has a characteristic transcriptional programme[10,11]. The influence of genome organization on transcription has been explored using chromosome conformation capture (Hi-C)[12,13], which confirmed that gross structure is driven by important features previously described in other eukaryotes[14,15]: clustering of ribosomal DNA loci[12,13,16], telomeric regions[12,13,17,18] and centromeres[12,13,19]. However, a unique feature of its genome architecture is the strong inter- and intrachromosomal association of heterochromatinized genes that are involved in antigenic variation, pathogenesis and sexual development[12,20–22]. Included are *ap2-g*, a transcription factor that induces sexual commitment (gametocytogenesis) (Fig. 1a), and virulence genes (such as *var*), which belong to multigene families that encode variant surface antigens crucial to infection and pathogenesis. Whether located in subtelomeric or central chromosomal regions, silent *var* genes are bound by heterochromatin protein 1 (HP1) and form clusters at the nuclear periphery[20,23]. This clustering is believed to be involved in the coordination of *var* gene mutually exclusive expression[12,22,24] as well as recombination among *var* genes, which generates antigenic diversity[18,25–28].

The transcriptional repression of these heterochromatinized genes is critical for maintaining an infection in the human host; however, other than HP1 (ref. 20), it is still unclear what protein factors underlie their organization. Importantly, *P. falciparum* lacks CTCF[29] and lamins[30], which are important for genome organization in other eukaryotes. Moreover, the *P. falciparum* genome encodes relatively few sequence-specific DNA-binding factors, most of which belong to the 27-member ApiAP2 family. ApiAP2 factors have been implicated in transcriptional regulation at multiple different stages of the parasite life cycle, but their role in genome organization has not been well established[31].

The focus of most genome structure studies in *P. falciparum* has been on the organization of silent, heterochromatinized genes. However, Hi-C resolution remains limited to ~10–25 kb in the extremely AT-rich *P. falciparum* genome[32], which is insufficient to identify finer-scale structure and leaves several open questions regarding: (1) the structuration of heterochromatin domains, (2) whether euchromatic genes form long-range or structured interactions similar to heterochromatic genes to contribute to gene regulation; and (3) which factors mediate these structures.

To tackle these questions, we adapted Micro-C[33], a nucleosome-based Hi-C derivative, to study *P. falciparum* genome organization at sub-kb resolution during the asexual replication cycle, which takes place in human red blood cells (RBCs) (Fig. 1a). The integration of these data with chromatin immunoprecipitation and sequencing (ChIP-seq) and RNA-sequencing (RNA-seq) data revealed that the *P. falciparum* genome is organized into heterochromatic and euchromatic domains whose boundaries are formed by ApiAP2 DNA-binding factors and a putative microrchidia chromatin remodeller, MORC. The variable

composition of these complexes defines the function of different boundary elements: some anchor the folding of heterochromatic subtelomeric regions, while others facilitate intra- and interchromosomal long-range interactions that create hubs of stage-specific transcription. This study reveals an unprecedented complexity of a small eukaryotic genome that relies on relatively few sequence-specific DNA-binding proteins to achieve drastic transcriptional changes that enable its complicated parasitic life cycle.

## Results

### Micro-C provides a high-resolution view of genome organization

To generate a high-resolution genome-wide contact map in *P. falciparum* that allows for comparison to nucleosome-scale datasets (that is, ChIP-seq and assay for transposase-accessible chromatin with sequencing (ATAC-seq)), we adapted Micro-C[33] to this organism. Micro-C uses micrococcal nuclease (MNase) to fragment the genome to the nucleosome level, providing a higher short-range resolution of chromatin conformation than the Hi-C protocol, which uses restriction enzymes. We performed four replicates in clonal wild-type (WT) parasites in the late stage of the red blood cell cycle, which showed high correlation (Extended Data Fig. 1a,b) and were subsequently combined to achieve higher resolution. We observed a strong nucleosome banding pattern in our Micro-C chimaeric fragments (Extended Data Fig. 1c), and the intrachromosomal distance-dependent genomic interactions frequency P(s) profile (that is, contact decay curve) revealed a typical polymer behaviour with a linear P(s) (slope ~−1.2) maintained for distances as short as 1 kb (Extended Data Fig. 1d). Thus, we confirm that Micro-C substantially increased resolution of the *P. falciparum* contact map down to 1 kb (Fig. 1b).

We observed previously described interactions among telomeres and centromeres and strong inter- and intrachromosomal interactions of HP1-heterochromatinized virulence genes such as the ~60-member *var* multigene family (Fig. 1c,d and Extended Data Fig. 1b,e). Except for *ap2-g*, we did not observe strong inter- or intrachromosomal contacts between HP1-enriched virulence genes and HP1-enriched non-virulence genes such as *crmp3* (Fig. 1d,e)[20]. In addition, HP1 domains overlapping virulence genes (for example, the central *var* gene clusters on chromosomes 4, 7, 8 and 12 that encompass 4–7 *var* genes each) and *ap2-g* formed well-defined insulated domains (Fig. 1d–g), while HP1 domains containing other non-virulence genes (for example, *crmp3*) do not form such insulated domains (Fig. 1d,g). These data suggest that HP1 alone is not sufficient either to insulate a heterochromatic domain from nearby euchromatin, or to dictate the coalescence of distant genomic loci into a heterochromatin compartment.

### Micro-C reveals AP2-P and MORC-enriched subtelomeric structures

Our high-resolution contact map allows the study of fine-scale organization within subtelomeric HP1-enriched domains in the *P. falciparum* genome (see Extended Data Fig. 2a for detailed explanations of

**Fig. 1 | Micro-C provides a high-resolution view of *P. falciparum* genome organization. a**, The asexual replication cycle begins when a merozoite that has just egressed from an infected human red blood cell invades a new red blood cell. During the cycle, the parasite develops from a ring (early stage) to a trophozoite (middle stage), which undergoes DNA replication and schizogony to form a schizont (late stage). Genes that are needed for egress of the merozoites from the infected red blood cell and invasion of a new red blood cell are expressed in late-stage parasites. A small percentage of parasites exit the asexual cycle to undergo sexual commitment through a process called gametocytogenesis, which generates male and female gametes that can be transmitted to the mosquito. Created with BioRender.com/itpkoxy. **b**, Comparison of Micro-C data to previously published Hi-C datasets[12,51] from *P. falciparum*. Contact maps of a portion of chromosome 12. HP1 ChIP/input ratio tracks from late-stage parasites[91] are shown at the bottom, with selected genes indicated. Res., resolution.

**c–f**, Micro-C contact maps in late-stage parasites over chromosomes 10 to 12 (5 kb resolution) (**c**), the entire chromosome 12 (2 kb resolution) (**d**), a 260-kb-wide section of chromosome 12 (1 kb resolution) (**e**) and a 500-kb-wide section of chromosome 12 (2 kb resolution) (**f**). **e** and **f** are delineated with dashed boxes in **d**. **d–f**, The bottom corner is normalized interaction frequency. The top corner is log$_2$-scaled observed/expected interaction frequency ratio. HP1 ChIP/input ratio tracks from late-stage parasites[91] are shown at the bottom, with selected genes indicated. Colour scales are shown at the right of Fig. 1d. exp., expected. **g**, Insulation score at boundaries of HP1 central chromosomal domains containing virulence genes (*n* = 26, *P* = 0.00148) or not (*n* = 100, *P* = 0.581) in late-stage parasites (Supplementary Data 6 and 14). Randomly chosen domains were used as a control. Boxes represent the median and interquartile range (IQR), and whiskers represent ±1.5× IQR. *P* values from two-sided *t*-tests are indicated.

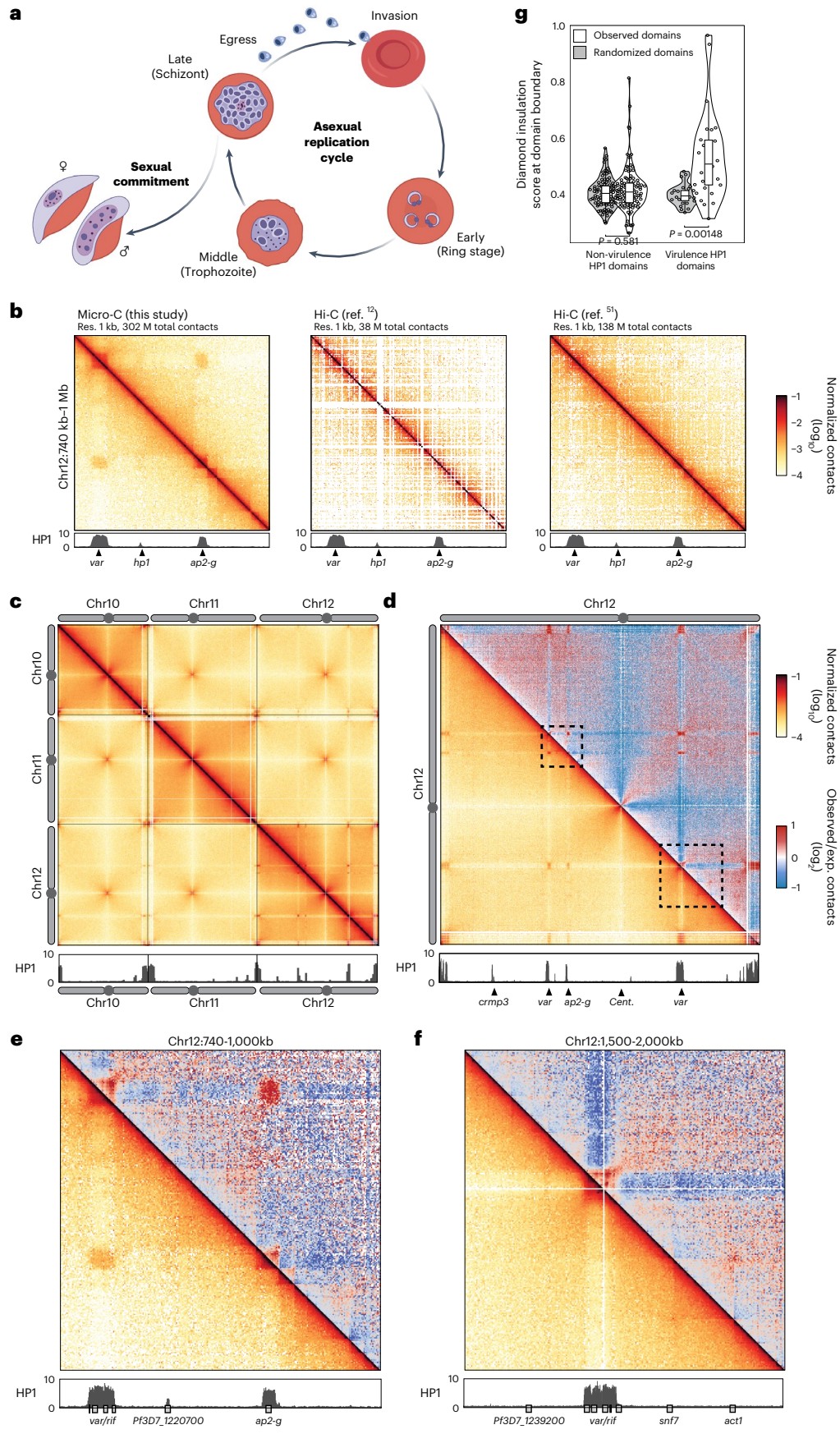

different Micro-C structures). HP1 heterochromatin encompasses each subtelomeric region, including the non-coding region adjacent to the telomere and several virulence genes downstream that encode variant surface antigens (Fig. 2a and Extended Data Fig. 2b). However, we detected a clear break in local contacts in the middle of this HP1 domain, forming two smaller self-associating domains: one non-coding and one coding (Fig. 2a and Extended Data Fig. 2a (left)). The non-coding domain starts just downstream of the telomere and ends just upstream of the first gene, which is usually a *var* gene (Fig. 2a and Extended Data Fig. 2b). This domain forms a fold-like structure anchored at its boundaries, indicated by a corner point structure in the Micro-C data (Fig. 2a,b and Extended Data Fig. 2a (middle)). In contrast, coding HP1 subtelomeric domains do not form a fold-like structure (Fig. 2a). Importantly, we observe these structures only at chromosome ends that contain extended subtelomeric regions and *upsB var* genes (see Extended Data Fig. 2b) and not at those that do not (such as chromosome 14), suggesting a link between this fold structure and *var* gene biology (Extended Data Fig. 3a).

We next sought to identify the molecular factors structuring this subtelomeric fold. Incidentally, we recently identified proteins that are enriched in the chromatin of *var* gene promoters using a CRISPR-based proteomics approach[34]. Included were two ApiAP2 DNA-binding factors—SPE2-interacting protein (SIP2, PF3D7_0604100) and AP2-P (PF3D7_1107800)—and a putative microrchidia (MORC, PF3D7_1468100) chromatin remodeller. In other eukaryotes, MORC proteins are ATPases that bind non-specifically to and compact DNA[35]. Thus, most MORC proteins, including in *Toxoplasma gondii*, have been implicated in gene repression and heterochromatin formation[36–41].

We determined genome-wide binding of AP2-P and MORC by generating epitope-tagged strains (Extended Data Fig. 3b,c) and performing ChIP-seq in clonal late-stage parasites (Fig. 1a) when these proteins show highest expression (Extended Data Fig. 3d,e). The ChIP-seq data confirmed the findings from our CRISPR-based proteomics study[34] and showed that the highest enrichment of AP2-P and MORC is upstream of subtelomeric *var* genes whose promoters are proximal to the telomere (*upsB var* genes, Extended Data Fig. 2b) (Fig. 2a,c, Extended Data Fig. 3f and Supplementary Data 1). We observe that MORC binds a broad region several kb downstream of the telomere (mean 8.3 kb ± 1.7 kb), and jointly with AP2-P in three large, discreet peaks 1–2 kb upstream of the transcription start site (TSS) of almost all *upsB* subtelomeric *var* genes (Fig. 2b,c, Extended Data Fig. 3f and Supplementary Data 1). While all *var* genes are transcriptionally silent in late-stage parasites, we did not observe this same enrichment pattern at other subtelomeric (*upsA*) or central chromosomal *var* genes (*upsC*) (Fig. 2c and Extended Data Fig. 2b). Importantly, we did not observe significant AP2-P and MORC enrichment at the boundaries between HP1 heterochromatin and euchromatin (Fig. 2a).

Despite being HP1 heterochromatinized, subtelomeric AP2-P/MORC-bound loci show substantial chromatin accessibility (Fig. 2a and Extended Data Fig. 3f) and are bound by other factors such as SIP2 (refs. 34,42) and telomere repeat-binding zinc finger protein (TRZ)[43], which was shown to play a role in telomere homeostasis (Fig. 2a,c and Extended Data Fig. 3f). Strikingly, the two regions of AP2-P and MORC subtelomeric enrichment precisely overlap the boundaries of the subtelomeric fold structure we identified with Micro-C (Fig. 2a,b). Enrichment of multiple factors at these subtelomeric structures where the coding region of the chromosome transitions to the non-coding subtelomere hints at a protein complex that could play an important role in the maintenance of chromatin structure.

We further found that subtelomeric AP2-P/MORC-bound loci form long-distance intra- and interchromosomal interactions, identified by a focal point of contact between subtelomeric ends (Fig. 2d–f and Extended Data Figs. 2a (right) and 3g). These contacts are distinct from telomere–telomere contacts. Chromosome 14, which lacks *var* genes and high AP2-P and MORC enrichment (Extended Data Fig. 3a), exhibits strong intra- and interchromosomal telomere–telomere, but not subtelomeric, contacts (Extended Data Fig. 3a,h). These data suggest that AP2-P and MORC may facilitate intra- and interchromosomal interactions of subtelomeric regions that contain *var* genes.

## AP2-P is required for subtelomeric structures and interactions

To determine whether AP2-P plays a role in *var* gene regulation or subtelomeric DNA structure, we performed a knockdown (KD) of this protein using an inducible *glmS* ribozyme system[44]. Despite a substantial KD at the protein level (Extended Data Figs. 3d and 4a), we did not observe an apparent growth phenotype or cell cycle arrest (Extended Data Fig. 4b). Interestingly, AP2-P KD followed by RNA-seq and differential expression analysis in late-stage parasites (Extended Data Fig. 4c and Supplementary Data 2) confirmed that AP2-P KD did not affect cell cycle progression (Extended Data Fig. 4d) and showed that there was no significant derepression of *var* genes (Extended Data Fig. 4e). However, AP2-P KD did result in the significant decrease of *morc* transcript levels (Extended Data Fig. 4f). To determine whether AP2-P KD results in MORC downregulation at the protein level, we fused *morc* to *gfp* in the AP2-P-3HA-*glmS* strain (Extended Data Fig. 4g). In the resultant strain, AP2-P co-immunoprecipitated with MORC, suggesting a direct interaction that could potentially stabilize both proteins (Extended Data Fig. 4h). Indeed, AP2-P KD resulted in MORC downregulation at the protein level in late-stage parasites (Fig. 3a and Extended Data Fig. 4i,j). These data demonstrate that AP2-P regulates MORC and that an AP2-P KD is a MORC KD as well.

While AP2-P (and thus MORC) downregulation did not substantially affect *var* gene transcription, AP2-P KD followed by Micro-C revealed structural consequences. First, AP2-P KD led to a loss of the

---

**Fig. 2 | Micro-C reveals subtelomeric fold structures defined by a multiprotein complex. a**, Micro-C contact maps of either end of chromosome 10 (100 kb wide, 500 bp resolution) and AP2-P, MORC, HP1 (ref. 91), TRZ[43] ChIP/input and ATAC-seq[101] tracks from late-stage parasites. Chromosome illustrations: left arm telomere (red), non-coding subtelomeric region (tan), virulence gene-encoding subtelomeric region (green), euchromatic genes (grey), right arm telomere (purple). The dashed red circle indicates the subtelomeric fold contact point. Non-vir., nonvirulence. **b**, Off-diagonal Micro-C contact maps aggregated over 20 subtelomeric loci with a non-coding subtelomeric region (±5 kb, 500 bp resolution) (Supplementary Data 15) and aggregated AP2-P and MORC ChIP/input ratios in late-stage parasites. For each subtelomeric locus, a 10-kb-wide contact map was extracted, centred at the contact (dashed red circle) between the broad MORC peak in non-coding subtelomeres and the MORC/AP2-P-enriched transition region. Contact maps are mirrored so the telomere is always positioned towards the top left corner. Top: normalized interaction frequency. Bottom: log₂(observed/expected interaction frequency). **c**, Metagene plots showing average AP2-P and MORC ChIP enrichment in clonal late-stage parasites

upstream and downstream of the predicted TSS for *upsB* (top) and *upsA/upsC var* genes (bottom). TRZ ChIP-seq data[43] from stage-matched parasites are included. One replicate was used for the AP2-P and MORC ChIP datasets. **d**, Micro-C contact map between both subtelomeric regions of chromosome 10 in late-stage parasites (100-kb-wide contact map, 500 bp resolution), presented otherwise as in **a**. **e**, Interchromosomal Micro-C contact map between subtelomeric regions of the left arm of chromosome 10 and of the left arm of chromosome 9 in late-stage parasites (60 kb wide, 1 kb resolution), presented otherwise as in **a**. The dashed blue circle indicates the subtelomeric contact point. **f**, Interchromosomal Micro-C contact maps aggregated over the 64 interchromosomal pairs of subtelomeric loci, characterized by the presence of a non-coding subtelomeric region (±15 kb, 500 bp resolution), and aggregated AP2-P and MORC ChIP/input ratios from late-stage parasites. For each pair, a 30-kb-wide contact map was extracted, centred at the MORC/AP2-P-enriched transition region of each chromosome. Contact maps have been mirrored so that the telomere is always positioned towards the top left corner. Inset: log₂(observed/expected interaction frequency) of the area highlighted with a dashed blue circle.

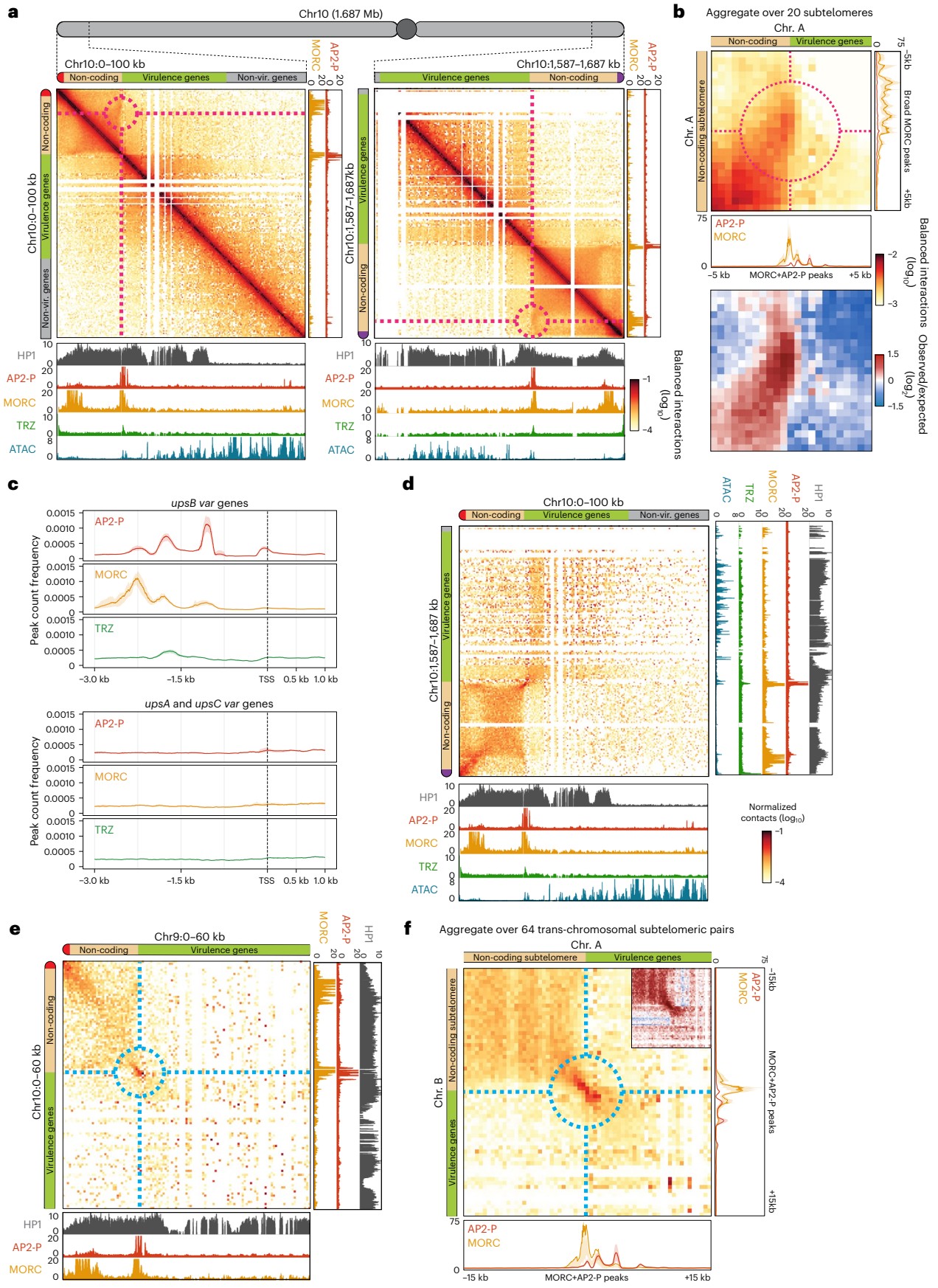

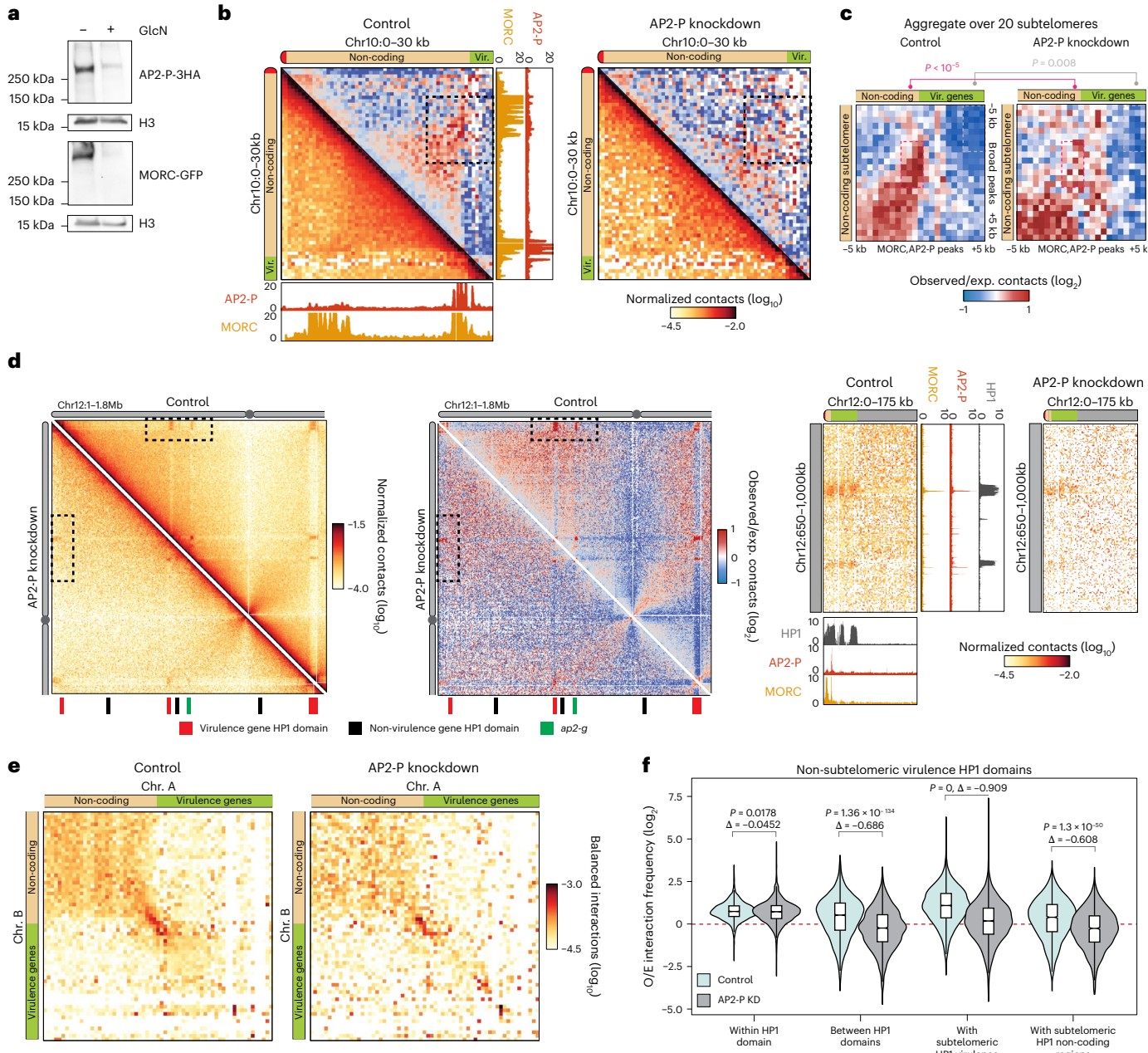

**Fig. 3 | AP2-P is required for non-coding subtelomeric fold structures and interchromosomal interactions. a**, Western blot of nuclear extracts from AP2-P-3HA-*glmS*:MORC-GFP late-stage parasites with or without glucosamine (GlcN) for 96 h. Antibodies against HA, GFP and histone H3 were used. Molecular weights at left. Representative result of 3 independent experiments. **b**, Subsampled (65 M contacts) control and AP2-P KD Micro-C contact maps centred at the interaction of the left end of chromosome 10 in late-stage parasites (30 kb wide, 500 bp resolution; bottom corner: normalized interaction frequency; top corner: log2(observed/expected interaction frequency)) presented as in Fig. 2a. Dashed black box: area of contact between the subtelomeric fold anchor points. **c**, Subsampled (65 M contacts) control and AP2-P KD off-diagonal Micro-C contact maps aggregated over 20 subtelomeric loci with a non-coding subtelomeric region (±5 kb, 500 bp resolution), as in Fig. 2b. Colour scale: log2(observed/expected interaction frequency). *P* values (Student's *t*-test): changes in contact values between control and AP2-P KD for dashed pink (subtelomeric fold structure contact point) or grey (control area) boxes. **d**, Left (normalized interaction frequency) and middle (log2(observed/expected interaction frequency)): Micro-C contact maps over a section of chromosome 12 in late-stage parasites for subsampled (65 M contacts,

5 kb resolution) control and AP2-P KD. HP1-enriched virulence genes (red), non-virulence genes (black) and *ap2*-g (green) at the bottom. Right: magnified area of dashed rectangle shown at left. Presented otherwise as in Fig. 2a. **e**, Interchromosomal Micro-C contact maps for subsampled (65 M contacts) control or AP2-P KD in late-stage parasites aggregated over the 64 interchromosomal pairs of subtelomeric loci with a non-coding subtelomeric region (±15 kb, 500 bp resolution). For each pair, a 30-kb-wide contact map was centred at the MORC/AP2-P-enriched transition region. Contact maps are mirrored so the telomere is positioned towards the top left corner. **f**, log2(observed/expected interaction frequency) in late-stage parasites for interactions (i) within non-subtelomeric virulence gene-containing HP1 domains (*n* = 4,469, *P* = 0.0178), (ii) between non-subtelomeric virulence gene-containing HP1 domains (*n* = 8,516, *P* = 1.36 × 10⁻¹³⁴), (iii) between non-subtelomeric virulence gene-containing HP1 domains and subtelomeric virulence gene-containing HP1 domains (*n* = 13,711, *P* = 0), or (iv) between non-subtelomeric virulence gene-containing HP1 domains and non-coding subtelomeric loci (*n* = 3,883, *P* = 1.3 × 10⁻⁵⁰). Boxes represent the median and IQR, whiskers represent ±1.5× IQR. *P* values from two-sided *t*-tests are indicated. O/E, observed/expected.

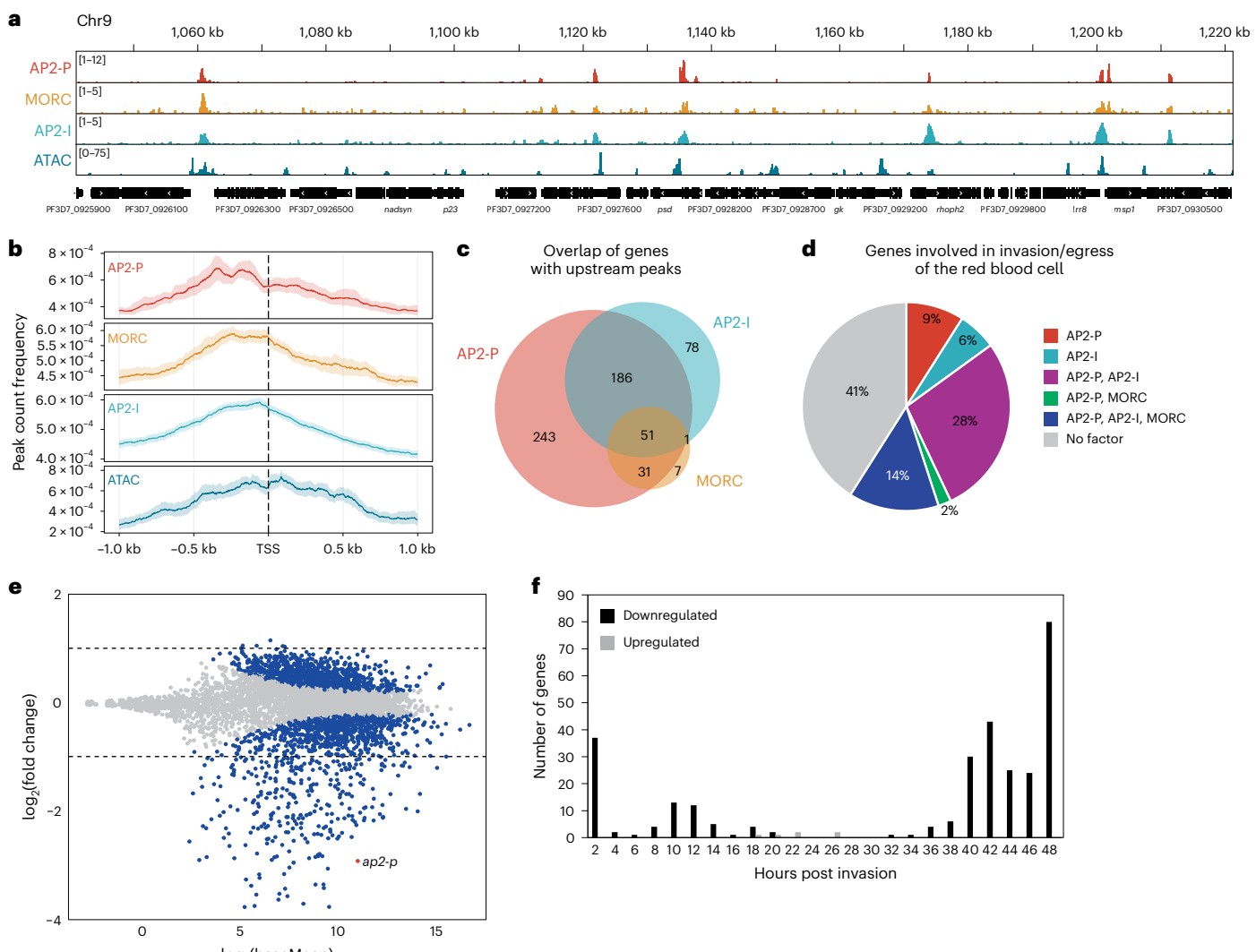

**Fig. 4 | AP2-P, AP2-I and MORC activate stage-specific genes. a**, ChIP-seq data showing enrichment (*y* axis, ChIP/Input) of AP2-P, MORC and AP2-I[46] in clonal late-stage parasites at a central region of chromosome 9. ATAC-seq data[101] from a closely corresponding stage showing chromatin accessibility [*y* axis, ATAC-seq (RPM)/gDNA (RPM)]. The *x* axis is DNA sequence, with genes represented by black boxes with white arrowheads to indicate transcription direction. **b**, Metagene plots showing average AP2-P and MORC enrichment in clonal late-stage AP2-P-3HA-*glmS* and MORC-3HA parasites, respectively, from 1 kb upstream to 1 kb downstream of the TSS of all genes with an upstream peak of AP2-P, excluding *var* genes. AP2-I ChIP-seq data[46] and ATAC-seq data[101] from stage-matched parasites is included. One replicate was used for the AP2-P and MORC ChIP datasets. **c**, Venn diagram showing overlap of genes bound by AP2-P, MORC and AP2-I in their upstream regions in late-stage parasites (Supplementary Data 1). **d**, Pie charts showing the proportion of genes that are important to the late stages of the parasite asexual replication cycle[48,49] (Supplementary Data 4) that are bound

by AP2-P, MORC and/or AP2-I in their upstream regions in late-stage parasites (Supplementary Data 1). **e**, Differential expression analysis of AP2-P KD. MA plot of log2(knockdown/control, M) plotted over the mean abundance of each gene (A) in late-stage parasites. Transcripts that were significantly higher (above *x* axis) or lower (below *x* axis) in abundance after AP2-P KD are highlighted in blue (*q* ≤ 0.05). *ap2-p* is highlighted in red. Dashed lines indicate a fold change ≥2 [log2(fold change) ≥1]. Three technical replicates were used for untreated and glucosamine-treated parasites. *P* values were calculated with a Wald test for significance of coefficients in a negative binomial generalized linear model as implemented in DESeq2 (ref. 98). *q* = Benjamini–Hochberg adjusted *P* value. **f**, Frequency plot showing the time in the red blood cell cycle (hours post invasion of the red blood cell) of peak transcript level (comparison to transcriptomics time course in ref. 47) for genes that are significantly downregulated (black) or upregulated (grey) more than 2-fold following AP2-P knockdown in late-stage parasites.

subtelomeric fold structure upstream of subtelomeric *var* genes (Fig. 3b). In particular, the subtelomeric focal contacts between genomic loci bound by AP2-P and MORC were disrupted, suggesting that AP2-P is responsible for anchoring these subtelomeric fold structures (Fig. 3c). In addition, AP2-P KD resulted in a significant reduction in intra- (Fig. 3d,f) and interchromosomal (Fig. 3e) interactions of HP1-enriched regions containing virulence genes (such as the *var* gene family) with (1) other HP1-enriched regions containing virulence genes in subtelomeric or central chromosomal regions and (2) non-coding subtelomeric HP1 domains. Although statistically significant, this decrease was much more modest for HP1-enriched

regions that do not contain virulence genes (Extended Data Fig. 5a). The reduction in trans-chromosomal contacts between subtelomeric regions was modest, suggesting that SIP2 or TRZ may also help to facilitate these interactions (Fig. 3e). Consistent with the fact that AP2-P and MORC are generally not enriched at the boundaries between euchromatin and HP1 heterochromatin (Fig. 2a), AP2-P KD did not affect insulation of HP1 self-associating domains that encompass virulence genes (Extended Data Fig. 5b). Taken together, these data suggest that AP2-P and MORC play a role in the formation of long-range intra- and interchromosomal interactions between heterochromatic regions.

## AP2-P, AP2-I and MORC activate stage-specific genes

In addition to their enrichment in subtelomeric heterochromatin, AP2-P and MORC bind to accessible euchromatic regions of the genome (Fig. 4a and Supplementary Data 1). Motif enrichment analysis of AP2-P ChIP-seq peaks identified a motif, 'GTGCA', that is very similar to the motif identified in ref. 45 (see reverse complement) and is shared with another ApiAP2 factor, AP2-I, which binds to the promoters of genes involved in late-stage parasite biology, such as red blood cell invasion[46] (Extended Data Fig. 6a,b). Comparison of AP2-I[46] with AP2-P and MORC ChIP-seq data revealed substantial overlap of these proteins throughout the genome (Extended Data Fig. 6c and Supplementary Data 1) and near the putative TSS of genes bound by AP2-P (Fig. 4a,b). Approximately 75% of all genes with an upstream peak of AP2-I also have AP2-P (Fig. 4c and Supplementary Data 1), and 90% of genes with an upstream peak of MORC also have AP2-P, AP2-I, or both, suggesting a dependency of MORC binding on an ApiAP2 factor (Fig. 4c and Supplementary Data 1).

Gene Ontology (GO) analysis of genes with an upstream AP2-P peak showed enrichment of 'cell–cell adhesion' ($q = 3.95 \times 10^{-11}$) and 'entry into host' ($q = 1.18 \times 10^{-9}$) categories, which include *var* genes and genes involved in red blood cell invasion (Supplementary Data 3). While *var* genes are silent in late-stage parasites, genes involved in red blood cell egress and invasion are specifically and highly transcribed at this stage[47–49] (Supplementary Data 4). Approximately 60% of these latter genes are bound by combinations of AP2-P, AP2-I and MORC (Fig. 4d). Indeed, AP2-P KD resulted in the significant (≥2-fold) upregulation of only 6 genes, but downregulation of 302 genes, suggesting that AP2-P plays a transcriptional activating role at this stage (Fig. 4e and Supplementary Data 2). Downregulated genes are most enriched for GO terms such as 'movement in host environment' ($q = 1.04 \times 10^{-22}$) and 'entry into host' ($q = 9.63 \times 10^{-22}$) (Supplementary Data 5), and most of them normally reach peak transcription levels in late-stage parasites[47] (Fig. 4f). Moreover, downregulated genes are enriched in AP2-P, MORC and AP2-I in their upstream regions (Extended Data Fig. 6d and Supplementary Data 1). Taken together, these data suggest that AP2-P, AP2-I and MORC form an activating complex that is essential for the transcription of genes that are specific and important during late stages of the red blood cell cycle.

## AP2-P activating complex defines euchromatic structures

We next investigated whether AP2-P and MORC are involved in the formation of euchromatic structures. In euchromatic regions of the parasite genome, we observed significant breaks in local contacts, or boundaries, that often overlap with genes showing high levels of transcription, such as *dblmsp* and *gap45* (Fig. 5a,b and Supplementary Data 6). Indeed, euchromatic boundary strength shows linear correlation with transcription levels of nearby genes (Extended Data Fig. 7a). Importantly, euchromatic boundaries show substantial overlap with genes associated with AP2-P, AP2-I and/or MORC peaks (Fig. 5a,c and Extended Data Fig. 7b). In fact, euchromatic boundary insulation correlates with enrichment of AP2-P, AP2-I and MORC (Fig. 5c), while non-coding subtelomeric boundary insulation correlates with enrichment of AP2-P, TRZ and MORC (Fig. 5c). These data suggest that different protein complex compositions may define different types of boundaries.

We further found that AP2-P-bound active genes have a higher boundary insulation score than genes with similar levels of transcription that are not bound by AP2-P (Fig. 5d and Supplementary Data 6). Moreover, AP2-P KD leads to weakening of euchromatic boundaries (Fig. 5e and Extended Data Fig. 7c). These data suggest that strong transcriptional activity could play a role in boundary formation, as seen in other microorganisms[33,50], but that AP2-P directly contributes to the structural maintenance of euchromatic boundaries.

Interestingly, we often observed long-range contacts anchored at euchromatic boundaries, bringing together multiple active genes (Supplementary Data 7). Such intrachromosomal long-range interactions can be seen between *sui1*, *act1* and *ron3* on chromosome 12 (Fig. 5f). On chromosome 2, *msp5* interacts with *ron6* and *cdpk1*, which are located across the centromere (Fig. 5f). One final example is on chromosome 10 where *chd1* interacts with *nprx*, *msp11* and PF3D7_1035200, which in turn interacts with *etramp10.2* (Extended Data Fig. 7d). Several of these loci also form interchromosomal contacts with each other, such as *msp5* (Chr2), which interacts with *sui1* and *act1* (Chr12, Extended Data Fig. 7e). Importantly, the loci that form these long-range contacts are enriched in AP2-P, AP2-I and MORC, and AP2-P KD reduced contact frequency between them (Fig. 5f,g and Extended Data Fig. 7d). Thus, AP2-P (and perhaps MORC and AP2-I) plays a role in the formation of euchromatic structural features that may influence gene transcription.

## Stage-specific structures associate with stage-specific genes

To determine whether euchromatic boundaries and long-range interactions change depending on the stage of the parasite, we performed Micro-C in early-stage parasites ('ring stage' in Fig. 1a) and observed similar data properties and quality to that from late-stage parasites (Extended Data Figs. 1a and 8a,b). Although weaker than in late-stage parasites, interactions among centromeres, telomeres and HP1-enriched regions containing virulence genes (Extended Data Fig. 8c–e), as well as the subtelomeric fold structures found in late-stage parasites (Extended Data Fig. 8f) are present in early-stage parasites. Weaker interactions in early-stage parasites were not due to differences

**Fig. 5 | AP2-P activating complex defines euchromatic boundaries and long-range interactions. a**, Micro-C contact maps (top) of central regions of chromosome 10 and 12 (130-kb-wide loci, 1 kb resolution) and corresponding AP2-P, MORC, AP2-I[46], TRZ[43] and HP1 (ref. 91) ChIP/input and ATAC-seq[101] tracks (bottom) from late-stage parasites. Cyan diamonds: Micro-C boundaries. Selected genes are indicated at the bottom, followed by colour scales: log₂(observed/expected interaction frequency) (top) and normalized interaction frequency (bottom). **b**, On-diagonal Micro-C contact map, aggregated over the 211 euchromatic boundaries (cyan diamonds, Supplementary Data 6) (±20 kb, 500 bp resolution) in late-stage parasites. Colour scales: log₂(observed/expected interaction frequency) (top) and normalized interaction frequency (bottom). **c**, Heat map showing AP2-P, MORC, AP2-I[46], TRZ[43] and HP1 (ref. 91) enrichment or ATAC-seq signal[101] around 317 boundaries (±5 kb, shown at the bottom with cyan diamonds, Supplementary Data 6) in late-stage parasites. Rows represent Micro-C boundaries ranked by insulation score in euchromatin, non-subtelomeric HP1-enriched domains, virulence gene-containing subtelomeric regions, or non-coding subtelomeric regions. The colours indicate the normalized ChIP-seq/ATAC-seq enrichment level. **d**, On-diagonal Micro-C contact maps, aggregated over the TSS of the 200 most highly expressed genes in late-stage parasites (±40 kb, 2 kb resolution) and associated (top) or not associated (bottom) with AP2-P (≤1 kb from an AP2-P peak). Colour scale: log₂(observed/expected interaction frequency). The dark red circle represents AP2-P binding, and the light red circle indicates the absence of AP2-P. The arrows indicate transcription activity. **e**, On-diagonal Micro-C contact maps, aggregated over all 211 euchromatin boundaries (cyan diamonds, ±20 kb, 500 bp resolution, Supplementary Data 6) in late-stage parasites. Colour scale: log₂(observed/ expected interaction frequency) in subsampled (65 M contacts) control (top) or AP2-P KD (bottom). $P$ values (Student's $t$-test) are shown for changes in contact values between control and AP2-P KD for areas highlighted by pink (boundaries, $P < 10^{-10}$) or grey (control area, $P = 0.84$) triangles. **f**, Micro-C contact map (top) of central regions of chromosome 12 and 2 and corresponding AP2-P, MORC, AP2-I[46], TRZ[43] and HP1 (ref. 91) ChIP/input ratio signals and ATAC-seq[101] data (bottom) in late-stage parasites. Cyan diamonds: Micro-C boundaries. Selected genes are indicated at the bottom of the contact map. Long-range interactions are indicated with black circles on the contact map. **g**, Off-diagonal Micro-C contact maps, aggregated over all 1,339 long-range interactions (±10 kb, 1 kb resolution) (Supplementary Data 7) and aggregated AP2-P, MORC, AP2-I[46], TRZ[43] and HP1 (ref. 91) ChIP/input signals from late-stage parasites. Colour scale: log₂(observed/expected interaction frequency) in the control map (left), subsampled control map (centre) or AP2-P KD map (right).

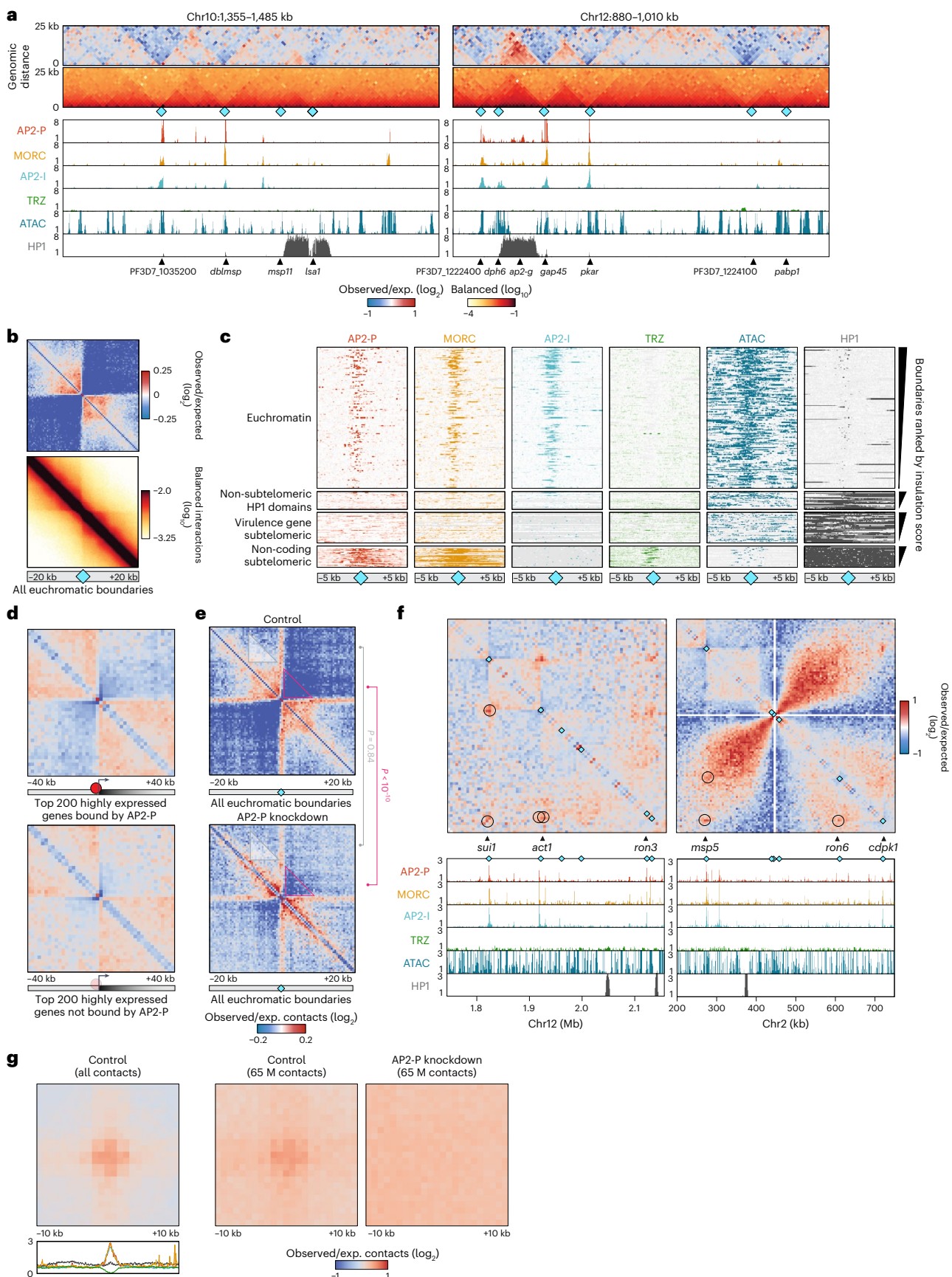

in data quantity or quality between the two time points (Supplementary Data 8, and Extended Data Figs. 1a and 8a,b). One possibility is that the organizational factors we elucidated in late-stage parasites are expressed at much lower levels in early-stage parasites (Extended Data Fig. 3d,e). Indeed, in our study, AP2-P and MORC ChIP-seq in early-stage parasites did not yield an acceptable signal-to-background ratio. Thus, we compared our Micro-C data from early-stage parasites to AP2-P ChIP-seq data from a slightly later time point[51] and found smaller but significant AP2-P peaks upstream of subtelomeric *upsB var* genes at the anchor point of the subtelomeric fold structure (Extended Data Fig. 8f,g). These data suggest that while AP2-P may play a role in the organization of subtelomeric *var* genes throughout the red blood cell cycle, its lower presence and binding in early-stage parasites might lead to weaker interactions among virulence gene-containing heterochromatic regions.

In contrast to heterochromatic structural features, euchromatic features change considerably between early- and late-stage parasites. We identified euchromatic boundaries (60; Fig. 6a,b and Supplementary Data 9) and long-range interactions (273; Fig. 6a,c and Supplementary Data 10) in early-stage parasites that dissipate in late-stage parasites, and vice-versa. Interestingly, the lower numbers of euchromatic structures in early-stage parasites correlate with overall lower levels of transcriptional activity compared with late-stage parasites[52,53]. Nonetheless, gene transcription disrupts local Micro-C contacts in a stage-specific manner (Fig. 6d). Surprisingly, we found that AP2-P[51] is enriched at euchromatic boundaries and long-range interactions in early- and late-stage parasites (Fig. 6a–c). These data suggest that AP2-P and associated factors contribute to heterochromatic and euchromatic structures over the course of the red blood cell cycle.

In both stages, genes near euchromatic boundaries have higher levels of transcription than those outside of boundaries, and boundary-associated genes that form long-range interactions are transcribed at even higher levels (Fig. 6e). Genes that form euchromatic long-range interactions have important functions in their respective stages[47]. Those in early-stage parasites are most involved in ribosome biogenesis, translation and fatty acid synthesis, while those in late-stage parasites are most involved in entry into host (invasion), mitosis and cell division (Supplementary Data 11). Thus, stage-specific euchromatic structures are associated with stage-specific gene transcription across the red blood cell cycle.

## Discussion

Here we adapted the Micro-C technique[33] to *P. falciparum* to map genome-wide contacts at near-nucleosome resolution, which allowed us to define two different types of heterochromatin: (1) HP1-heterochromatinized *ap2-g* or virulence genes (such as the *var* family) that are well insulated from neighbouring euchromatin and

form significant intra- and interchromosomal interactions and (2) other HP1-enriched genes that do not show these same characteristics. These data suggest that other as-yet unidentified protein factors are involved in insulation of and contacts among *ap2-g* and virulence genes (Extended Data Fig. 9). In addition, we identified two new chromatin structures in blood-stage parasites. The first are heterochromatic subtelomeric fold structures that are distinct from telomere loops (Extended Data Fig. 9) and could play a role in coordinating *var* gene alignment. The second are long-range intra- and interchromosomal interactions among stage-specific genes that could facilitate their temporal co-activation (Extended Data Fig. 9). Integration of genome-wide datasets and functional gene characterization identified the proteins facilitating these structures: AP2-P and MORC.

AP2-P and MORC are most enriched upstream of subtelomeric *var* genes at a Micro-C boundary between the HP1-enriched non-coding and coding domains, which is also enriched in TRZ and SIP2. This site is (1) the anchor point of a fold structure formed by the non-coding subtelomeric region and (2) the main point of contact between subtelomeric regions of the same and different chromosomes. Both types of structure are disrupted upon AP2-P KD and show weaker contact frequencies in early-stage parasites when AP2-P and MORC are expressed at very low levels, suggesting a direct role in heterochromatic structure and contact formation. This conclusion is supported by two recent studies showing a decrease in virulence gene clustering upon knockout of AP2-P[51] or knockdown of MORC[54]. Since telomere interactions facilitate chromosome end clustering, the AP2-P/MORC-mediated *var* gene-specific subtelomeric contact points could play a role in *var* gene biology.

Because the level to which we knocked AP2-P down did not result in death or cell cycle arrest, we were able to gain clear insight into its chromatin-related function in late-stage parasites. Importantly, AP2-P KD did not substantially affect *var* gene transcription, as observed elsewhere[51]; however, it is possible that AP2-P/MORC play a role in *var* gene genetics. The subtelomeric *upsB var* genes in particular undergo ectopic recombination during mitosis[25,26] that leads to antigenic diversity, which is crucial to maintaining chronic infection in the human host. Unlike V(D)J recombination among immunoglobulin genes[55], *var* gene recombination does not rely on homology[26]. Subtelomeric *var* genes are located at different distances from chromosome ends (mean 26 kb ± 11.9 kb); hence telomere clustering would not bring them into close proximity to each other. Regardless of the length of the non-coding subtelomeric region, the AP2-P/MORC-bound fold anchors are ~8 kb downstream of the telomere (mean 8.3 kb ± 1.7kb) and ~2 kb upstream of the first *var* gene on the chromosome end (mean 2.1 kb ± 0.22kb) (Fig. 2b,c). As it can topologically entrap DNA and multimerize in other eukaryotes[56], *Pf*MORC could bind, dimerize and cinch up the non-coding subtelomeric sequence to bring the *var* gene close to

**Fig. 6 | Stage-specific euchromatic structural features are associated with stage-specific gene transcription. a**, Micro-C contact map (5 kb resolution) of a central region of chromosome 8 in early- (left) and late-stage (right) wild-type parasites. Micro-C boundaries are indicated with cyan diamonds. Selected genes are indicated at the bottom. Early[51] and late AP2-P ChIP-seq and stage-matched ATAC-seq[101] data are shown at the bottom of each contact map. Long-range interactions identified with chromosight[83] are indicated with black circles on the contact map. A dashed circle indicates a lost long-range contact. Colour scale indicates log$_2$(observed/expected interaction frequency). **b**, On-diagonal Micro-C contact maps, aggregated over the 60 early-stage (magenta diamond, Supplementary Data 9) and 211 late-stage (cyan diamond, Supplementary Data 6) euchromatic boundaries identified by chromosight[83] (±10 kb, 1 kb resolution) in early- and late-stage parasites. Colour scale indicates log$_2$(observed/expected interaction frequency). 1D aggregated AP2-P ChIP/input signals from early[51]- and late-stage parasites are shown below the 2D aggregated Micro-C map. Euchr., euchromatic. **c**, Off-diagonal Micro-C contact maps, aggregated over all early- (Supplementary Data 10) and late-stage (Supplementary Data 7)

long-range interactions (±10 kb, 1 kb resolution) identified in early- and late-stage parasites. Colour scale indicates log$_2$(observed/expected interaction frequency). 1D aggregated AP2-P ChIP/input signals from early[51]- and late-stage parasites are shown below the 2D aggregated Micro-C map. Triangles indicate the anchor point of the interaction. **d**, On-diagonal Micro-C contact maps from early- and late-stage parasites, aggregated over the 500 most highly transcribed genes in early- or late-stage parasites (±15 kb, 1 kb resolution). Colour scale indicates log$_2$(observed/expected interaction frequency). **e**, Transcript levels (log$_{10}$(FPKM)) of genes outside of Micro-C boundaries (grey; $n = 5,254$ for early; $n = 5,087$ for late), associated with a euchromatic Micro-C boundary (cyan; $n = 25$ for early; $n = 40$ for late), or associated with a euchromatic long-range interaction whose anchor points overlap with a euchromatic boundary (orange; $n = 24$ for early, $n = 176$ for late) (see Methods) in early- and late-stage parasites (Supplementary Data 6, 7, 9 and 10). Statistical difference in levels of expression between each set of genes was assessed using two-sided *t*-tests. Boxes represent the median and IQR, whiskers represent ±1.5× IQR. FPKM, fragments per kilobase of transcript per million mapped reads.

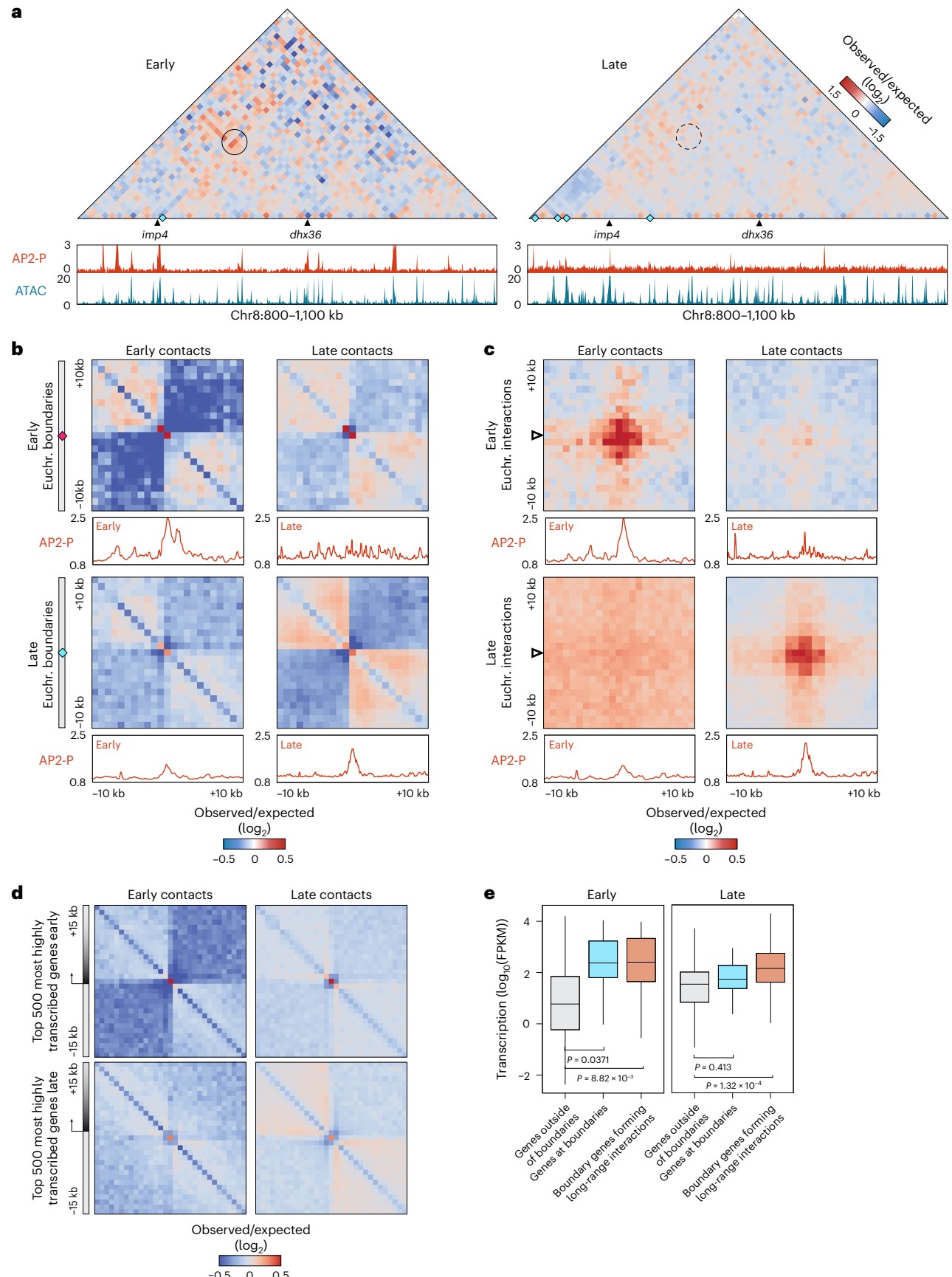

the telemere. Then, AP2-P, MORC and SIP2 or TRZ could facilitate intra- and interchromosomal subtelomeric interactions to bring *var* genes together so that they can recombine properly (Extended Data Fig. 9).

While heterochromatic structures are present in early- and late-stage parasites, euchromatic boundaries and long-range interactions change, suggesting that these structures are involved in stage-specific gene activation. Indeed, euchromatic boundary strength correlates with the level of gene transcription, and associated long-range interactions are found among highly transcribed genes at each stage. AP2-P is enriched at these stage-specific euchromatic structures and plays a role in their formation and in the activation of associated genes. AP2-P shows synergism with AP2-I and MORC in these roles at least in late-stage parasites (Extended Data Fig. 9).

Long-range interactions between genes have been observed in different eukaryotes and can be associated with co-repression or co-expression. In humans, loops, often mediated by cohesin and CTCF, between *cis* regulatory elements can activate genes in specific cellular or developmental contexts[57–59]. In contrast, *P. falciparum* does not have CTCF[29] and long-range euchromatic contacts do not seem to involve an enhancer element or cohesin[60]. AP2-P/MORC-mediated euchromatic long-range contacts more closely resemble interactions between co-expressed and/or functionally related genes in *Drosophila*[61–63], where looping between distant genes helps to coordinate and fine-tune their expression during embryonic development[62]. We propose a similar mechanism in *P. falciparum* between stage-specific genes. Transcription might not always happen from individual promoters in a dispersed manner in the nucleus, but in foci where the transcriptional machinery is concentrated[7,9]. In other eukaryotes, these foci could represent phase-separated compartments where multiple promoters associate with enhancers[64,65] or nuclear speckles, which are enriched in transcription and transcript processing factors[66]. Recently, such activating compartments were observed for variant surface glycoprotein genes[67,68] and RNAPII transcriptional start sites[69] in the parasite *Trypanosoma brucei*.

In *P. falciparum*, gene relocation to transcription machinery-enriched compartments may facilitate rapid stage-specific activation. Association of functionally related genes has also been observed in sporozoites, the mosquito-to-human transmission stage[20]. Thus, spatio-temporal regulation of genes may be crucial to driving the entire parasite life cycle. Further investigation of the properties, dynamics and significance of the coalescence of co-regulated genes is needed.

Among the Apicomplexan parasites studied, MORC has been observed via mass spectrometry to interact with many different ApiAP2 factors[34,37,51,54,70–72], yet the nature and function of these interactions is unclear in *Plasmodium*. We provide genome-wide evidence for the cooperation between MORC and three ApiAP2 factors, which presumably provide sequence specificity. While MORC is associated with heterochromatin and transcriptional repression in other eukaryotes[35,37], we and others[34,51,54,72] find AP2-P and *Pf*MORC at silent *and* active genes. This versatile binding probably results from their association with different DNA-binding factors such as SIP2 and AP2-I. Indeed, cooperative binding of two ApiAP2 factors and MORC plays an important role in the repression of stage-specific genes in a related apicomplexan parasite, *Toxoplasma gondii*[37,70]. Our data also show that combinatorial ApiAP2 binding could provide a higher capacity for complex transcriptional regulation than a 'one transcription factor to one gene' model, especially in this organism that has relatively few sequence-specific DNA-binding factors[73].

Whether in eu- or heterochromatin, AP2-P and MORC are enriched at anchor points for long-range DNA–DNA interactions. We therefore propose a unifying model where AP2-P and MORC facilitate higher-order chromatin structure, which influences different context-specific chromatin-associated processes such as recombination and transcription. In this way, MORC and AP2-P may replace organizational proteins such as CTCF, which *Plasmodium* lacks. Indeed,

*Pf*MORC has a unique structure compared with its orthologues, containing a Kelch beta propeller domain that may facilitate complex interactions[74].

Our study of genome organization in a eukaryotic parasite with a massive global health impact has shed light on divergent and potentially targetable molecular mechanisms used to achieve similar complex phenomena observed in model eukaryotes: spatial co-regulation of functionally related genes.

## Methods

### Parasite culture

Blood-stage 3D7 or NF54 *P. falciparum* parasites were cultured as previously described[21]. Briefly, parasites were cultured in human RBCs supplemented with 10% v/v Albumax I (Thermo Fisher, 11020), hypoxanthine (0.1 mM final concentration, C.C.Pro Z-41-M) and 10 mg gentamicin (Sigma, G1397) at 4% haematocrit and under 5% $O_2$, 5% $CO_2$ at 37 °C. Parasites were synchronized by sorbitol (5%, Sigma, S6021) lysis during the ring stage, followed by plasmagel (Plasmion, Fresenius Kabi) enrichment for late blood stages 24 h later. Another sorbitol treatment 6 h afterwards places the 0 hpi (hours post invasion of the red blood cell) time point 3 h after the plasmagel enrichment. Thus, the window of synchronicity for cultures is ±3 h. In this paper, 'early-stage', 'middle-stage' and 'late-stage' refers to 12, 24 and 36 hpi ±3 h, respectively. Parasite development was monitored by Giemsa staining. Parasites were collected at 1–5% parasitemia.

### Micro-C

The Micro-C protocol was performed as previously described[33]. NF54 WT parasites were synchronized, and at least 5 replicates (~$1.5 × 10^{10}$ early-stage each and ~$8 × 10^8$ late-stage each) were collected. While concentrated blood was used to minimize human genomic DNA contamination, further purification with Plasmodipur filters (Europroxima, 8011) was needed for early-stage parasites. With these filters, ~80% of total reads sequenced mapped to the *P. falciparum* genome, as opposed to ~10% without the filters. To filter, most of the media was removed from the parasite culture. The infected RBCs (iRBCs) were transferred to a 20 ml syringe connected to a Plasmodipur filter and slowly passed through the filter. A single filter was used per 5 ml of packed iRBC diluted in ~40 ml media. After filtration (or without), parasites were centrifuged and lysed with saponin (0.075% in Dulbecco's phosphate-buffered saline (DPBS)) and washed with DPBS at 37 °C. Parasites were resuspended in DPBS at 25 °C and cross-linked for 10 min by adding methanol-free formaldehyde (Thermo Fisher, 28908) to 1% final concentration with gentle agitation. The reaction was quenched by adding 1 M Tris-HCl pH 7.5 to a final concentration of 0.75 M and incubating at 25 °C for 5 min with gentle agitation. Parasites were centrifuged for 5 min at 3,250 *g*, washed with DPBS at 25 °C, resuspended in the second cross-linking solution (3 mM disuccinimidyl glutarate; Thermo Fisher, 20593, in DPBS) and incubated for 45 min at 25 °C with agitation. The reaction was quenched with 1 M Tris-HCl pH 7.5 to a final concentration of 0.75 M and incubated at 25 °C for 5 min with agitation. The double cross-linked parasites were washed with DPBS at 25 °C, and the pellets were snap frozen and stored at −80 °C until further use.

For each time point, one replicate was split into four and used to titrate the MNase (Worthington Biochem, LS004798). Each pellet was resuspended in 1 ml of MB#1 [50 mM NaCl, 10 mM Tris-HCl pH 7.5, 5 mM $MgCl_2$, 1 mM $CaCl_2$, 0.2% NP-40, 1× protease inhibitor (PI, Roche, 4693159001)] and incubated on ice for 20 min with regular flicking. The cells were centrifuged (10,000 *g*, 5 min, 4 °C), washed with MB#1, resuspended in 760 µl of MB#1 and aliquoted in 4 × 190 µl (each aliquot corresponding to ~$4 × 10^9$ parasites for early-stage and ~$2 × 10^8$ for late-stage parasites). MNase (10 µl) at the chosen concentration (corresponding to 10 U, 20 U, 50 U and 100 U) was added, and the mixture was incubated for exactly 10 min at 37 °C with continuous shaking at 850 r.p.m. To stop the reaction, 500 mM EGTA

(Thermo Fisher, J60767) was added to a final concentration of 4 mM and was followed by a 10 min incubation at 65 °C. The digested nuclei were then washed twice (5 min centrifugation at 16,000 g, 4 °C) with MB#2 (50 mM NaCl, 10 mM Tris-HCl pH 7.5, 10 mM MgCl₂). The nuclei were resuspended in a digestion mix [10 mM Tris-HCl pH 7.5, 1 mM EDTA, 1% SDS, 1 mg ml⁻¹ Proteinase K (Thermo Fisher EO0491), 0.125 mg ml⁻¹ RNase A (Thermo Fisher, EO0491)] and de-cross-linked for 10 h at 65 °C. The samples were centrifuged (16,000 g, 10 min, 4 °C) and DNA contained in the supernatant was extracted using the DNA Clean and Concentrator kit (Zymo, D4013). The fragment sizes were assessed using an Agilent BioAnalyzer to choose the appropriate MNase concentration yielding a 90/10 monomer/dimer ratio.

The following steps were performed with 3 and 4 replicates for early- and late-stage parasites, respectively. The cell pellet (~1.5 × 0¹⁰ early- and ~8 × 10⁸ late-stage parasites) was lysed and digested with the chosen concentration of MNase, as described above. After MNase digestion, all buffer volumes were doubled for early-stage parasites. After the MB#2 washes, each pellet was centrifuged (16,000 g, 5 min, 4 °C), resuspended in 45 µl of end-chewing mix [1× NEB Buffer 2.1, 2 mM ATP (Thermo Fisher, R1441), 5 mM dithiothreitol (Thermo Fisher, 707265ML), 0.5 U µl⁻¹ T4 PNK (NEB, M0201S)] and incubated at 37 °C for 15 min on an agitator with interval mixing (800 r.p.m. 10 s, resting 30 s). A volume of 5 µl of 5 U µl⁻¹ Klenow fragment (NEB, M0210L) was added to achieve a final concentration of 0.5 U µl⁻¹ and the mixture incubated at 37 °C for 15 min with the same agitation programme as the previous step. A volume of 25 µl of end-labelling mix [0.2 mM Biotin-dATP (Jena Bioscience, NU-835-BIO14-S), 0.2 mM Biotin-dCTP (Jena Bioscience, NU-809-BIOX-S), 0.2 mM dTTP (Thermo Fisher, 18255018), 0.2 mM dGTP (Thermo Fisher, 18254011), 0.1 mg ml⁻¹ BSA (Invitrogen, AM2616)] was added to the mixture and the mixture incubated at 25 °C for 45 min with agitation (800 r.p.m. 10 s, resting 30 s). EDTA (500 mM; Invitrogen, AM9260G) was added to a final concentration of 30 mM and the mixture incubated for 30 min at 65 °C. The pellet was centrifuged (10,000 g, 5 min, 4 °C) and washed with cold MB#3 (10 mM Tris-HCl pH 7.5, 10 mM MgCl₂). The biotinylated DNA fragments were then covalently linked by proximity ligation by resuspending the pellet in 250 µl of ligation mix [1× T4 DNA ligase buffer (NEB, B0202S), 0.1 mg ml⁻¹ BSA (Invitrogen, AM2616), 33 U µl⁻¹ T4 DNA ligase (NEB, M0202L)] and incubating at 25 °C for 2.5 h with slow rotation. The ligated DNA fragments were then centrifuged (16,000 g, 5 min, 4 °C), and the biotin-dNTPs were removed from unligated ends by resuspending the pellet in 100 µl of 1× NEB buffer #1 (NEB, B7001S) and 5 U µl⁻¹ exonuclease III (NEB, M0206L) and incubating for 15 min at 37 °C with agitation (800 r.p.m. 10 s, resting 30 s).

The DNA was purified, 12.5 µl of 10% SDS (Thermo Fisher, 15553027) was added, and the DNA was de-crosslinked by incubating for 10 h at 65 °C. RNase A (Thermo Fisher, EN0531) was added to a final concentration of 0.2 mg ml⁻¹ and the mixture incubated for 2 h at 37 °C. Proteinase K (Thermo Fisher, EO0491) was added to a final concentration of 0.2 mg ml⁻¹ and the mixture incubated for 2 h at 55 °C. After centrifugation (16,000 g, 10 min, 4 °C), the DNA contained in the supernatant was recovered using the DNA Clean and Concentrator (Zymo, D4013, using a 5:1 ratio of DNA-binding buffer to sample). The ligated DNA was mixed with loading dye (Thermo Fisher, R1161) and loaded onto a 2% TAE agarose gel to separate dimers from monomers. The gel was run at 100 V until a satisfying separation was obtained. A band from 250–400 bp was cut and DNA was extracted in 150 µl of double-distilled H₂O using the Gel DNA Recovery kit (Zymo, D4007).

The biotinylated dimers were isolated with Dynabeads MyOne Streptavidin C1 (Thermo Fisher, 65001). The beads were washed with 500 µl TBW (1 M NaCl, 5 mM Tris-HCl pH 7.5, 0.5 mM EDTA, 0.10% Tween 20) on a magnet and resuspended in 150 µl 2× BW (2 M NaCl, 10 mM Tris-HCl pH 7.5, 1 mM EDTA). The ligated fragments were added and rotated for 20 min at 25 °C. The beads were then washed twice with 500 µl TBW at 55 °C on an agitator (2 min, 800 r.p.m. 10 s, resting 30 s)

and once with elution buffer (EB) (10 mM Tris-HCl pH 7.5) at 25 °C. The library preparation was performed with the samples on beads using the NEBNext Ultra II Directional DNA Library Prep kit (NEB, E7645), with several modifications. First, the end repair and adaptor ligations steps were performed with interval mixing (800 r.p.m. 10 s, resting 30 s). After the adaptor ligation, the beads were washed twice with TBW at 55 °C and once with EB (as described above) and resuspended in 20 µl EB for the PCR reaction. No size-selection step was performed. Finally, after the PCR, the beads were placed again on a magnet, and the supernatant was purified with AMPure beads as recommended.

This protocol was repeated for the AP2-P KD in the AP2-P-3HA strain (clone H5 in 3D7 background). AP2-P-3HA-glmS (clone H5) parasites were synchronized, and the culture was split into two at 12 hpi. Glucosamine (Sigma, G1514, final concentration 2.5 mM) was added to one culture for two rounds of parasite replication (~96 h). Parasites were then resynchronized and 4 technical replicates (with and without glucosamine) were collected at 36 hpi.

### Generation of strains

The AP2-P-3HA-glms strain was generated using a two-plasmid system (pUF1 and pL7) based on the CRISPR/Cas9 system previously described[75]. A 3D7 wild-type bulk ring-stage culture was transfected with 25 µg pUF1-Cas9 and 25 µg of pL7-PF3D7_1107800-3HA-glmS containing a single guide RNA (sgRNA)-encoding sequence targeting the 3′ untranslated region (UTR) of PF3D7_1107800 (Supplementary Data 12). The pL7-PF3D7_1107800-3HA-glmS plasmid also contained a homology repair construct synthesized by GenScript Biotech (Supplementary Data 12). This homology repair construct comprises a 3× haemagglutinin (3HA)-encoding sequence followed by a glmS ribozyme-encoding sequence[44], which are flanked by ~300 bp homology repair regions upstream and downstream of the Cas9 cut site. Two shield mutations were made in the upstream homology repair region to prevent further Cas9 cleavage of the modified locus. The sgRNA sequence was designed using Protospacer[76]. The sgRNA sequence uniquely targeted a single sequence in the genome. After transfection, drug selection was applied for 5 days at 2.67 nM WR99210 (Jacobus Pharmaceuticals) and 1.5 µM DSM1 (MR4/BEI Resources). Parasites reappeared ~3 weeks after transfection, and 5-fluorocytosine was used to negatively select the pL7 plasmid.

The MORC-3HA and AP2-P-3HA-glmS:MORC-GFP strains were generated using the method of selection-linked integration (SLI) previously described[77], with a slight modification wherein a GSG-encoding sequence (5′-GGTAGTGGT-3′) was added directly upstream of the T2A skip peptide-encoding sequence to enhance cleavage of the tagged protein from the downstream drug selection protein. The homology region corresponding to the 771 bp at the 3′ end of morc (PF3D7_1468100) was amplified using JB_155F/R (Supplementary Data 12). The PCR fragment was then fused to a sequence encoding a 3HA or GFP epitope tag, followed by a skip peptide, and then by the neomycin resistance marker. For the MORC-3HA strain, a 3D7 wild-type bulk ring-stage culture was transfected with 50 µg of pSLI plasmid (containing a yeast dihydroorotate dehydrogenase selection marker) and selected first with 1.5 µM DSM1 (MR4/BEI Resources). Parasites reappeared ~3 weeks after transfection and positive selection for integration was performed via the addition of 400 µg ml⁻¹ G418 (Sigma, G8168). For the AP2-P-3HA-glmS:MORC-GFP strain, a clonal (clone H5) culture of AP2-P-3HA-glmS was transfected at ring stage with 50 µg of pSLI plasmid (containing a blasticidin selection marker) and selected first with 5 µg ml⁻¹ blasticidin (Invivogen, ant-bl-05). Parasites reappeared ~5 weeks after transfection and positive selection for integration was performed via the addition of 400 µg ml⁻¹ G418 (Sigma, G8168).

All cloning was performed using KAPA HiFi DNA Polymerase (Roche, 07958846001), In-Fusion HD Cloning kit (Clontech, 639649) and XL10-Gold Ultracompetent E. coli (Agilent Technologies, 200315). All parasite lines were cloned by limiting dilution, and integration at

the targeted genomic locus was confirmed by PCR (Extended Data Figs. 3b,c and 4g, and Supplementary Data 12) and Sanger sequencing.

## Protein fractionation and western blot analysis

Parasites were washed once with DPBS (Thermo Fisher, 14190), then resuspended in cytoplasmic lysis buffer [25 mM Tris-HCl pH 7.5, 10 mM NaCl, 1.5 mM MgCl$_2$, 1% IGEPAL CA-630 and 1× PI (Roche, 4693132001)] at 4 °C and incubated on ice for 30 min. The cytoplasmic lysate was cleared by centrifugation (13,500 $g$, 10 min, 4 °C). The pellet (containing the nuclei) was resuspended in 3.3 times less volume of nuclear extraction buffer (25 mM Tris-HCl pH 7.5, 600 mM NaCl, 1.5 mM MgCl$_2$, 1% IGEPAL CA-630, PI) than cytoplasmic lysis buffer at 4 °C, transferred to 1.5 ml sonication tubes (Diagenode, C30010016, 300 µl per tube) and sonicated for 5 min total (10 cycles of 30 s on/off) in a Diagenode Pico Bioruptor at 4 °C. This nuclear lysate was cleared by centrifugation (13,500 $g$, 10 min, 4 °C). Protein samples were supplemented with NuPage Sample Buffer (Thermo Fisher, NP0008) and NuPage Reducing Agent (Thermo Fisher, NP0004) and denatured for 10 min at 70 °C. Proteins were separated on a 4–12% Bis-Tris NuPage gel (Thermo Fisher, NP0321) and transferred to a PVDF membrane with a Trans-Blot Turbo Transfer system (Bio-Rad). The membrane was blocked for 1 h with 1% milk in PBST (PBS, 0.1% Tween 20) at 25 °C. HA-tagged proteins, GFP-tagged proteins and histone H3 were detected with anti-HA (Abcam, ab9110, 1:1,000 in 1% milk-PBST), anti-GFP (Chromotek, PABG1) and anti-H3 (Abcam, ab1791, 1:2,500 in 1% milk-PBST) primary antibodies, respectively, followed by donkey anti-rabbit secondary antibody conjugated to horseradish peroxidase ('HRP', Sigma, GENA934, 1:5,000 in 1% milk-PBST). For some blots, HA-tagged antibodies were detected with an anti-HA-HRP antibody (Cell Signaling, C29F4 HRP Conjugate 14031). HRP signal was developed with SuperSignal West Pico Plus or Femto chemiluminescent substrate (Thermo Fisher, 34580 or 34096, respectively) and imaged with a ChemiDoc XRS+ system (Bio-Rad). Bands on the western blots were quantified using the Bio-Rad Image Lab software.

## Chromatin immunoprecipitation and sequencing

Clonal populations of AP2-P-3HA-*glmS* (two replicates: $8 \times 10^8$ and $7 \times 10^8$) and MORC-3HA-*glmS* (two replicates: $2.5 \times 10^9$ and $10^9$) parasites were tightly synchronized and collected at 36 hpi. Parasite culture was centrifuged at 800 $g$ for 3 min at 25 °C. Medium was removed and the RBCs were lysed with 10 ml 0.075% saponin (Sigma, S7900) in DPBS at 37 °C. The parasites were centrifuged at 3,250 $g$ for 3 min at 25 °C and washed with 10 ml DPBS at 37 °C. For the AP2-P-3HA-*glmS* parasites, the supernatant was removed, and the parasite pellet was resuspended in 10 ml PBS at 25 °C. The parasites were cross-linked by adding methanol-free formaldehyde (Thermo Fisher, 28908) (final concentration 1%) and incubating with gentle agitation for 10 min at 25 °C. The cross-linking reaction was quenched by adding glycine (final concentration 125 mM; Sigma, G8899) and incubating with gentle agitation for 5 min at 25 °C. Parasites were centrifuged at 3,250 $g$ for 5 min at 4 °C and the supernatant removed. Parasite pellets were snap frozen and stored at −80 °C.

For the MORC-3HA-*glmS* parasites after saponin lysis, the supernatant was removed, and the parasite pellet was resuspended in 20 ml of PBS at 25 °C. MgCl$_2$ (Invitrogen, AM9530G) was added to a final concentration of 1 mM. A volume of 80 µl of ChIP Cross-link Gold (Diagenode, C01019027) was added and the sample was incubated at 25 °C for 30 min with gentle agitation. Parasites were centrifuged at 3,250 $g$ for 5 min at 4 °C and the supernatant removed. The pellet was washed twice with DPBS at 4 °C and centrifuged at 3,250 $g$ for 5 min at 4 °C. The parasites were resuspended in 20 ml DPBS at 4 °C and were further cross-linked by adding methanol-free formaldehyde (Thermo Fisher, 28908) (final concentration 1%) and incubating with gentle agitation for 15 min at 25 °C. The cross-linking reaction was quenched by adding glycine (final concentration 125 mM; Sigma, G8899) and incubating with gentle agitation for 5 min at 25 °C. Parasites were centrifuged at

3,250 $g$ for 5 min at 4 °C and the supernatant removed. Parasite pellets were snap frozen and stored at −80 °C.

For each ChIP, 200 µl of Protein G Dynabeads (Invitrogen, 10004D) were washed twice with 1 ml ChIP dilution buffer (16.7 mM Tris-HCl pH 8, 150 mM NaCl, 1.2 mM EDTA pH 8, 1% Triton X-100, 0.01% SDS) using a DynaMag magnet (Thermo Fisher, 12321D). The beads were resuspended in 1 ml ChIP dilution buffer with 8 µg of anti-HA antibody (Abcam, ab9110) and incubated on a rotator at 4 °C for 6 h.

The cross-linked parasites were resuspended in 4 ml of lysis buffer (10 mM HEPES pH 8, 10 mM KCl, 0.1 mM EDTA pH 8, PI) at 4 °C, and 10% Nonidet-P40 was added (final concentration 0.25%). The parasites were lysed in a prechilled dounce homogenizer (100 strokes). The lysates were centrifuged for 10 min at 13,500 $g$ at 4 °C, the supernatant was removed, and the pellet was resuspended in 3.6 ml SDS lysis buffer (50 mM Tris-HCl pH 8, 10 mM EDTA pH 8, 1% SDS, PI) at 4 °C. The liquid was distributed into 1.5 ml sonication tubes (Diagenode, C30010016, 300 µl per tube) and sonicated for 12 min total (24 cycles of 30 s on/off) in a Diagenode Pico Bioruptor at 4 °C. The sonicated extracts were centrifuged at 13,500 $g$ for 10 min at 4 °C and the supernatant, corresponding to the chromatin fraction, was kept. The DNA concentration for each time point was determined using the Qubit dsDNA High Sensitivity Assay kit (Thermo Fisher, Q32851) with a Qubit 3.0 fluorometer (Thermo Fisher). For each time point, chromatin lysate corresponding to 100 ng of DNA was diluted in SDS lysis buffer (final volume 200 µl) and kept as 'input' at −20 °C. Chromatin lysate corresponding to 8 µg (for AP2-P-3HA-*glmS*) and 10 µg (for MORC-3HA) of DNA was diluted 1:10 in ChIP dilution buffer at 4 °C.

Using a DynaMag magnet, the antibody-conjugated Dynabeads were washed twice with 1 ml ChIP dilution buffer and resuspended in 100 µl of ChIP dilution buffer at 4 °C. Then the washed antibody-conjugated Dynabeads were added to the diluted chromatin sample and incubated overnight with rotation at 4 °C. The beads were collected on a DynaMag into 8 different tubes per sample, the supernatant was removed, and the beads in each tube were washed for 5 min with gentle rotation with 1 ml of the following buffers, sequentially:

- o Low-salt wash buffer (20 mM Tris-HCl pH 8, 150 mM NaCl, 2 mM EDTA pH 8, 1% Triton X-100, 0.1% SDS) at 4 °C.
- o High-salt wash buffer (20 mM Tris-HCl pH 8, 500 mM NaCl, 2 mM EDTA pH 8, 1% Triton X-100, 0.1% SDS) at 4 °C.
- o LiCl wash buffer (10 mM Tris-HCl pH 8, 250 mM LiCl, 1 mM EDTA pH 8, 0.5% IGEPAL CA-630, 0.5% sodium deoxycholate) at 4 °C.
- o TE wash buffer (10 mM Tris-HCl pH 8, 1 mM EDTA pH 8) at 25 °C.

After the washes, the beads were collected on a DynaMag, the supernatant was removed, and the beads for each time point were resuspended in 800 µl of elution buffer and incubated at 65 °C for 30 min with agitation (1,000 r.p.m. 30 s on/off). The beads were collected on a DynaMag and the eluate, corresponding to the 'ChIP' samples, was transferred to a different tube.

For purification of the DNA, both 'ChIP' and 'Input' samples were incubated for -10 h at 65 °C to reverse the cross-linking. TE buffer (200 µl) followed by 8 µl of RNase A (Thermo Fisher, EN0531) (final concentration of 0.2 mg ml$^{-1}$) were added to each sample, which was then incubated for 2 h at 37 °C. Proteinase K (4 µl; New England Biolabs, P8107S) (final concentration of 0.2 mg ml$^{-1}$) was added to each sample, which was then incubated for 2 h at 55 °C. A volume of 400 µl phenol:chloroform:isoamyl alcohol (25:24:1) (Sigma, 77617) was added to each sample, which was then mixed by vortexing and centrifuged for 10 min at 13,500 $g$ at 4 °C to separate phases. The aqueous top layer was transferred to another tube and mixed with 30 µg glycogen (Thermo Fisher, 10814) and 5 M NaCl (200 M final concentration). Ethanol (800 µl, 100%) at 4 °C was added to each sample, which was then incubated at −20 °C for 30 min. The DNA was pelleted by centrifugation for 10 min at 13,500 $g$ at 4 °C, washed with 500 µl 80% ethanol at 4 °C and centrifuged for 5 min at 13,500 $g$

at 4 °C. After removing the ethanol, the pellet was dried at 25 °C and all DNA for each sample was resuspended in 30 μl 10 mM Tris-HCl pH 8 total. The DNA concentration and average size of the sonicated fragments were determined using a DNA high sensitivity kit and the Agilent 2100 Bioanalyzer. Libraries for Illumina next generation sequencing were prepared using the MicroPlex library preparation kit (Diagenode, C05010014), with KAPA HiFi polymerase (KAPA Biosystems) substituted for the PCR amplification. Libraries were sequenced on the NextSeq 500 platform (Illumina).

### RNA extraction and stranded RNA sequencing
An AP2-P-3HA-*glmS* clone was synchronized simultaneously and the culture was split into two at 12 hpi. Glucosamine (Sigma, G1514; final concentration 2.5 mM) was added to one culture for two rounds of parasite replication (~96 h). Parasites were then resynchronized and 3 technical replicates (with and without glucosamine) were collected at 12 or 36 hpi. RBCs were lysed in 0.075% saponin (Sigma, S7900) in PBS at 37 °C, centrifuged at 3,250 *g* for 5 min, washed in PBS, centrifuged at 3,250 *g* for 5 min, and resuspended in 700 μl QIAzol reagent (Qiagen, 79306). RNA was extracted using an miRNeasy Mini kit (Qiagen, 1038703) with the recommended on-column DNase treatment. Total RNA was poly(A) selected using the Dynabeads mRNA Purification kit (Thermo Fisher, 61006). Library preparation was performed using the NEBNext Ultra II Directional RNA Library Prep Kit for Illumina (New England Biolabs, E7760S), and paired-end sequencing was performed on the Nextseq 500 platform (Illumina).

### Parasite growth assay
Parasite growth was measured as described previously[78]. Briefly, two AP2-P-3HA-*glms* clones and a WT clone were tightly synchronized. Each culture was split and glucosamine (Sigma, G1514, 2.5 mM final concentration) was added to one half for ~96 h before starting the growth curve. The parasites were tightly resynchronized and diluted to ~0.2% parasitaemia (5% haematocrit) at ring stage. The growth curve was performed in a 96-well plate (200 μl culture per well) with 3 technical replicates per condition. Every 24 h, 5 μl of the culture were fixed in 45 μl of 0.025% glutaraldehyde in PBS for 1 h at 4 °C. After centrifuging at 800 *g* for 5 min, free aldehyde groups were quenched by resuspending the iRBC pellet in 200 μl of 15 mM NH$_4$Cl in PBS. A 1:10 dilution of the quenched iRBC suspension was incubated with Sybr Green I (Sigma, S9430) to stain the parasite nuclei. Quantification of the iRBCs was performed on a CytoFLEX S cytometer (Beckman Coulter) and analysis with FlowJo Software (Supplementary Data 13).

### Co-immunoprecipitation
Co-immunoprecipitation and western blot were performed as described previously[60] with minor modifications. Briefly, AP2-P-3HA-*glmS*:MORC-GFP parasites were synchronized, then collected and cross-linked at 36 hpi ($1.1 \times 10^8$ parasites). For each IP, 25 μl Protein G Dynabeads (Invitrogen, 10004D) were washed and resuspended in 200 μl 0.02% Tween 20 in DPBS with 1 μg of anti-HA antibody (Abcam, ab9110) or IgG (Sigma, I9131), and then incubated with rotation at 4 °C for 3 h. Washed antibody-bound beads were incubated with diluted nuclear lysates for 4 h with rotation at 4 °C. Lysates and the bead eluate were separated on a 4–15% TGX Stain-Free gel (Bio-Rad) and transferred to a PVDF membrane using a Trans-Blot Turbo Transfer system (Bio-Rad). AP2-P-3HA was detected with anti-HA antibody (Abcam, ab9110; 1:1,000 in 1% milk-PBST), MORC-GFP was detected with anti-GFP (Chromotek, PABG1; 1:1,000 in 1% milk-PBST), and histone H3 was detected with anti-H3 (Abcam, ab1791; 1:2,500 in 1% milk-PBST) antibody, followed by donkey HRP anti-rabbit secondary antibody (Sigma, GENA934; 1:5,000 in 1% milk-PBST). HRP signal was developed with SuperSignal West Pico Plus or West Femto chemiluminescent substrate (Thermo Fisher, 34580 or 34096, respectively) and imaged with a ChemiDoc XRS+ system (Bio-Rad).

## Computational processing and analysis
**Micro-C processing and analysis.** Micro-C paired-end reads (150 bp paired end) were mapped to the *P. falciparum* genome[32] (plasmoDB. org, v.3, release 55) and processed into pairs and multiresolution normalized contact matrix.mcool files using hicstuff (https://github.com/koszullab/hicstuff) with the following non-default options: '−enzyme mnase −mapping iterative −duplicates −binning 100'. For each condition, correlation between replicates was estimated using hicrep[79]. Replicates showed good correlation (Extended Data Fig. 1a,b); their pairs files were subsequently merged using pairtools[80] and multiresolution binned contact matrix files (.mcool files) of merged replicates were regenerated using cooler[81]. When comparing control to AP2-P KD Micro-C results, only control replicates generated with the same genetic background as the AP2-P KD were used, and control contacts were subsampled with pairtools to 65,106 contacts to match AP2-P KD Micro-C data. Number of mapped reads (million) and final contacts (million, after removal of PCR duplicates) for each Micro-C sample after merging replicates can be found in Supplementary Data 8.

Hi-C data were imported and manipulated in R using HiCExperiment[82]. Genomic distance-dependent contact frequency was computed using HiContacts[82]. Observed/expected contact frequency was computed using the 'detrend()' function from HiContacts[82]. Aggregated contact maps were generated using the 'aggregate()' function in HiContacts[82]. All contact maps, including aggregate plots, were generated in R using HiContacts[82].

Insulation score for any given genomic locus of interest (for example, an HP1 domain border) was estimated by dividing the summed frequency of contacts within a 20 kb window centred at the genomic locus of interest that are not spanning this locus, by the total summed frequency of contacts within this 20 kb window. Higher scores indicate higher insulation of the genomic locus of interest.

Boundaries and long-range interactions were annotated in Micro-C data using the automated structural feature caller chromosight[83], using 1 kb resolution for boundaries and 2 kb resolution for long-range interactions and filtering for *q*-values ≤ $10^{-4}$. Identification of statistically significant local interactions requires estimation of the genomic distance-dependent interaction frequency decay, a metric that does not have any meaning when considering interchromosomal interactions. This prevented us from comprehensively annotating interchromosomal interactions. Boundaries were split into 'euchromatin', 'HP1 domain', 'virulence-coding subtelomeres' or 'non-coding subtelomeres' according to their genomic location (Supplementary Data 6 and 9), and their insulation score was computed as described above. For Fig. 6e, long-range interactions were filtered to only retain those anchored at identified euchromatic boundaries (±5 kb, Supplementary Data 7 and 10).

An overlap between a euchromatic protein-coding gene and AP2-P, AP2-I and/or MORC peaks was considered if peaks were ≤1 kb away from that gene. The enrichment in AP2-P, AP2-I and/or MORC peaks over euchromatic protein-coding genes was compared to the rest of protein-coding genes using Fisher's exact test.

**Hi-C processing and analysis.** Two publicly available Hi-C datasets[12,51] were reprocessed using hicstuff with the following options: '−enzyme MboI −mapping iterative −duplicates −binning 100'.

**ChIP-seq processing and analysis.** Sequenced reads (150 bp paired end) were mapped to the *P. falciparum* genome[32] (plasmoDB.org, v.3, release 56) using Bowtie[84]. PCR duplicates were filtered using samtools' fixmate and markdup[85] commands, and only alignments with a mapping quality ≥30 were retained (samtools view -q 30)[85]. The paired-end deduplicated ChIP and input BAM files were used as treatment and control, respectively, for peak calling with the MACS2 subcommands[86]. In brief, for each ChIP experiment, pileup files were first generated using the MACS2 pileup command, and the larger of the two files

(that is, input or control) was downsampled using MACS2 bdgopt. *q*-values and fold enrichment of ChIP/input were then calculated using MACS2 bdgcmp. Final peak calling was performed using MACS2 bdg-callpeak using a *q*-value cut-off of 0.001 (-c 3). For the 2 biological replicates of AP2-P and MORC, consensus peaks shared between biological replicates 1 and 2 were defined using the bedtools intersect command[87]. ChIP/input ratio tracks were generated using deeptool's bamCompare command[88]. Integrative Genomics Viewer[89] was used to inspect tracks and MACS2 peaks.

Binding peaks were associated with the nearest protein-coding genes using bedtools closest command[87] along with *P. falciparum* reference genome feature file (gff) (plasmoDB.org, v.3, release 56). Only regions 500 bp upstream or downstream near the protein-coding genes were considered further for downstream analysis. To perform functional analysis on the genes closest to the peaks, Gene Ontology Enrichment tool from PlasmoDB web interface (plasmoDB.org, v.3, release 56) was used for Ontology Term – Biological Process, with a *P* value cut-off of 0.05. Fold change quantification and statistical analysis for all peaks and peaks in centromeric regions was performed in R[90].

Non-subtelomeric HP1 'domains' were annotated as follows: the HP1 ChIP-seq track (IP/input)[91] was used to identify all genomic loci with a 5-fold enrichment of IP/Input. Neighbouring loci were merged when closer than 6 kb to each other. Because HP1 is strongly enriched at telomeres, HP1 loci overlapping with non-coding or virulence gene-coding subtelomeric regions were filtered out. This resulted in 64 HP1 domains, of which 14 overlapped with virulence genes (mean domain breadth of 28 kb) and the other 50 did not overlap with virulence genes (mean domain breadth of 8 kb). A table of HP1 domains and their location is provided in Supplementary Data 14.

Metagene plots in Figs. 2c and 4b, and Extended Data Fig. 6d were generated using the plotAvgProf2 command in the ChIPSeeker package[92]. ChIP/input ratios were calculated genome-wide using 'bamCompare' and normalized across regions of interest using 'computeMatrix' in the deeptools package[88]. Final matrices were plotted using 'plotProfile'. The tidyCoverage package was also used to generate coverage heat maps and aggregate coverage tracks in R[93].

De novo motif discovery analysis was performed using MEME-ChIP suite[94]. ChIP-seq peak summits identified using MACS2 were extended ±100 bp and used as the input. These extended summits were then converted to fasta format using the bedtools getfasta command[87]. The resulting file was used as an input for the MEME-ChIP motif search algorithm.

Pie charts were made in R[90]. A Venn diagram was made with DeepVenn[95].

**RNA-seq processing and analysis.** Sequenced reads were mapped to the *P. falciparum* genome (plasmoDB.org, v.3, release 56) using 'STAR'[96], restricting the number of multiple alignments allowed for a read using the option '−outFilterMultimapNmax 1'. Alignments were subsequently filtered for duplicates and a mapping quality ≥20 using samtools[85]. Gene counts were quantified with htseq-count[97] and differentially expressed genes were identified in R using the package DESeq2 (ref. [98]). Gene Ontology enrichment analysis was performed on differentially expressed genes (*q* < 0.05) using the built-in tool at PlasmoDB.org[99] (v.3, release 56) with default settings for Biological Process (*P* < 0.05).

RNA-seq-based cell cycle progression was estimated in R[90] by comparing the normalized expression values (that is, RPKM, reads per kilobase per exon per one million mapped reads) of each sample to the microarray data from ref. [10] using the statistical model in ref. [100].

Barplots and dotplots were made using the ggplot package in R[90]. Histograms were made using data from ref. [47].

### Reporting summary

Further information on research design is available in the Nature Portfolio Reporting Summary linked to this article.

## Data availability

All datasets (ChIP-seq, RNA-seq, Micro-C) generated in this study are available in NCBI with BioProject accession PRJNA1146886. Previously published datasets utilized in this study are available at NCBI with the following accession numbers: AP2-I ChIP from ref. [46]: SRR5114665; AP2-I ChIP Input from ref. [46]: SRR5114667; TRZ ChIP from ref. [43]: SRR3085676; TRZ ChIP Input from ref. [43]: SRR3085677; HP1 ChIP from ref. [91]: SRR12281320; HP1 ChIP Input from ref. [91]: SRR12281322; ATAC-seq from ref. [101]: SRR6055333; ATAC-seq from ref. [101]: SRR6055330; ATAC-seq gDNA control from ref. [101]: SRR6055335; Hi-C from ref. [12]: SRR957166; Hi-C from ref. [51]: SRR19611536; AP2-P ChIP from ref. [51]: SRR17171688; AP2-P ChIP Input from ref. [51]: SRR17171686. Source data are provided with this paper.

## Code availability

All code used in this study is published, publicly available and referenced throughout the paper.

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

## Acknowledgements

This work was supported by the Agence Nationale de la Recherche (grants ANR-21-CE15-0002-02 ApiMORCing and ANR-21-CE15-0010-01 PlasmoVarOrg to J.M.B. and ANR-23-CHBS-0002 to R.K.); the European Research Council (grant PlasmoEpiRNA 947819 to S.B. and 771813 to R.K.); Emerging Infectious Diseases junior seed grant from the Institut Pasteur to J.C. and C.R. P.S. was supported by the Institut

Pasteur Roux-Cantarini postdoctoral fellowship. J.S. was supported by a postdoctoral fellowship from l'Association pour la Recherche sur le Cancer (ARC). We acknowledge the use of the Biomics and Flow Cytometry platforms at the Institut Pasteur and the invaluable support of PlasmoDB and the Laboratoire d'Excellence (LabEx) ParaFrap (ANR-11-LABX-0024). We thank R. Bartfai for kindly lending some Plasmodipur filters.

## Author contributions

J.M.B. and S.B. conceptualized the project. J.C., J.M.B. and S.B. designed the methodology (Micro-C adaptation to *P. falciparum*). P.S., J.S., S.B. and J.M.B. conducted formal analysis. J.M.B., P.S., J.C., M.D.C., C.R. and P.C. peformed experiments. J.M.B., P.S. and J.S. prepared data visualizations. J.M.B. wrote the original draft of the paper. All authors reviewed and edited the paper. J.M.B., S.B. and R.K. acquired funding. J.M.B., S.B., R.K. and A.S. supervised the project.

## Competing interests

The authors declare no competing interests.

## Additional information

**Extended data** is available for this paper at https://doi.org/10.1038/s41564-025-02038-z.

**Correspondence and requests for materials** should be addressed to Jessica M. Bryant.

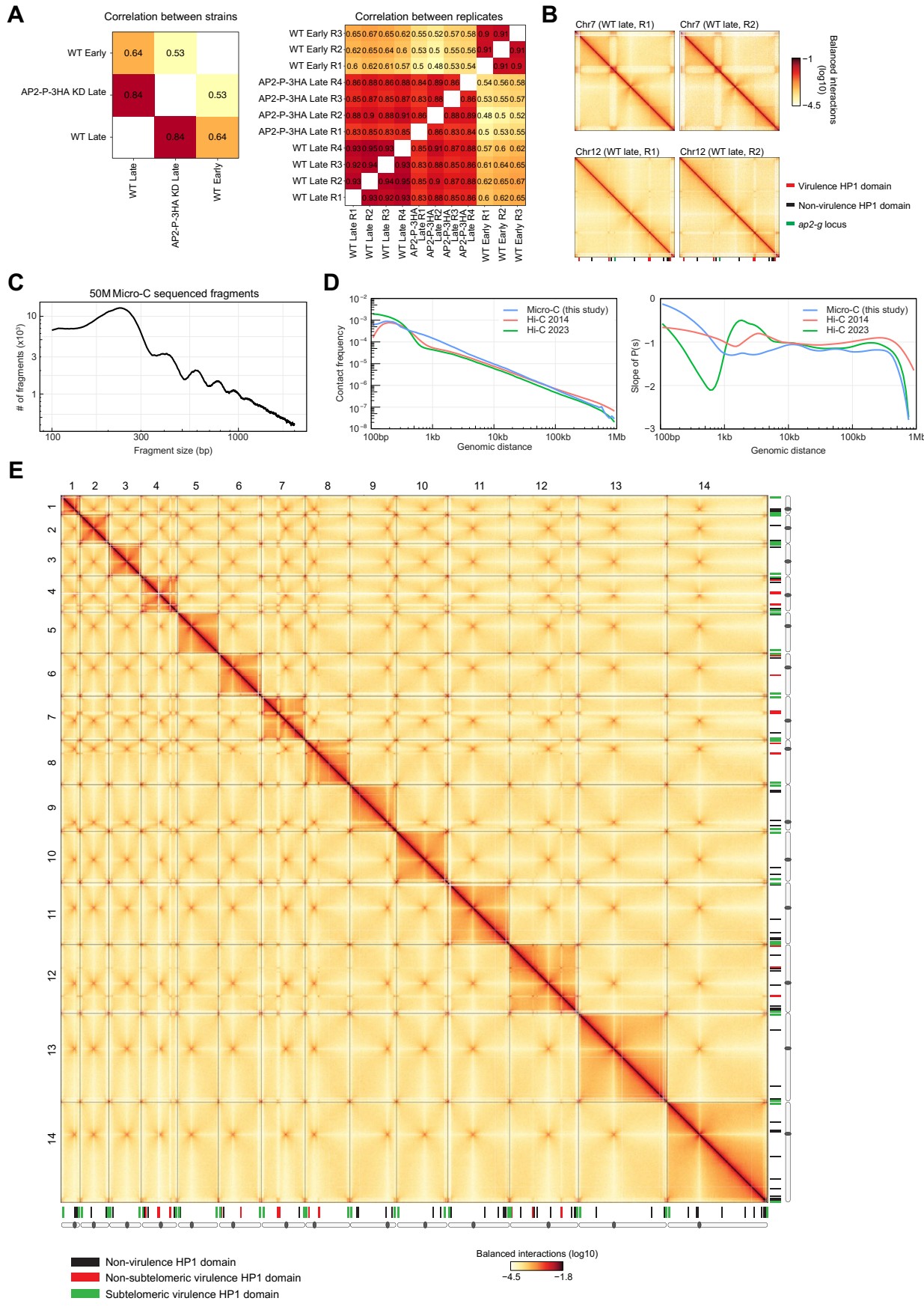

**Extended Data Fig. 1 | See next page for caption.**

**Extended Data Fig. 1 | Quality control of the Micro-C technique in
*P. falciparum*. a**) Chromosome-wide, stratum-corrected (10kb-500kb)
correlation scores between WT and AP2-P-3HA-*glmS* strains from early and late
stages (left) and between each replicate for each strain from each stage (right).
**b**) Micro-C contact maps for chromosomes 7 and 12 comparing two replicates of
WT late-stage parasites. For chromosome 12, HP1 domains that do not contain
virulence genes are indicated with black boxes, HP1 domains that contain
virulence genes with red boxes, and *ap2-g* with a green box. **c**) Fragment size
distribution in the Micro-C library. Only 50 x 10⁶ fragments have been used.

**d**) Comparison of genomic distance-dependent interaction frequencies in
Micro-C data (this study) and previously published Hi-C datasets[12,51] from
*P. falciparum*. Left: P(s); right: slope of P(s). **e**) Micro-C contact map across all
*P. falciparum* chromosomes. Chromosomes are depicted as light grey rectangles
on the axes, and centromeres are represented with dark grey ovals. HP1 domains
that do not contain virulence genes are indicated with black boxes, subtelomeric
HP1 domains that contain virulence genes with green boxes, and non-
subtelomeric HP1 domains that contain virulence genes with red boxes.

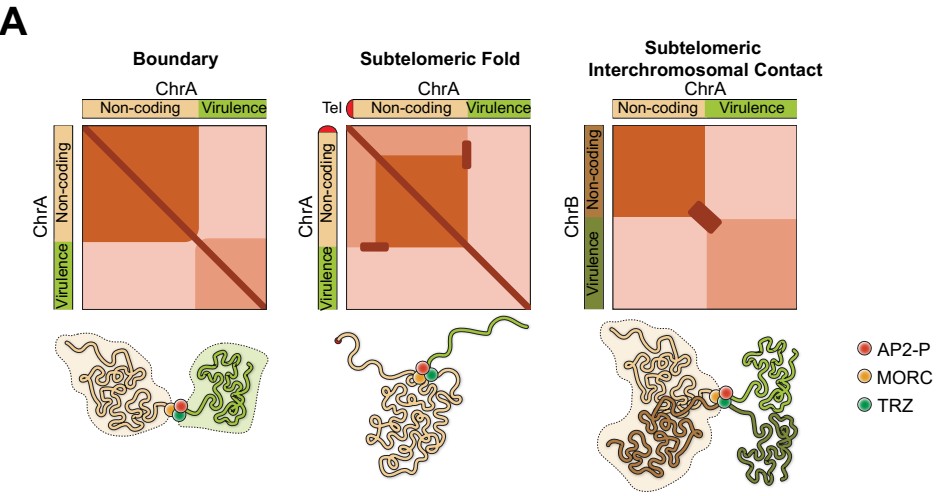

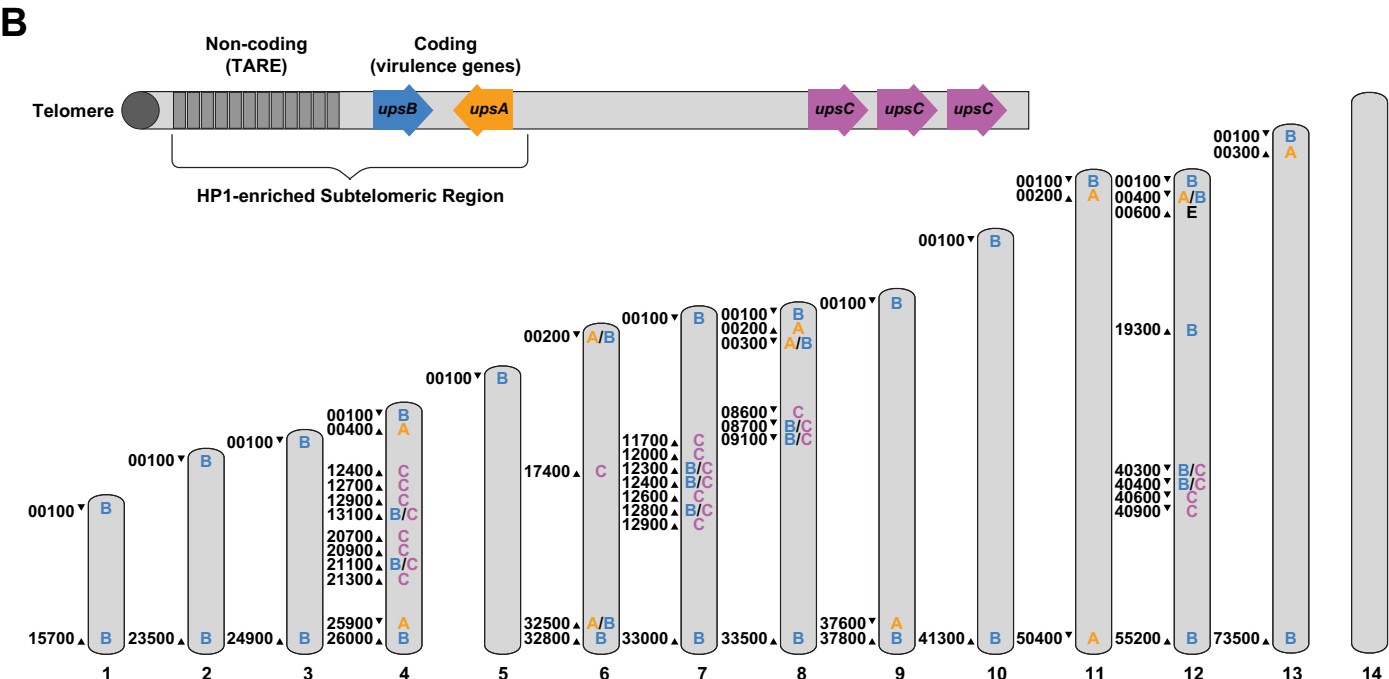

**Extended Data Fig. 2 | Schematics of different Micro-C structures and *var* gene categories and location in the genome. a)** Graphical representation of the different structures observed with Micro-C in subtelomeric regions. Top: Theoretical contact maps, with darker colors indicating higher frequency of contacts. Red is the left arm telomere, tan is the non-coding subtelomeric region, and green is the virulence gene-encoding subtelomeric region. Bottom: Illustrations of hypothetical chromosomal conformations corresponding to the contact maps above. Red balls are AP2-P, yellow balls are MORC, and green balls are TRZ. A boundary separates two neighboring self-interacting domains on the same chromosome, here between the non-coding and coding subtelomeric regions. A subtelomeric fold is an insulated self-interacting domain that folds upon itself, with the two boundaries of the domain forming the base of the fold.

A subtelomeric interchromosomal contact forms between the regions on two different chromosomes where there are boundaries between the non-coding and coding subtelomeric self-interacting domains. **b)** Schematic of how *var* genes are categorized and located within a chromosome (top) and throughout the genome (bottom). Subtelomeric *var* genes are indicated in blue (*upsB*) and orange (*upsA*), and central *var* genes are indicated in purple (*upsC*). Chromosomes are represented with gray bars, and chromosome number is indicated under each bar. *var* gene type is indicated on the chromosome, and gene ID (excluding the preceding chromosome number) is listed to the left of its position on the chromosome. The direction of *var* gene transcription is indicated with an arrowhead.

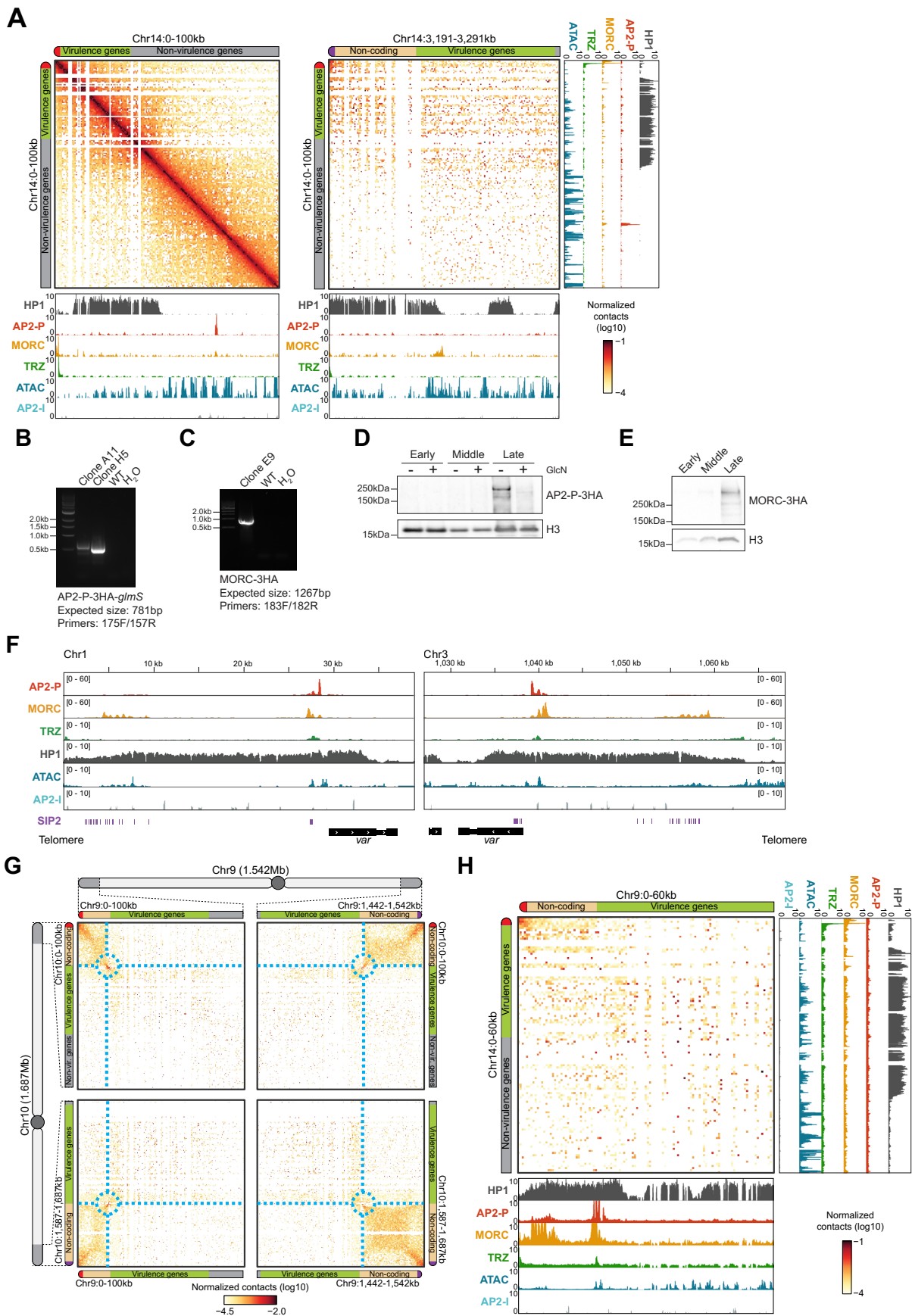

**Extended Data Fig. 3 | See next page for caption.**

**Extended Data Fig. 3 | Micro-C reveals subtelomeric fold structures defined by a multiprotein complex.** (**a**) Micro-C contact maps of the left end of chromosome 14 (left) or between the telomere of each arm of chromosome 14 (right) (100kb-wide, 1kb resolution) and corresponding AP2-P, MORC, HP1[91], AP2-I[46], and TRZ[43] ChIP/input and ATAC-seq[101] tracks from late-stage parasites. Presented otherwise as in (Fig. 2a). (**b**) and (**c**) DNA gels showing PCR validation of the AP2-P-3HA-*glmS* (**b**) and MORC-3HA (**c**) strains with the indicated primers (Supplementary Data 12). No genomic DNA (H₂O) and genomic DNA from WT parasites (WT) were used as controls. DNA size is indicated with a ladder at the left side of each gel and expected band sizes are indicated at the bottom of each gel. These experiments were repeated two independent times with similar results. (**d**) Western blot analysis of nuclear extracts from AP2-P-3HA-*glmS* parasites at early, middle, and late stages with or without glucosamine (GlcN) for 48h. Antibodies against HA and histone H3 were used. Molecular weights at left. Representative result of two independent experiments. (**e**) Western blot analysis of nuclear extracts from MORC-3HA parasites at early, middle, and late stages.

Antibodies against HA and histone H3 were used. Molecular weights at left. Representative result of two independent experiments. (**f**) ChIP-seq data (ChIP/Input) of AP2-P, MORC, AP2-I[46], TRZ[43], and HP1[91] and ATAC-seq data[101] from late-stage parasites at the subtelomeric region on the left arm of chromosome 1 and the right arm of chromosome 3. Putative SIP2 binding sites (SPE2 sequence)[42] are indicated with vertical purple lines. The *x*-axis is DNA sequence, with the *upsB var* gene represented by a black box with white arrowheads to indicate transcription direction. The telomere is also indicated. (**g**) Interchromosomal Micro-C contact map between subtelomeric regions of either arm of chromosome 10 and either arm of chromosome 9 in late-stage parasites (100kb-wide, 1kb resolution). Presented otherwise as in (Fig. 2a). (**h**) Interchromosomal Micro-C contact map between subtelomeric region of the left arm of chromosome 9 and the left arm of chromosome 14 (60kb-wide, 1kb resolution) and corresponding AP2-P, MORC, HP1[91], AP2-I[46], and TRZ[43] ChIP/input and ATAC-seq[101] tracks from late-stage parasites. Presented otherwise as in (Fig. 2a).

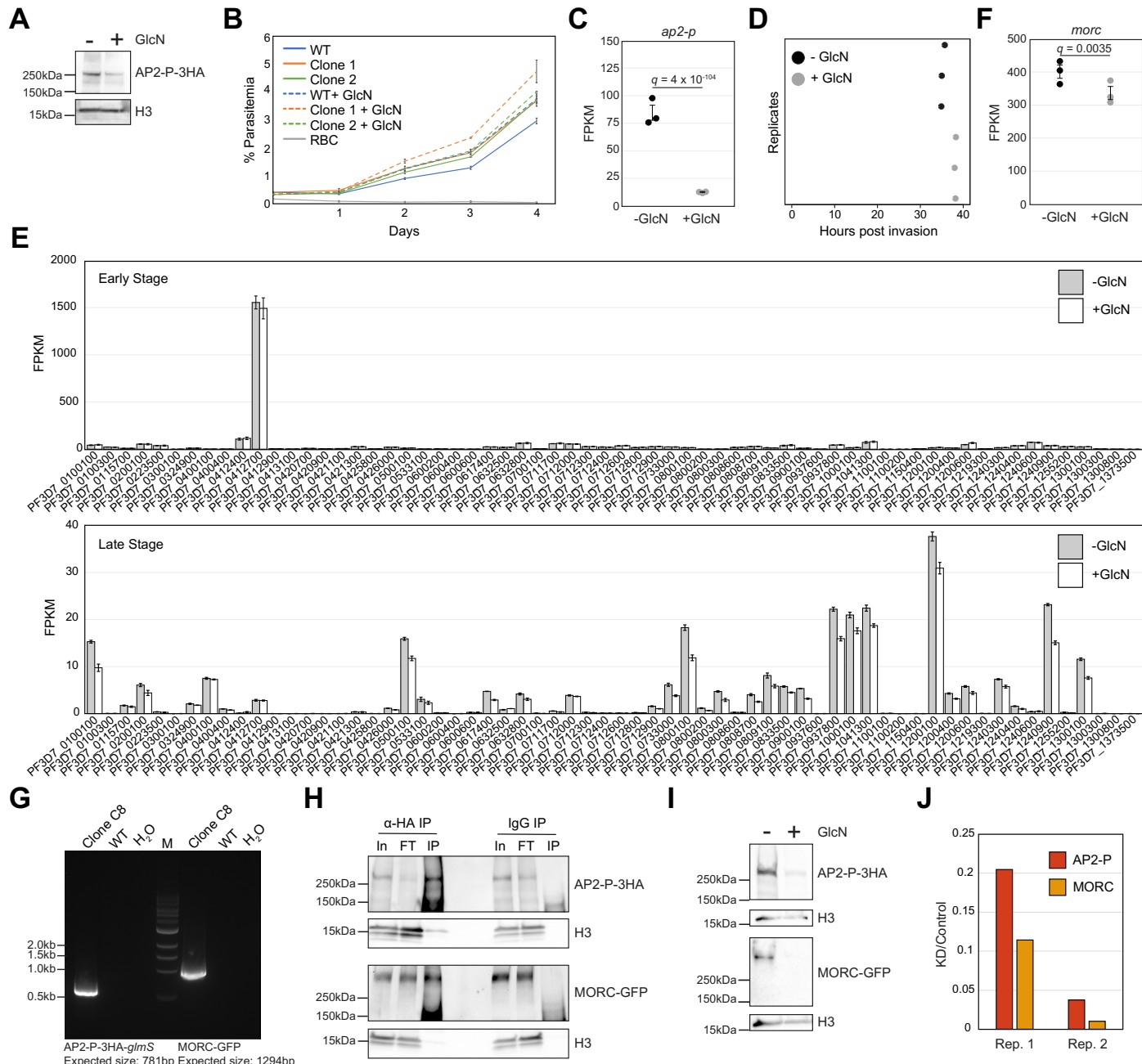

**Extended Data Fig. 4 | AP2-P interacts with MORC, and AP2-P KD leads to MORC KD, but no phenotype in growth, or *var* gene transcription. (a)** Western blot of nuclear extracts from AP2-P-3HA-*glmS* late-stage parasites with or without glucosamine (GlcN) for 48 hr. Antibodies against HA and histone H3 were used. Molecular weights at left. Representative result of two independent experiments. **(b)** Growth curve showing parasitemia (Supplementary Data 13) for WT and two clones of AP2-P-3HA-*glmS* parasites with or without glucosamine (GlcN). Uninfected red blood cells (RBC) are a control. Error bars: standard deviation of three technical replicates (with mean as center). A two-way ANOVA with Tukey post hoc test was used for statistical analysis. No significant differences were found. Gating strategies in Extended Data Fig. 10. **(c)** *ap2-p* transcript levels in AP2-P-3HA-*glmS* late-stage parasites without or with glucosamine (GlcN). Error bars: standard deviation of three technical replicates (with mean as center). P-values calculated with Wald test (multiple comparisons) for significance of coefficients in a negative binomial generalized linear model (DESeq2[98]). *q* (Benjamini-Hochberg adjusted P-value) = 4 x 10$^{-104}$. **(d)** Cell cycle progression estimation of synchronous AP2-P-3HA-*glmS* individual replicates with or without

glucosamine (GlcN). **(e)** *var* gene transcript levels in clonal early- and late-stage AP2-P-3HA-*glmS* parasites with or without glucosamine (GlcN). Error bars: standard error of three technical replicates (with mean as center), calculated with ggplot2 (geom_errorbar). q values were calculated as in **(c)**. **(f)** As in **(c)**, but for *morc* transcript levels. *q* = 0.0035. **(g)** DNA gel showing PCR validation of the AP2-P-3HA-*glmS*:MORC-GFP strain with the indicated primers (Supplementary Data 12). WT or no genomic DNA (H$_2$O) served as controls. DNA size ladder and expected band sizes are shown. This experiment was repeated two independent times with similar results. **(h)** Western blot of nuclear extracts from late-stage AP2-P-3HA-*glms*:MORC-GFP parasites before (input, "In") and after (flow-through, "FT") immunoprecipitation with an antibody against HA (left) or IgG (right). Antibodies against HA, GFP, and histone H3 were used. Molecular weights at left. **(i)** Western blot as in Fig. 3a, but with a different clone and 192 h of glucosamine treatment. **(j)** Quantification of AP2-P and MORC proteins in Fig. 3a and Extended Data Fig. 4i. Ratios for AP2-P or MORC band intensities were normalized to ratios for H3.

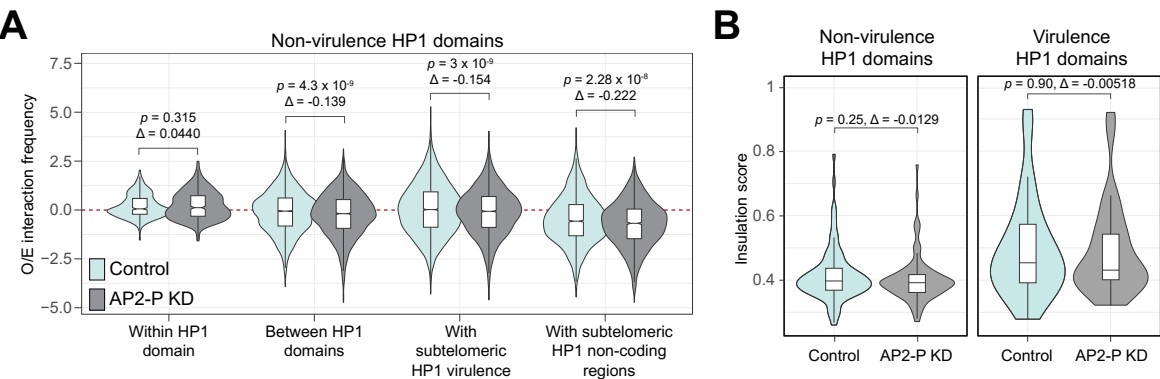

**Extended Data Fig. 5 | AP2-P KD does not affect interactions amongst non-virulence HP1 domains or insulation of HP1 domains.** (a) Log$_2$(observed/expected interaction frequency) ratio for (i) interactions within non-subtelomeric non-virulence gene-containing HP1 domains (n = 939, $p$ = 0.315), (ii) interactions between pairs of non-subtelomeric non-virulence gene-containing HP1 domains (n = 9,138, $p$ = 4.3 x 10$^{-9}$), (iii) interactions between a non-subtelomeric non-virulence gene-containing HP1 domain and virulence gene-containing subtelomeric HP1 domain (n = 10,588, $p$ = 3.01 x 10$^{-9}$), or (iv) interactions between a non-subtelomeric non-virulence gene-containing HP1 domain and non-coding subtelomeric loci (n = 3,854, $p$ = 2.28 x 10$^{-8}$).

Contacts in the subsampled control dataset are shown in cyan and those in the AP2-P KD dataset are shown in grey. Boxes represent the median and IQR, whiskers represent ± 1.5*IQR. P-values from two-sided t-tests are indicated. (**b**) Insulation score at boundaries of non-subtelomeric HP1 domains containing virulence genes (n = 26, $p$ = 0.90) or not (n = 100, $p$ = 0.25) in late-stage parasites (Supplementary Data 6, 14). Contacts in the subsampled control dataset are shown in cyan and those in the AP2-P KD dataset are shown in grey. Boxes represent the median and IQR, whiskers represent ± 1.5*IQR. P-values from two-sided t-tests are indicated.

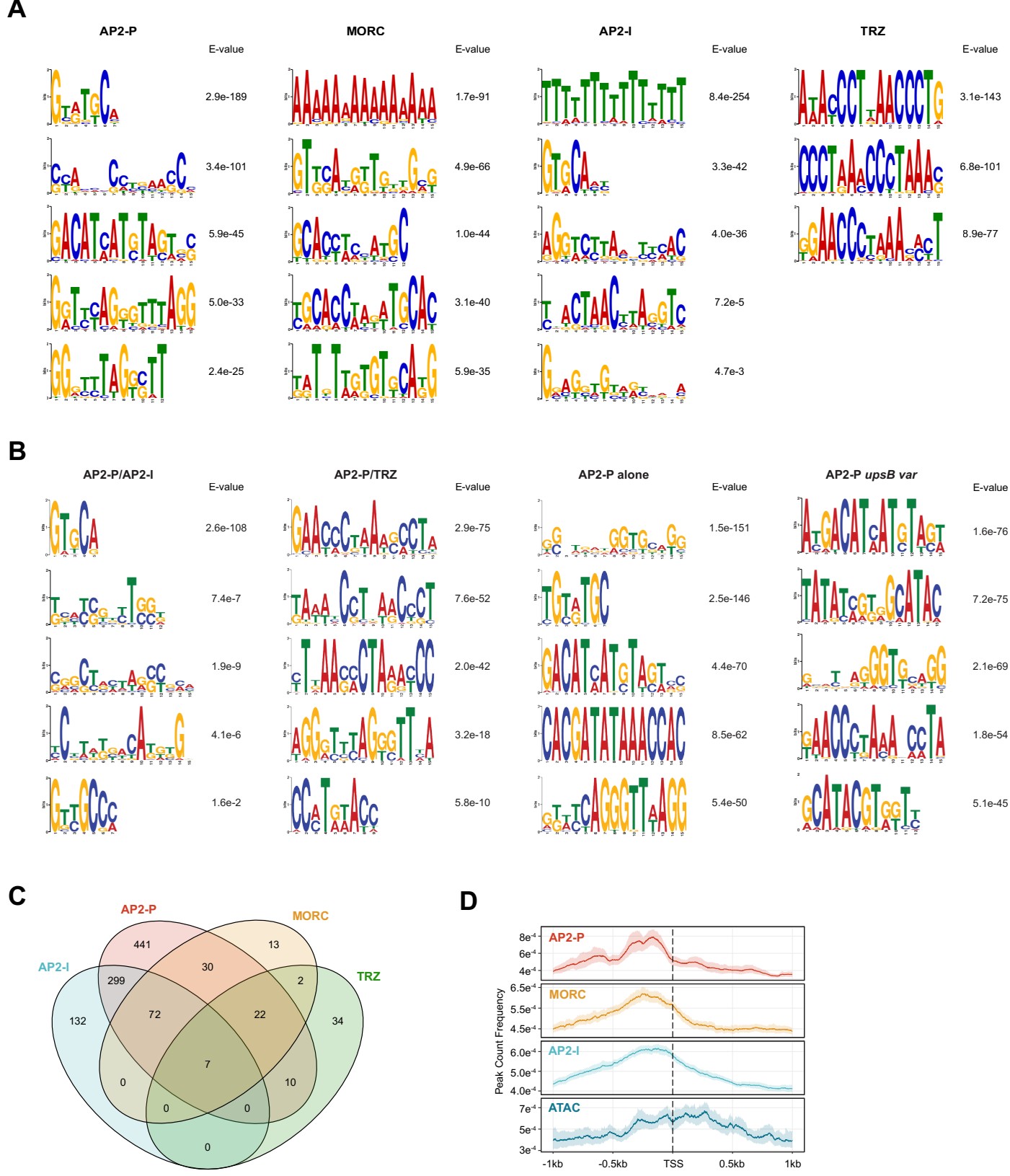

**Extended Data Fig. 6 | See next page for caption.**

**Extended Data Fig. 6 | AP2-P, MORC, AP2-I, and TRZ genome-wide binding patterns.** (**a**) Motif discovery analysis of AP2-P, MORC, AP2-I[46], and TRZ[43] ChIP-seq peaks in late-stage parasites (Supplementary Data 1). Top hits with minimum threshold E-value < 0.05 are shown. (**b**) Motif discovery analysis of regions where a ChIP-seq peak of AP2-P overlaps with a peak of AP2-I[46] or TRZ[43], where AP2-P peaks do not overlap with any other assessed factor, and AP2-P binds upstream of *upsB var* genes in late-stage parasites. Top hits with minimum threshold E-value < 0.05 are shown. **c**) Venn diagram showing overlap of AP2-P, MORC, AP2-I[46], and

TRZ[43] ChIP-seq peaks in late-stage parasites (Supplementary Data 1). **d**) Metagene plots showing average AP2-P and MORC enrichment (y-axis = peak count frequency) in clonal AP2-P-3HA-*glmS* and MORC-3HA parasites, respectively, in late-stage parasites from 1 kb upstream of the TSS to 1 kb downstream of all genes that are significantly downregulated more than two-fold upon AP2-P KD. AP2-I ChIP-seq data[46] and ATAC-seq data[101] from stage-matched parasites are included. One replicate was used for the AP2-P and MORC ChIP datasets.

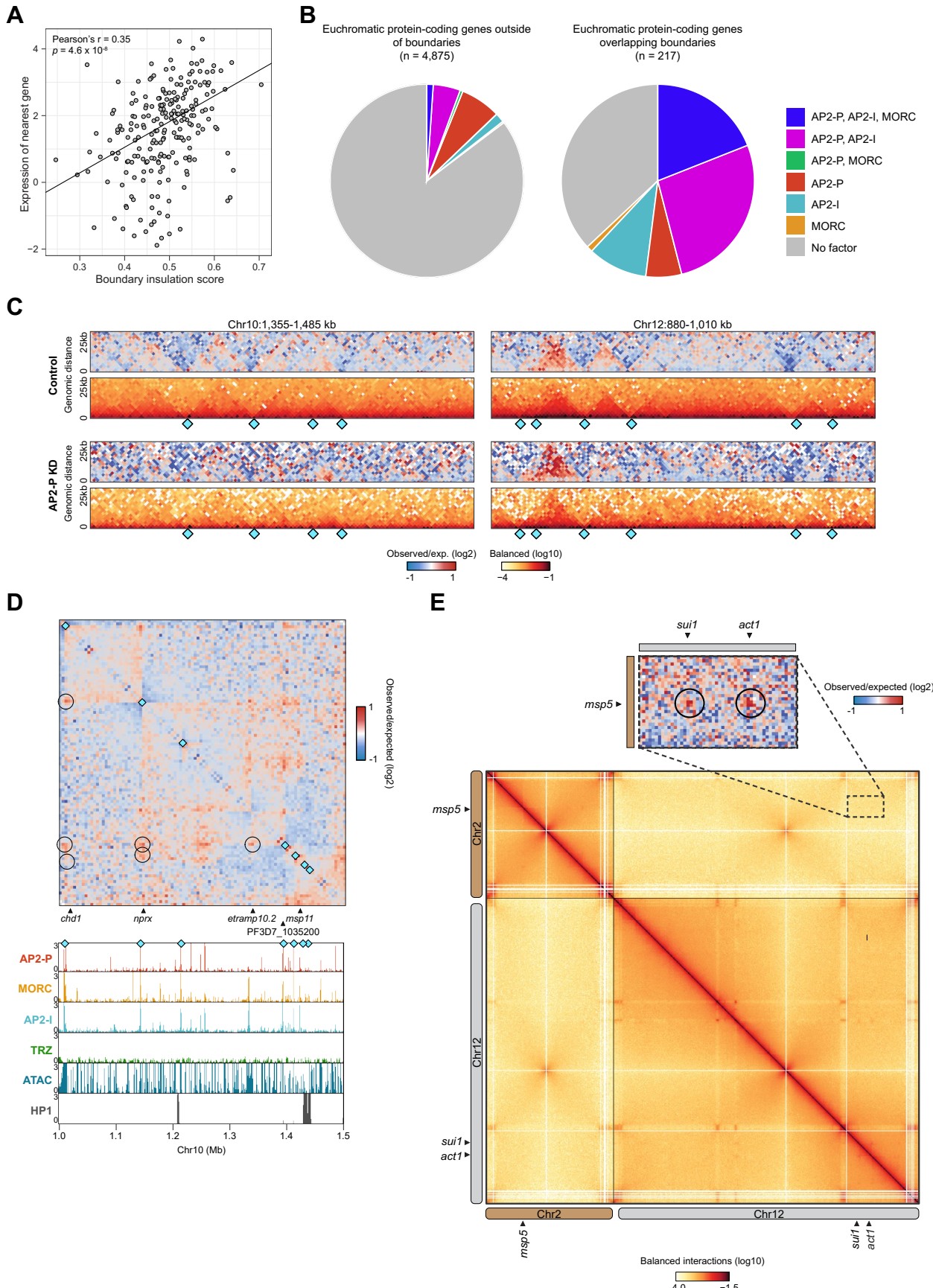

**Extended Data Fig. 7 | See next page for caption.**

**Extended Data Fig. 7 | AP2-P activating complex defines euchromatic boundaries and long-range interactions.** (**a**) Two-sided Pearson correlation between insulation score of 211 euchromatic boundaries (Supplementary Data 6) and the level of expression of the nearest gene in late-stage parasites. (**b**) Pie charts showing the proportion of euchromatic protein-coding genes outside (left) and overlapping (right) Micro-C boundaries that are associated with a peak of AP2-P, MORC, and/or AP2-I (peak ≤ 1kb away from gene). (**c**) Micro-C contact maps of central regions of chromosome 10 (left) and 12 (right) in control (top) and AP2-P KD (bottom) late-stage parasites (130kb-wide loci, 1kb resolution), as presented in (Fig. 5a). Micro-C boundaries are indicated with cyan diamonds. Colour scales indicate $\log_2$-scaled observed/expected interaction frequency ratios (top) and normalized interaction frequency (bottom). (**d**) Micro-C contact map of a central region of chromosome 10 in late-stage parasites. Micro-C boundaries are indicated with cyan diamonds. Selected genes are indicated at the bottom. AP2-P and MORC ChIP/Input ratio signals from late-stage parasites are shown at the bottom. Stage-matched AP2-I[46], TRZ[43], HP1[91] ChIP-seq and ATAC-seq[101] data are included. Long-range interactions identified with chromosight[83] are indicated with black circles on the contact map. (**e**) Micro-C contact maps showing intra- and interchromosomal contacts (normalized interaction frequency) in chromosomes 2 and 12. Interchromosomal interactions between the *msp5* locus (Chr2) and *sui1* or *act1* loci (Chr12) are highlighted in the inset showing observed/expected contact frequency ratio.

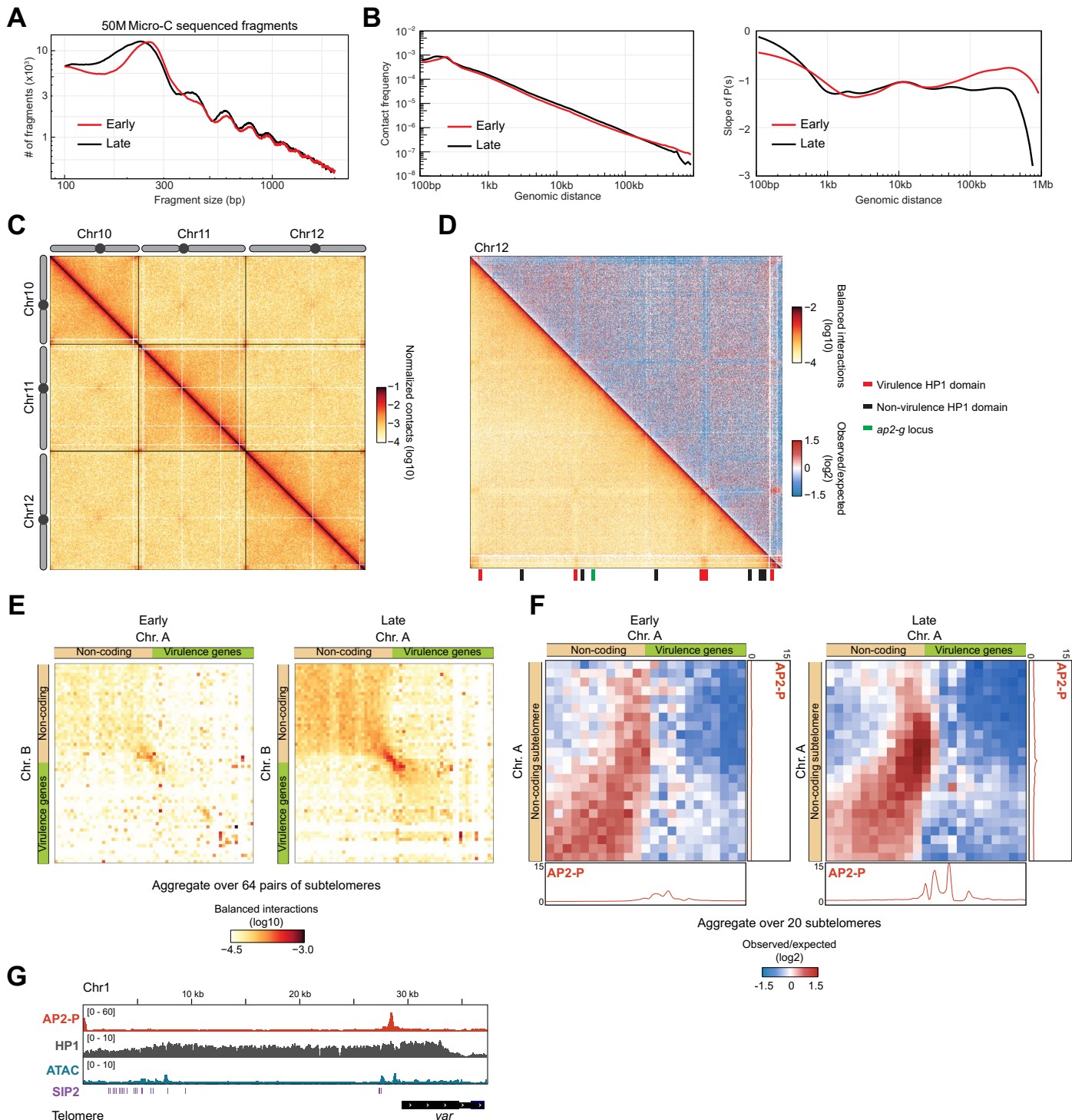

**Extended Data Fig. 8 | See next page for caption.**

**Extended Data Fig. 8 | Heterochromatic Micro-C interactions are maintained in early-stage parasites. a**) Fragment size distribution in Micro-C libraries from early- and late-stage parasites. Only 50 x 10$^6$ fragments have been used. **b**) Comparison of genomic distance-dependent interaction frequencies in Micro-C data from early- and late-stage parasites. Left: P(s); right: slope of P(s). **c**) Micro-C contact maps in early-stage parasites over chromosomes 10 to 12 (5kb resolution). Chromosomes are indicated with grey rectangles and centromeres with dark grey circles. **d**) Micro-C contact maps (bottom left: normalized interaction frequency, top right: log$_2$(observed/expected interaction frequency)) over chromosome 12 (2kb resolution) in early-stage parasites. HP1-enriched regions encompassing virulence genes (red), non-virulence genes (black), and *ap2*-g (green) are indicated at the bottom. **e**) Interchromosomal Micro-C contact maps for early- (left) and late-stage (right) parasites aggregated over the 64 interchromosomal pairs of subtelomeric loci, characterized by the presence of a non-coding subtelomeric region (±15kb, 500bp resolution). Contact maps extracted from subtelomeric loci located at the end of chromosomes have been mirrored so that the telomere is always positioned towards the top left

corner, as in Fig. 3e. **f**) Off-diagonal Micro-C contact maps (log$_2$-scaled observed/expected interaction frequency ratio) aggregated over 20 subtelomeric loci (Supplementary Data 15) in early- (left) and late-stage (right) parasites, characterized by the presence of a non-coding subtelomeric region (±5kb, 500bp resolution) as in (Fig. 2b). This map highlights the fold-like structure detected by Micro-C. Contact maps extracted from subtelomeric loci located at the end of chromosomes have been mirrored so that the telomere is always positioned towards the top left corner. 1D aggregated AP2-P ChIP/input signals from early[51]- and late-stage parasites are shown below and on the right of the 2D aggregated Micro-C map. **g**) ChIP-seq data showing enrichment of AP2-P[51] in early-stage parasites and HP1[91] in late-stage parasites at the subtelomeric region on the left arm of chromosome 1. ATAC-seq data[101] from a closely corresponding stage (15 hpi) shows chromatin accessibility. Putative SIP2 binding sites (SPE2 sequence)[42] are indicated with vertical purple lines. The *x*-axis is DNA sequence, with the *upsB var* gene represented by a black box with white arrowheads to indicate transcription direction. The telomere is also indicated.

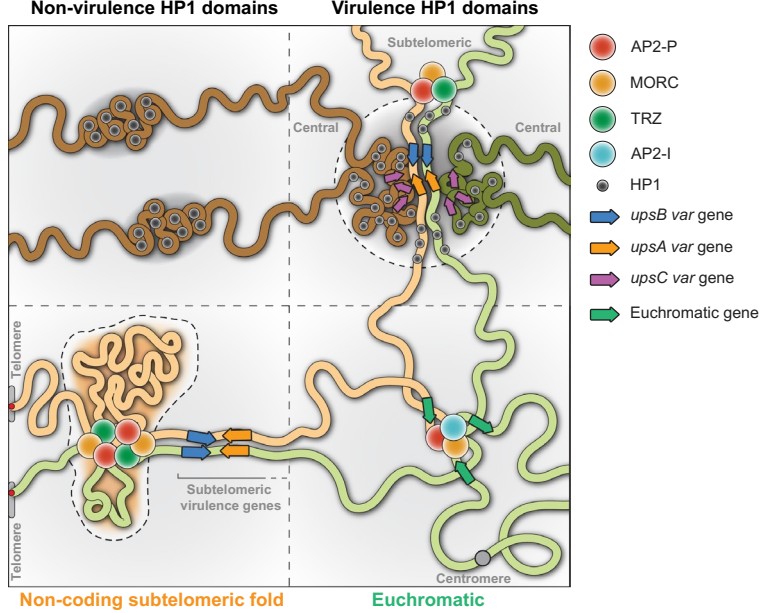

**Extended Data Fig. 9 | Schematic of heterochromatic and euchromatic structures observed with Micro-C.** In this study, we identified four different types of chromatin structures. HP1 heterochromatin domains that encompass virulence genes, but not those that encompass non-virulence genes, form well-insulated self-associating domains that interact with other virulence-gene-containing HP1 heterochromatin domains. Heterochromatic non-coding subtelomeric sequences upstream of *upsB var* genes are bound by AP2-P, MORC, and TRZ and form fold-like structures that facilitate trans interactions with other subtelomeric *var* genes. Euchromatic, actively transcribed loci bound by AP2-P, MORC, and AP2-I form long-range interactions in a stage-specific manner.

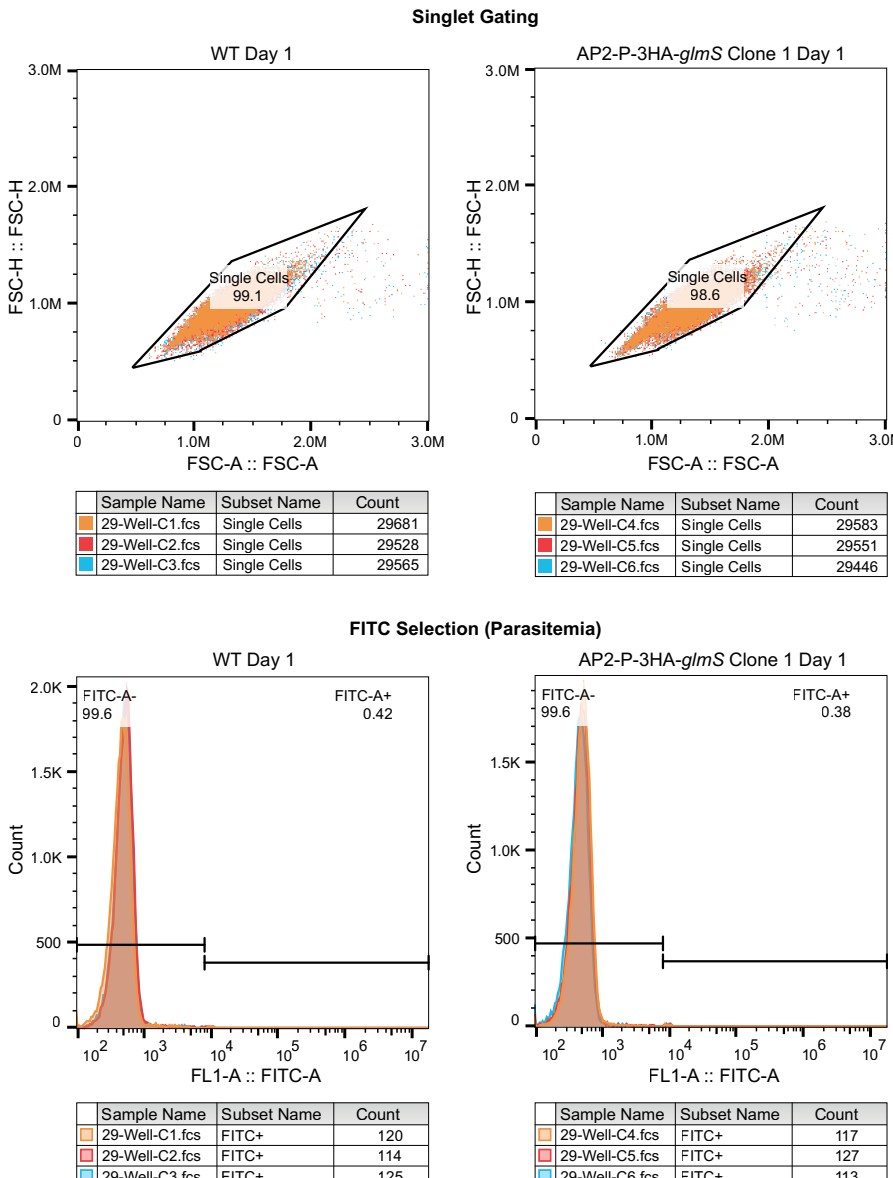

**Extended Data Fig. 10 | Gating strategy used to analyze flow cytometry data.** Flow cytometry gating strategy for determining parasitemia shown in Extended Data Fig. 4b and Supplementary Data 13. Single red blood cells were selected by forward scatter height and area (FSC-H and FSC-A, respectively). This population was then selected for FITC positivity to determine parasitemia (percentage parasite-infected red blood cells). Data are for WT and AP2-P-3HA-*glmS* Clone 1 on Day 1 of the growth curve. Orange, red, and blue represent triplicates.

# Reporting Summary

## Statistics

For all statistical analyses, confirm that the following items are present in the figure legend, table legend, main text, or Methods section.

| n/a | Confirmed | |
|---|---|---|
| ☐ | ☒ | The exact sample size (*n*) for each experimental group/condition, given as a discrete number and unit of measurement |
| ☐ | ☒ | A statement on whether measurements were taken from distinct samples or whether the same sample was measured repeatedly |
| ☐ | ☒ | The statistical test(s) used AND whether they are one- or two-sided<br>*Only common tests should be described solely by name; describe more complex techniques in the Methods section.* |
| ☒ | ☐ | A description of all covariates tested |
| ☒ | ☐ | A description of any assumptions or corrections, such as tests of normality and adjustment for multiple comparisons |
| ☐ | ☒ | A full description of the statistical parameters including central tendency (e.g. means) or other basic estimates (e.g. regression coefficient) AND variation (e.g. standard deviation) or associated estimates of uncertainty (e.g. confidence intervals) |
| ☒ | ☐ | For null hypothesis testing, the test statistic (e.g. *F*, *t*, *r*) with confidence intervals, effect sizes, degrees of freedom and *P* value noted<br>*Give P values as exact values whenever suitable.* |
| ☒ | ☐ | For Bayesian analysis, information on the choice of priors and Markov chain Monte Carlo settings |
| ☒ | ☐ | For hierarchical and complex designs, identification of the appropriate level for tests and full reporting of outcomes |
| ☐ | ☒ | Estimates of effect sizes (e.g. Cohen's *d*, Pearson's *r*), indicating how they were calculated |

*Our web collection on statistics for biologists contains articles on many of the points above.*

## Software and code

Policy information about availability of computer code

| Data collection | BioRad Image Lab software 5.2 |
|---|---|
| Data analysis | cooler v0.9.3<br>hicstuff v3.2.4<br>hicrep 1.12.2 (R version)<br>pairtools v1.1.0<br>HiCExperiment 1.4.0<br>HiContacts 1.6.0<br>chromosight v1.6.3<br>bowtie2 v2.5.1<br>samtools v1.19.2<br>macs2 2.2.7.1<br>bedtools v2.31.0<br>ChIPseeker 1.40.0<br>deeptools 3.5.3<br>tidyCoverage 1.0.0<br>MEME 5.5.6<br>ggplot<br>DeepVenn (web interface)<br>STAR 2.7.11b<br>htseq-count 2.0.6 |

DESeq2 1.44.0
BioRad Image Lab software 5.2
FlowJo 10.9.0

All python packages were installed in python 3.10.12 with micromamba 1.5.1.  All R packages were installed in R 4.4.0 with  Bioconductor 3.19.

For manuscripts utilizing custom algorithms or software that are central to the research but not yet described in published literature, software must be made available to editors and reviewers. We strongly encourage code deposition in a community repository (e.g. GitHub). See the Nature Portfolio guidelines for submitting code & software for further information.

## Data

Policy information about availability of data

All manuscripts must include a data availability statement. This statement should provide the following information, where applicable:
- Accession codes, unique identifiers, or web links for publicly available datasets
- A description of any restrictions on data availability
- For clinical datasets or third party data, please ensure that the statement adheres to our policy

All data sets (ChIP-seq, RNA-seq, Micro-C) generated in this study are available in NCBI BioProject accession #PRJNA1146886.
Previously published data sets utilized in this study are available at the following NCBI accession numbers:
• AP2-I ChIP from (Santos et al., 2017)46: SRR5114665
• AP2-I ChIP Input from (Santos et al., 2017)46: SRR5114667
• TRZ ChIP from (Bertschi et al., 2017)43: SRR3085676
• TRZ ChIP Input from (Bertschi et al., 2017)43: SRR3085677
• HP1 ChIP from (Carrington et al., 2021)90: SRR12281320
• HP1 ChIP Input from (Carrington et al., 2021)90: SRR12281322
• ATAC-seq from (Toenhake et al., 2018)100: SRR6055333
• ATAC-seq from (Toenhake et al., 2018)100: SRR6055330
• ATAC-seq gDNA control from (Toenhake et al., 2018)100: SRR6055335
• Hi-C from (Ay et al., 2014)12: SRR957166
• Hi-C from (Subudhi et al., 2023)51: SRR19611536
• AP2-P ChIP from (Subudhi et al., 2023)51: SRR17171688
• AP2-P ChIP Input from (Subudhi et al., 2023)51: SRR17171686

## Research involving human participants, their data, or biological material

Policy information about studies with human participants or human data. See also policy information about sex, gender (identity/presentation), and sexual orientation and race, ethnicity and racism.

| Reporting on sex and gender | NA |
|---|---|
| Reporting on race, ethnicity, or other socially relevant groupings | NA |
| Population characteristics | NA |
| Recruitment | NA |
| Ethics oversight | NA |

Note that full information on the approval of the study protocol must also be provided in the manuscript.

# Field-specific reporting

Please select the one below that is the best fit for your research. If you are not sure, read the appropriate sections before making your selection.

☒ Life sciences  ☐ Behavioural & social sciences  ☐ Ecological, evolutionary & environmental sciences

For a reference copy of the document with all sections, see nature.com/documents/nr-reporting-summary-flat.pdf

# Life sciences study design

All studies must disclose on these points even when the disclosure is negative.

| Sample size | Three  replicates were used for the early-stage Micro-C and four replicates for the late-stage Micro-C experiments. RNA-seq was performed in triplicate. Two biological replicates were performed for AP2-P and MORC ChIP-seq. Input material was used in quantities recommended by manufacturers or protocols. No sample size calculations were performed. Sample size was determined by what is recommended by the ENCODE standards. |
|---|---|

| Data exclusions | No data were excluded from the analysis. |
|---|---|
| Replication | Reproducibility of replicated Micro-C experiments was assessed by hicrep, by visual inspection and by computing Pearson correlation between replicates.<br>All western blots and DNA gels were performed/replicated at least twice independently, with the exception of ED Fig. 4H.<br>RNA-seq was replicated once.<br>ChIP-seq experiments were replicated twice independently (two biological replicates each).<br>For each Micro-C condition/stage, four technical replicates were performed. The late-stage Micro-C experiment was performed twice independently. |
| Randomization | Our study did not require randomization. Covariates are not relevant to this study. |
| Blinding | Our study did not require blinding. It did not involve animal or human cohorts or subjective counting. |

# Reporting for specific materials, systems and methods

We require information from authors about some types of materials, experimental systems and methods used in many studies. Here, indicate whether each material, system or method listed is relevant to your study. If you are not sure if a list item applies to your research, read the appropriate section before selecting a response.

## Materials & experimental systems

| n/a | Involved in the study |
|---|---|
| ☐ | ☒ Antibodies |
| ☐ | ☒ Eukaryotic cell lines |
| ☒ | ☐ Palaeontology and archaeology |
| ☒ | ☐ Animals and other organisms |
| ☒ | ☐ Clinical data |
| ☒ | ☐ Dual use research of concern |
| ☒ | ☐ Plants |

## Methods

| n/a | Involved in the study |
|---|---|
| ☐ | ☒ ChIP-seq |
| ☐ | ☒ Flow cytometry |
| ☒ | ☐ MRI-based neuroimaging |

## Antibodies

| Antibodies used | For western blot analysis, HA-tagged proteins, GFP-tagged proteins, and histone H3 were detected with anti-HA (Abcam ab9110, 1:1,000 in 1% milk-PBST) or anti-HA-HRP (Cell Signaling C29F4 HRP Conjugate #14031), anti-GFP (Chromotek PABG1) and anti-H3 (Abcam ab1791, 1:2,500 in 1% milk-PBST) primary antibodies, respectively, followed by donkey anti-rabbit secondary antibody conjugated to horseradish peroxidase ("HRP", Sigma GENA934, 1:5,000 in 1% milk-PBST). For ChIP-seq, HA-tagged proteins were immunoprecipitated with anti-HA (Abcam ab9110) antibody. |
|---|---|
| Validation | The anti-HA (Abcam ab9110, datasheet: https://www.abcam.com/en-us/products/primary-antibodies/ha-tag-antibody-chip-grade-ab9110?srsltid=AfmBOookpyx0G2pXgn6p2yEg5tCLpYY5Xa39t4DpGwrR0gUFLDoy1jdU#tab=datasheet) and anti-H3 (Abcam ab1791, datasheet: https://www.abcam.com/en-us/products/primary-antibodies/histone-h3-antibody-nuclear-marker-and-chip-grade-ab1791#tab=datasheet) are ChIP-grade antibodies that have been guaranteed by Abcam for ChIP and western blot. We have validated anti-HA and anti-H3 antibodies in previous studies (https://doi.org/10.15252/msb.20209569 and https://doi.org/10.15252/embr.202357090) for western blot analysis and ChIP-seq. The anti-GFP antibody (Chromotek PABG1, datasheet: https://www.ptglab.com/products/GFP-antibody-rabbit-polyclonal-PABG1.htm?srsltid=AfmBOoqVxarICqyRnVQMeS_sZpEvAZy112J4LetSdHbTfMXZXv0A5wuF#product-information) has been guaranteed by Chromotek for western blot analysis. We have validated the anti-GFP antibody for western blot analysis in a separate, unpublished story and would be happy to supply the data upon request. |

## Eukaryotic cell lines

Policy information about cell lines and Sex and Gender in Research

| Cell line source(s) | The wildtype cell line used for Micro-C is a clone of the NF54 reference strain. The parent cell line of all cell lines generated for this study is a bulk culture of 3D7 reference strain. |
|---|---|
| Authentication | We used PCR and Sanger sequencing to confirm correct integration of all epitope tag and glmS ribozyme sequences at our endogenous genes of interest. |
| Mycoplasma contamination | All strains have tested negative for mycoplasma. |
| Commonly misidentified lines<br>(See ICLAC register) | No commonly misidentified lines were used in this study. |

# Plants

| Seed stocks | NA |
|---|---|
| Novel plant genotypes | NA |
| Authentication | NA |

# ChIP-seq

## Data deposition

☒ Confirm that both raw and final processed data have been deposited in a public database such as GEO.

☒ Confirm that you have deposited or provided access to graph files (e.g. BED files) for the called peaks.

**Data access links**
*May remain private before publication.*

All NextGen sequencing-based data sets (ChIP-seq, RNA-seq, Micro-C) generated in this study are available in NCBI BioProject accession #PRJNA1146886.

**Files in database submission**

We provide raw reads for ChIP-seq, RNA-seq and Micro-C experiments. We also provide processed files for Micro-C (.mcool), and ChIP-seq (.bigwig and .narrowPeak).

**Genome browser session**
(e.g. UCSC)

NA

## Methodology

**Replicates**

AP2-P and MORC ChIP-seq was performed in two biological replicates, with significant overlap between peaks identified by macs2.

**Sequencing depth**

AP2-P ChIP-Seq Replicate 1: 10191901 total reads, 10187115 unique mapped reads were recovered (paired-end, fragment size ~ 172); AP2-P ChIP-Seq Replicate 2: 6709230 total reads, 6701798 unique mapped reads were recovered (paired-end, fragment size ~ 152); MORC ChIP-seq Replicate 1: 6601103 total reads, 5624668 unique mapped reads were recovered (paired-end, fragment size ~ 196); MORC ChIP-seq Replicate 2: 38829706 total reads, 28029942 unique mapped reads were recovered (paired-end, fragment size ~ 218)

**Antibodies**

HA-tagged proteins (AP2-P and MORC) were immunoprecipitated with anti-HA (Abcam ab9110) ChIP grade antibody.

**Peak calling parameters**

Sequenced reads (150 bp paired end) were mapped to the P. falciparum genome (plasmoDB.org, version 3, release 56) using bowtie. PCR duplicates were filtered using samtools' fixmate and markdup commands and only alignments with a mapping quality ≥ 30 were retained (samtools view -q 30). The paired end deduplicated ChIP and input BAM files were used as treatment and control, respectively, for peak calling with the MACS2 subcommands. For each ChIP experiment, pileup files were first generated using the MACS2 pileup command, and the larger of the two files (i.e., Input or control) was down-sampled using MACS2 bdgopt. q-values and fold enrichment of ChIP/input was then calculated using MACS2 bdgcmp. Final peak calling was performed using MACS2 bdgcallpeak using a q-value cut-off of 0.001 (-c 3). For the two biological replicates of AP2-P and MORC, consensus peaks shared between biological replicates 1 and 2 were defined using the bedtools intersect command.

**Data quality**

All peaks used for analysis had a q-value < 0.001 and a fold enrichment > 2. 452 out of 946 AP2-P consensus peaks had a fold enrichment > 5. The 149 consensus MORC peaks had a fold enrichment > 5.

**Software**

bowtie2 v2.5.1
samtools v1.19.2
macs2 2.2.7.1
bedtools v2.31.0
ChIPseeker 1.40.0
deeptools 3.5.3
tidyCoverage 1.0.0
MEME 5.5.6
DeepVenn (web interface)

All python packages were installed in python 3.10.12 with micromamba 1.5.1. All R packages were installed in R 4.4.0 with Bioconductor 3.19.

# Flow Cytometry

## Plots

Confirm that:

☒ The axis labels state the marker and fluorochrome used (e.g. CD4-FITC).

☒ The axis scales are clearly visible. Include numbers along axes only for bottom left plot of group (a 'group' is an analysis of identical markers).

☒ All plots are contour plots with outliers or pseudocolor plots.

☒ A numerical value for number of cells or percentage (with statistics) is provided.

## Methodology

| | |
|---|---|
| Sample preparation | Flow cytometry was used to determine parasitemia during a growth curve analysis. Two AP2-P-3HA-glms clones and a WT clone were tightly synchronized. Each culture was split, and glucosamine (Sigma G1514, 2.5 mM final concentration) was added to one half for approximately 96 h before starting the growth curve. The parasites were tightly re-synchronized and diluted to ~0.2% parasitemia (5% hematocrit) at ring stage. The growth curve was performed in a 96-well plate (200 μL culture per well) with three technical replicates per condition.. Every 24 h, 5 μL of the culture were fixed in 45 μL of 0.025% glutaraldehyde in PBS for 1 h at 4°C. After centrifuging at 800 g for 5 min, free aldehyde groups were quenched by re-suspending the iRBC pellet in 200 μL of 15 mM NH4Cl in PBS. A 1:10 dilution of the quenched iRBC suspension was incubated with Sybr Green I (Sigma S9430) to stain the parasite nuclei. |
| Instrument | CytoFLEX S cytometer (Beckman Coulter) |
| Software | FlowJo 10.9.0 |
| Cell population abundance | Red blood cells (RBC) range ~ 96%-99%<br>Singlets range ~ 96 %-99% of total RBCs identified population<br>Infected RBCs (FITC +) Range: 0.2%-4.5% of total singlets population<br>Non-infected RBCs (FITC-) range: 95%-99.8% of total singlets population |
| Gating strategy | Red blood cells (RBCs) were identified in a FSC and SSC comparison. Singlets were gated using FSC (H) vs. FSC (A) of previously identifed RBCs. P. falciparum infected RBC (iRBC) were selected inside the singlet (single) population based on FITC positive signal. |

☒ Tick this box to confirm that a figure exemplifying the gating strategy is provided in the Supplementary Information.

