## [Peer Review File · Nature Microbiology]

High resolution map of the *Plasmodium falciparum* genome reveals MORC/Ap2-mediated links between distant, functionally related genes

Corresponding Author: Dr Jessica Bryant

Version 0:

Reviewer comments:

Reviewer #1

(Remarks to the Author)

This well-crafted study describes the application of Micro-C to analyze chromatin structure in *P. falciparum* asexual blood stages. Previous HiC studies of *P. falciparum* confirmed the clustering of telomeres, rDNA, heterochromatin and centromere within the nucleus that had already been described using microscopy. However, due to low sequence resolution, little new information about genome organization was gained from these earlier experiments.

By improving resolution 25-fold the authors of this study were able to identify new types of long range chromatin interactions and implicate AP2-P and MORC in their formation or maintenance.

This approach will offer much greater understanding into the role of nuclear organization in regulating gene expression not just in malaria parasites and protozoan pathogens more, generally.

I have no concerns with respect to the execution or analysis of the experiments in this study.

The main weakpoint of the study however is its focus on only a single time point in asexual blood stages, which limits its utility for understanding the possible roles of long-distance interactions at other time-points during asexual development or gametocytogenesis.

I have included major questions and minor points below.

Björn Kafsack

Major Points/Questions:

#####

1. The image chosen for Figure 3A suggests very efficient knock down of AP2-P and PfMORC. Since References 66-67 found that AP2-P and PfMORC were essential for asexual growth, it will be important to quantify the GlcN induced reduction in AP2-P and MORC at the protein level.
2. Is it possible to demonstrate that AP2-P-HA and MORC are in the same complex by IP?
3. Since the authors suggest two distinct complexes that mediate different long-range interactions (AP2-P/MORC/SIP/TRZ and AP-P/PfMORC/AP2-I) it would be good to include AP2-I peaks on Figure S2C/H/J to show that AP2-I peaks are absent from the subtelomeric chromatin fold. Figure 5 already shows that TRZ is absent from AP2-I containing peaks. Is the presence of TRZ and AP2-I anti-correlated in AP2-P/MORC peak regions?
4. The text states on page 10 that AP2-P ChIP peaks were enriched for GTGCA but Fig S4A doesn't reflect this. This motif also differs substantially from the AP2-P binding motif identified by protein binding microarray (Campbell et al. PLOS Path 2010) but matches the PBM motif for AP2-I, so AP2-P is probably targeted there by AP2-I. Is that motif present in the virulence HP1-fold anchor site, where AP2-I is presumably not bound? What motif enrichment do you see for AP2-P peaks without nearby AP2-I peaks? Or AP2-P peaks in the subtelomeric fold anchor versus non-subtelomeric peaks.
5. The long-range euchromatic interactions: Rather than just showing the aggregate change in long-range euchromatic

interactions in response to AP2-P knockdown,
please also provide the plots for individual boundaries highlighted in 5A in response to the AP2-P knockdown.
Since the strength of this study is the power of Micro-C, it wouldn't be problematic if boundaries still form but are weaker.

6. The authors mention that the AP2-G heterochromatin island interacts with the sub-telomeric virulence HP1 loci.
Does the ap2-g locus have AP2-P/MORC/TRZ peaks? Does the heterochromatin island downstream of *gdv1* on chromosome 9 form similar interactions as ap2-g?

#####

Minor points (in no particular order):

Since the readers of Nature Micro are less likely to be familiar with the nomenclature of chromatin/nuclear organization it might be worth expanding on the concepts more.

This could be done expanding the legend description of Fig S2B and explicitly directing the reader there in the main text.

Similarly, for the long-range interactions in euchromatin with AP2-P/MORC, the authors might consider using "boundary region" instead of "boundary" as the latter implies a point rather than a region containing one or more genes.

Fig 1D-F needs a color scale for log₂ observed/expected ratio,

also, while they are equivalent, intuitively expected/observed might make more sense to the reader since high values would then indicate greater than expected interactions & vice versa

Later in the manuscript observed/expected is indeed used.

Fig S1C: Mark centromeres and virulent & non-virulent HP1 domains on the edges of the interaction plots

page 7 1st para & elsewhere: non-genic is not an accurate term for TARE containing regions since they produce transcripts and are thus genes.

Fig 2C: put the count frequencies in the bottom panel on the same scale as the top panel.

Figure 2 and elsewhere: make all dotted lines and circles highlighting interactions grey instead of different colors.

Page 9 first sentence: prior to the knockdown results, the presented data is entirely correlative. I'd suggest weakening the statement to "may facilitate".

Figure 6 is nice but could be a sub-panel of a figure if more space is needed.

Text says PfMORC was epitope tagged instead of "fused to GFP", which is not strictly an epitope tag,

The 20% decrease in transcript levels are unlikely to explain the almost complete loss of *morc* at the protein levels, and warrants further discussion.

Similarly, it might be worth adding discussion of the finding in ref 66 in light of the finding that MORC expression depends on AP2-P.

Page 9 paragraph 1 last sentence: the data justify replacing "suggest" with something stronger.

Change "virulent HP1 domains" to "virulence HP1 domains"

Figure S3B: Using color for the strain and solid/dashed for +/- glucosamine would make this figure easier to parse.

As noted above the lack of growth phenotype is worth discussing in light of the knockdown efficiency and essential nature of AP2-P and PfMORC in refs 66-67.

Figure 3D and similar comparisons: is it possible to generate a single heatmap that shows the change in interactions between control & knockdown, rather than showing them side-by-side?

Fig 3F: The figure specifies SUBTELOMERIC Virulence HP1 domains but the text doesn't make the distinction between subtelomeric & non-subtelomeric

Can you also show change in interactions of non-subtelomeric HP virulence clusters (centromeric var clusters).

Fig. 4: please specify the hours-post-invasion range each experiment of "late-stage parasites" in the text and figure legends as the meaning of late-stage varies substantially between researchers from 24-48 hpi to 40-48 hpi.

Fig. 4F: Unsure as to the point of this figure since RNA-seq was only performed at one time-point in "late-stage" parasites.

What percentage of down-regulated genes have upstream AP2-P peak indicating they are direct targets? what percentage has *morc* peaks?

Page 10: "while var genes are silent" should be "while var gene promoters are silent" since transcription of the intron is active in late stages

Page 11: I'd suggest removing "functionally related" from the section title as they are only related at the highest levels of the GO hierarchy.

And at the timepoint chosen for this study, a substantial fraction of the expressed genes contribute to structures involved in egress/invasion since merozoites are being formed.

Similarly, the claim that AP2-P/MORC activate genes in a spatio-TEMPORAL manner isn't well supported since interactions were only explored at a single timepoint.

5A: What is the y-axis on the interaction count heatmaps?
5D: what is "chromosight" ?

Page 12: the fact that the interaction spans the centromere isn't particularly striking since they form in 3D rather than 2D space and earlier figures already showed interactions of subtelomeric regions on the same chromosome.
Page 12 last sentence: remove "trans" since different chromosomes was already specified.

Reviewer #2

(Remarks to the Author)

In this paper by Singh et al, the authors have established the MicroC technique for identifying inter- and intrachromosomal interactions in *P. falciparum* at near-nucleosomal resolution. The paper shows that this approach generates an improved resolution relative to previously published HiC chromosome conformation capture studies. This is a critical prerequisite to understand how tight gene control is achieved in the condensed, gene-rich Plasmodium genome, and how this depends on spatial chromatin architecture. By correlating MicroC and ChIPseq data of two regulatory trans-factors, AP2-P and MORC, the study proposes that distinct protein complexes involving AP2-P and MORC in combination with further trans-factors create fold structures that organize the genome into functionally distinct compartments.

Of note, a comprehensive study of the AP2-P factor was published by Subudhi et al. in *Nature Microbiology* 2023 (<https://doi.org/10.1038/s41564-023-01497-6>), which utilized a conditional KO system of exon 2 to investigate the function of AP2-P across the *P. falciparum* life cycle. There are several discrepancies between the two studies, which may partially be explain by the different conditional depletion systems used (GlmS mediated conditional KD here) and should be clearly discussed, for example a lack of an effect on virulence gene expression in the present study, as well as the absence of a growth phenotype.

The in-depth correlation of AP2-P relative to MORC and other published transcription factor data sets with chromatin contact points determined by MicroC provides, however, novelty, and elegantly links previous and new datasets into a biological concept. Unfortunately, this study is restricted to a single parasite stage during the asexual blood phase and thus provides only limited mechanistic insight. It would be highly relevant to see the dynamics of these contacts and boundaries (at least) during the asexual life cycle to better understand how these chromatin architectural features influence gene expression. For example, the AP2-P and MORC enriched euchromatic contact points are insulating genes which are highly transcribed in the schizont stage from neighbouring genes, but it remains to be shown whether these boundaries disappear in the ring stage, or are replaced by other factors. Likewise, it remains unclear whether the subtelomeric genic/non-genic boundaries play a role in suppressing upsB var genes, and whether this is stage dependent.

In summary, in my opinion the paper does not meet the standard for publication in *Nature Microbiology* in its current form. It would significantly benefit from more replicates of MORC ChIP; ChIP of MORC in AP2-KD parasites; statistical evidence for loss of boundaries in AP2-KD parasites; time resolution by inclusion of additional life stages for ChIP and MicroC (please find further suggestions below).

Major:

For MORC, only a single ChIP replicate has been included at a single stage in the life cycle. To verify the validity of the data, at least a second biological replicate is necessary (in agreement with ENCODE standards).

Figure 1G: the definition of "non-virulent HP1 domains" versus "virulent HP1 domains" is unclear to me, and the number of non-virulent domains displayed in the figure seems rather high considering that most of the heterochromatin compartment covers clusters of var, rif, stevor, and other exported protein families considered virulence factors. I am not aware of that many large HP1 domains comprising "non-virulent genes", only a few single genes. Please define this further and provide information on the size and location of these domains, as well as on the "insulation scores" for each of these regions, for example in a supplementary table. Is the same phenomenon evident in the 3D7 AP2-P WT line, and how does it change after AP2-P KD?

Suppl. Figure 2C, J: In the 3D7v55 reference genome, the Chr 14 sequence is truncated to 1500 bp (as opposed to 20 Kb in other chromosomes) and has only a 1 kb non-genic sequence, therefore the graphical depiction of the chromosomal regions in this figure is wrong. Consequently, the authors should show a different chromosome to support the link between var gene biology and presence of the "fold like structure" between genic and non-genic regions (page 7) or remove this conclusion and figure all together. To my knowledge and in agreement with Fig S2A, there is no "complete" chromosome end in this reference genome that lacks a telomere proximal upsB var gene. Chromosomes 6 and 11, which have a telomeric upsA var gene annotated, are also largely missing non-genic telomeric sequences. Likewise, the absence of MORC and AP2-P seems to be related to this chromosomal end deletion. I think this needs to be clarified. (Of note: AP2-P data from Subudhi et al. 2023 data actually do show a peak upstream of the var pseudogene of the left Chr 14 end).

Figure 2C: AP2-P and MORC enrichment upstream of subtelomeric var gene – are any of these sites consistent with the SPE2/SIP2 binding sites (Flueck et al. 2010)? From Suppl. Figure 2H it appears like SIP2 binding sites are enriched under the dominant MORC peak only. Of note, in the chromosome 3 depiction the SIP2 binding sites are shifted into the var gene. Please check whether this is correct, as the SPE2 motifs actually lie further upstream. It would be interesting to do motif searches for these particular regions enriched by AP2-P / MORC (similar to Figure S4 but for the subtelomeric sites only).

It looks like the level of enrichment at the three subtelomeric peaks of MORC and AP2-P is contrary, with the dominant MORC peak 1 kb closer to the telomere than the dominant AP2-P peak. Given the distance of the peaks, do you predict that there is a direct interaction between these proteins, and can you provide experimental evidence for this, for example by Co-IP or ChIP-re-ChIP?

Did you find AP2-P and/or MORC enriched in promoters of central var genes, as suggested by Subudhi et al?

How do the observed intra- and interchromosomal contacts in the euchromatin and heterochromatin compartment change over the life cycle, when expression of the genes near the contact points varies? Can similar structures be observed in rings?

Suppl. Fig. 3E: In schizonts, usually var exon 2 transcripts dominate, whereas coding mRNAs are highly expressed in rings. Do the observed transcripts reflect exon 2 transcripts or coding transcripts? Do you see any differences in var expression in the ring stages?

The effect of AP2-P KD on MORC expression (Figure 3A) is interesting and highly relevant. According to Subudhi et al. 2023 AP2-P ChIP data, there is an AP2-P peak at the TSS of MORC (in rings). Is this evident in the data presented here? This may suggest that the loss of boundary integrity observed after AP2-P depletion is a consequence of the downregulation of MORC. How does the genomic distribution of MORC change upon AP2-P KD in the double transgenic parasite line?

I find the KD MicroC graphs in Figure 3 difficult to judge – to me, the KD data appear more noisy in general. Can the data shown in Fig 3B and C be quantified, for example by comparing the normalized contacts at the boundary to normalized contacts of telomeres, centromeres and central var clusters, which are expected not to be affected if the role of AP2-P KD in boundary formation is specific (similar to 3F)? If I understand correctly, in table 6 the insulation scores are reported for subtelomeric and non-subtelomeric heterochromatic genes etc. Does the category “non-genic subtelomeric” represent the identified subtelomeric fold structure contact points? Is there a significant difference in the insulation scores or contacts of these different chromosomal locations between KD and WT?

Fig 5A: What is plotted on the Y-axis here? If I understand correctly, the boundary is a point of decreased interaction with the surrounding genome. So what exactly are the genes “outside the boundaries” (Fig 5C) - genes (how) distant from boundaries? The experiments were conducted in four technical replicates. To get an idea of the biological variability, how well did AP2-P KD (no glucosamine) and NF54 MicroC data correlate regarding (1) the boundaries between telomeric and subtelomeric heterochromatin, (2) contacts between virulence gene clusters, and (3) euchromatic contact points?

In the introduction (page 5), the authors suggest that MicroC is useful to address the question of enhancers-promoter pairing. Enhancers remain poorly defined in *P. falciparum*. Does the MicroC data provide any evidence for the existence of enhancers, for example within genes?

Minor comments:

The MORC mutation is in the literature and protein databases commonly spelled “microrchidia” not “microorchidia”, although I agree that the latter would be biologically more appropriate.

Please report the correlation coefficients for the four NF54 MicroC replicates.

Fig 1 C-F please include a scale for the blue red part of the graphs (log2 scaled observed/expected interaction frequency)

Page 7, last sentence: it is unclear what findings were confirmed by the ChIPseq results of AP2-P and MORC.

Fig 1 G: please define the “non-virulent HP1 domains” further, for example by providing an excel sheet of coordinates and associated genes.

Table 6: correct the word “euchromatinatic”

Please also provide p-values for Suppl. Figure 3H.

Page 10, Motifs: please note that the AP2i “GTGCA” motif is also part of the SPE2 motif and is not only (weakly) evident in the predicted AP2-P binding sites, but also in one of the MORC motifs. As mentioned above, it would make sense to investigate motifs in heterochromatic and euchromatic boundaries separately.

Page 12, Table 7: there are no inter-chromosomal interactions listed in the table, only intra-chromosomal.

Figure 6 – colors are missing

Reviewer #3

(Remarks to the Author)

The manuscript titled “Micro-C reveals MORC/ApiAP2-mediated links between distant, functionally related genes in the human malaria parasite” by Singh et al. offers valuable new insights into the three-dimensional organization of the *Plasmodium falciparum* genome. Through a combination of cutting-edge genomic techniques, including high-resolution Micro-C, ChIP-seq, RNA-seq, and other genomic assays, the authors present a comprehensive and nuanced depiction of the genome’s structural and functional dynamics during specific stages of parasite infection.

The clarity of the presentation, coupled with the outstanding quality of the writing, allows the intricate findings to be easily accessible to the reader. The authors provide compelling evidence, backed by high-resolution Micro-C maps, to support their conclusions on genome folding and chromatin landscape. The data are thoroughly analyzed and convincingly discussed, with the clear and easily understandable figures.

In sum, this manuscript offers significant contributions to our understanding of *P. falciparum* biology, shedding light on novel forms of coordination between chromatin factors (like AP2-P; MORC and HP1), genome architecture and gene expression as well as offering an appealing hypothesis on a mechanism bringing var genes in close proximity for their known recombination events. I have no reservations in recommending this work for publication in *Nature Microbiology*.

However, I would like to suggest a few minor revisions to further improve the manuscript. First, when discussing Hi-C maps and similar 3C-based data, I recommend consistently using the term “contact” instead of “interaction,” as seen in a few instances within the text and figures. Additionally, in the second paragraph of the Results section (page 7), the term “compacts” would be preferable to “condenses” when describing DNA organization.

Lastly, I noted a minor inconsistency in the PCR band sizes on agarose gels, as depicted in Figures S2D, E, and S3G, which may be due to a labeling issue in the DNA ladder lane. It would also be beneficial to include P-values (even if non-significant) in Figure S3H for consistency with similar graphs, such as Figure 3F.

Decision Letter:

5th November 2024

Dear Jessica,

Thank you for your patience while your manuscript "Micro-C reveals MORC/ApiAP2-mediated links between distant, functionally related genes in the human malaria parasite" was under peer-review at Nature Microbiology. It has now been seen by 3 referees, whose expertise and comments you will find at the end of this email. As previously discussed, although they find your work of some potential interest, they have raised a number of concerns that will need to be addressed before we can consider publication of the work in Nature Microbiology. Thank you also for preparing a revision plan. We have now read it and feel it sounds very promising. Editorially, we agree that studying one additional life cycle stage will be sufficient here. Please do perform another replicate Micro-C experiment for the rings stage. Please also address all other referees comments as outlined in your plan.

Should further experimental data allow you to address these criticisms, we would be happy to look at a revised manuscript.

Please include a data availability statement as a separate section after Methods but before references, under the heading "Data Availability". This section should inform readers about the availability of the data used to support the conclusions of your study. This information includes accession codes to public repositories (data banks for protein, DNA or RNA sequences, microarray, proteomics data etc...), references to source data published alongside the paper, unique identifiers such as URLs to data repository entries, or data set DOIs, and any other statement about data availability. At a minimum, you should include the following statement: "The data that support the findings of this study are available from the corresponding author upon request", mentioning any restrictions on availability. If DOIs are provided, we also strongly encourage including these in the Reference list (authors, title, publisher (repository name), identifier, year). For more guidance on how to write this section please see: <http://www.nature.com/authors/policies/data/data-availability-statements-data-citations.pdf>

* If you have not done so already we suggest that you begin to revise your manuscript so that it conforms to our Article format instructions at <http://www.nature.com/nmicrobiol/info/final-submission>. Refer also to any guidelines provided in this letter.

When submitting the revised version of your manuscript, please pay close attention to our [href="https://www.nature.com/nature-portfolio/editorial-policies/image-integrity">Digital Image Integrity Guidelines](https://www.nature.com/nature-portfolio/editorial-policies/image-integrity) and to the following points below:

Link Redacted

Note: This url links to your confidential homepage and associated information about manuscripts you may have submitted or be reviewing for us. If you wish to forward this e-mail to co-authors, please delete this link to your homepage first.

Nature Microbiology is committed to improving transparency in authorship. As part of our efforts in this direction, we are now requesting that all authors identified as 'corresponding author' on published papers create and link their Open Researcher and Contributor Identifier (ORCID) with their account on the Manuscript Tracking System (MTS), prior to acceptance. This applies to primary research papers only. ORCID helps the scientific community achieve unambiguous attribution of all scholarly contributions. You can create and link your ORCID from the home page of the MTS by clicking on 'Modify my Springer Nature account'. For more information please visit [please visit www.springernature.com/orcid](http://www.springernature.com/orcid).

If you wish to submit a suitably revised manuscript we would hope to receive it within 6 months. If you cannot send it within this

time, please let us know.

Yours sincerely,

Reviewer Expertise:

Referee #1: Parasite transcription, chromatin biology

Referee #2: Malaria molecular biology

Referee #3: Genome organization

Reviewer Comments:

Reviewer #1 (Remarks to the Author):

This well-crafted study describes the application of Micro-C to analyze chromatin structure in *P. falciparum* asexual blood stages. Previous HiC studies of *P. falciparum* confirmed the clustering of telomeres, rDNA, heterochromatin and centromere within the nucleus that had already been described using microscopy. However, due to low sequence resolution, little new information about genome organization was gained from these earlier experiments.

By improving resolution 25-fold the authors of this study were able to identify new types of long range chromatin interactions and implicate AP2-P and MORC in their formation or maintenance. This approach will offer much greater understanding into the role of nuclear organization in regulating gene expression not just in malaria parasites and protozoan pathogens more, generally.

I have no concerns with respect to the execution or analysis of the experiments in this study.

The main weakpoint of the study however is its focus on only a single time point in asexual blood stages, which limits its utility for understanding the possible roles of long-distance interactions at other time-points during asexual development or gametocytogenesis.

I have included major questions and minor points below.

Björn Kafsack

Major Points/Questions:

#####

1. The image chosen for Figure 3A suggests very efficient knock down of AP2-P and PfMORC. Since References 66-67 found that AP2-P and PfMORC were essential for asexual growth, it will be important to quantify the GlcN induced reduction in AP2-P and MORC at the protein level.
2. Is it possible to demonstrate that AP2-P-HA and MORC are in the same complex by IP?
3. Since the authors suggest two distinct complexes that mediate different long-range interactions (AP2-P/MORC/SIP/TRZ and AP-P/PfMORC/AP2-I) it would be good to include AP2-I peaks on Figure S2C/H/J to show that AP2-I peaks are absent from the subtelomeric chromatin fold. Figure 5 already shows that TRZ is absent from AP2-I containing peaks. Is the presence of TRZ and AP2-I anti-correlated in AP2-P/MORC peak regions?
4. The text states on page 10 that AP2-P ChIP peaks were enriched for GTGCA but Fig S4A doesn't reflect this. This motif also differs substantially from the AP2-P binding motif identified by protein binding microarray (Campbell et al. PLOS Path 2010) but matches the PBM motif for AP2-I, so AP2-P is probably targeted there by AP2-I. Is that motif present in the virulence HP1-fold anchor site, where AP2-I is presumably not bound? What motif enrichment do you see for AP2-P peaks without nearby AP2-I peaks? Or AP2-P peaks in the subtelomeric fold anchor versus non-subtelomeric peaks.
5. The long-range euchromatic interactions: Rather than just showing the aggregate change in long-range euchromatic interactions in response to AP2-P knockdown, please also provide the plots for individual boundaries highlighted in 5A in response to the AP2-P knockdown. Since the strength of this study is the power of Micro-C, it wouldn't be problematic if boundaries still form but are weaker.
6. The authors mention that the AP2-G heterochromatin island interacts with the sub-telomeric virulence HP1 loci. Does the ap2-g locus have AP2-P/MORC/TRZ peaks? Does the heterochromatin island downstream of gdv1 on chromosome 9 form similar interactions as ap2-g?

#####

Minor points (in no particular order):

Since the readers of Nature Micro are less likely to be familiar with the nomenclature of chromatin/nuclear organization it might be worth expanding on the concepts more.

This could be done expanding the legend description of Fig S2B and explicitly directing the reader there in the main text.

Similarly, for the long-range interactions in euchromatin with AP2-P/MORC, the authors might consider using "boundary region" instead of "boundary" as the latter implies a point rather than a region containing one or more genes.

Fig 1D-F needs a color scale for log₂ observed/expected ratio,

also, while they are equivalent, intuitively expected/observed might make more sense to the reader since high values would then indicate greater than expected interactions & vice versa

Later in the manuscript observed/expected is indeed used.

Fig S1C: Mark centromeres and virulent & non-virulent HP1 domains on the edges of the interaction plots

page 7 1st para & elsewhere: non-genic is not an accurate term for TARE containing regions since they produce transcripts and are thus genes.

Fig 2C: put the count frequencies in the bottom panel on the same scale as the top panel.

Figure 2 and elsewhere: make all dotted lines and circles highlighting interactions grey instead of different colors.

Page 9 first sentence: prior to the knockdown results, the presented data is entirely correlative. I'd suggest weakening the statement to "may facilitate".

Figure 6 is nice but could be a sub-panel of a figure if more space is needed.

Text says PfMORC was epitope tagged instead of "fused to GFP", which is not strictly an epitope tag,

The 20% decrease in transcript levels are unlikely to explain the almost complete loss of morc at the protein levels, and warrants further discussion.

Similarly, it might be worth adding discussion of the finding in ref 66 in light of the finding that MORC expression depends on AP2-P.

Page 9 paragraph 1 last sentence: the data justify replacing "suggest" with something stronger.

Change "virulent HP1 domains" to "virulence HP1 domains"

Figure S3B: Using color for the strain and solid/dashed for +/- glucosamine would make this figure easier to parse.

As noted above the lack of growth phenotype is worth discussing in light of the knockdown efficiency and essential nature of AP2-P and PfMORC in refs 66-67.

Figure 3D and similar comparisons: is it possible to generate a single heatmap that shows the change in interactions between control & knockdown, rather than showing them side-by-side?

Fig 3F: The figure specifies SUBTELOMERIC Virulence HP1 domains but the text doesn't make the distinction between subtelomeric & non-subtelomeric

Can you also show change in interactions of non-subtelomeric HP virulence clusters (centromeric var clusters).

Fig. 4: please specify the hours-post-invasion range each experiment of "late-stage parasites" in the text and figure legends as the meaning of late-stage varies substantially between researchers from 24-48 hpi to 40-48 hpi.

Fig. 4F: Unsure as to the point of this figure since RNA-seq was only performed at one time-point in "late-stage" parasites.

What percentage of down-regulated genes have upstream AP2-P peak indicating they are direct targets? what percentage has morc peaks?

Page 10: "while var genes are silent" should be "while var gene promoters are silent" since transcription of the intron is active in late stages

Page 11: I'd suggest removing "functionally related" from the section title as they are only related at the highest levels of the GO hierarchy.

And at the timepoint chosen for this study, a substantial fraction of the expressed genes contribute to structures involved in egress/invasion since merozoites are being formed.

Similarly, the claim that AP2-P/MORC activate genes in a spatio-TEMPORAL manner isn't well supported since interactions were only explored at a single timepoint.

5A: What is the y-axis on the interaction count heatmaps?

5D: what is "chromosight" ?

Page 12: the fact that the interaction spans the centromere isn't particularly striking since they form in 3D rather than 2D space and earlier figures already showed interactions of subtelomeric regions on the same chromosome.

Page 12 last sentence: remove "trans" since different chromosomes were already specified.

Reviewer #2 (Remarks to the Author):

In this paper by Singh et al, the authors have established the MicroC technique for identifying inter- and intrachromosomal interactions in *P. falciparum* at near-nucleosomal resolution. The paper shows that this approach generates an improved resolution relative to previously published HiC chromosome conformation capture studies. This is a critical prerequisite to understand how tight gene control is achieved in the condensed, gene-rich Plasmodium genome, and how this depends on spatial chromatin architecture. By correlating MicroC and ChIPseq data of two regulatory trans-factors, AP2-P and MORC, the study proposes that distinct protein complexes involving AP2-P and MORC in combination with further trans-factors create fold structures that organize the genome into functionally distinct compartments.

Of note, a comprehensive study of the AP2-P factor was published by Subudhi et al. in *Nature Microbiology* 2023 (<https://doi.org/10.1038/s41564-023-01497-6>), which utilized a conditional KO system of exon 2 to investigate the function of AP2-P across the *P. falciparum* life cycle. There are several discrepancies between the two studies, which may partially be explain by the different conditional depletion systems used (GlmS mediated conditional KD here) and should be clearly discussed, for example a lack of an effect on virulence gene expression in the present study, as well as the absence of a growth phenotype.

The in-depth correlation of AP2-P relative to MORC and other published transcription factor data sets with chromatin contact points determined by MicroC provides, however, novelty, and elegantly links previous and new datasets into a biological concept. Unfortunately, this study is restricted to a single parasite stage during the asexual blood phase and thus provides only limited mechanistic insight. It would be highly relevant to see the dynamics of these contacts and boundaries (at least) during the asexual life cycle to better understand how these chromatin architectural features influence gene expression. For example, the AP2-P and MORC enriched euchromatic contact points are insulating genes which are highly transcribed in the schizont stage from neighbouring genes, but it remains to be shown whether these boundaries disappear in the ring stage, or are replaced by other factors. Likewise, it remains unclear whether the subtelomeric genic/non-genic boundaries play a role in suppressing *upsB* var genes, and whether this is stage dependent.

In summary, in my opinion the paper does not meet the standard for publication in *Nature Microbiology* in its current form. It would significantly benefit from more replicates of MORC ChIP; ChIP of MORC in AP2-KD parasites; statistical evidence for loss of boundaries in AP2-KD parasites; time resolution by inclusion of additional life stages for ChIP and MicroC (please find further suggestions below).

Major:

For MORC, only a single ChIP replicate has been included at a single stage in the life cycle. To verify the validity of the data, at least a second biological replicate is necessary (in agreement with ENCODE standards).

Figure 1G: the definition of “non-virulent HP1 domains” versus “virulent HP1 domains” is unclear to me, and the number of non-virulent domains displayed in the figure seems rather high considering that most of the heterochromatin compartment covers clusters of var, rif, stevor, and other exported protein families considered virulence factors. I am not aware of that many large HP1 domains comprising “non-virulent genes”, only a few single genes. Please define this further and provide information on the size and location of these domains, as well as on the “insulation scores” for each of these regions, for example in a supplementary table. Is the same phenomenon evident in the 3D7 AP2-P WT line, and how does it change after AP2-P KD?

Suppl. Figure 2C, J: In the 3D7v55 reference genome, the Chr 14 sequence is truncated to 1500 bp (as opposed to 20 Kb in other chromosomes) and has only a 1 kb non-genic sequence, therefore the graphical depiction of the chromosomal regions in this figure is wrong. Consequently, the authors should show a different chromosome to support the link between var gene biology and presence of the “fold like structure” between genic and non-genic regions (page 7) or remove this conclusion and figure all together. To my knowledge and in agreement with Fig S2A, there is no “complete” chromosome end in this reference genome that lacks a telomere proximal *upsB* var gene. Chromosomes 6 and 11, which have a telomeric *upsA* var gene annotated, are also largely missing non-genic telomeric sequences. Likewise, the absence of MORC and AP2-P seems to be related to this chromosomal end deletion. I think this needs to be clarified. (Of note: AP2-P data from Subudhi et al. 2023 data actually do show a peak upstream of the var pseudogene of the left Chr 14 end).

Figure 2C: AP2-P and MORC enrichment upstream of subtelomeric var gene – are any of these sites consistent with the SPE2/SIP2 binding sites (Flueck et al. 2010)? From Suppl. Figure 2H it appears like SIP2 binding sites are enriched under the dominant MORC peak only. Of note, in the chromosome 3 depiction the SIP2 binding sites are shifted into the var gene. Please check whether this is correct, as the SPE2 motifs actually lie further upstream. It would be interesting to do motif searches for these particular regions enriched by AP2-P / MORC (similar to Figure S4 but for the subtelomeric sites only).

It looks like the level of enrichment at the three subtelomeric peaks of MORC and AP2-P is contrary, with the dominant MORC peak 1 kb closer to the telomere than the dominant AP2-P peak. Given the distance of the peaks, do you predict that there is a direct interaction between these proteins, and can you provide experimental evidence for this, for example by Co-IP or ChIP-re-ChIP?

Did you find AP2-P and/or MORC enriched in promoters of central var genes, as suggested by Subudhi et al?

How do the observed intra- and interchromosomal contacts in the euchromatin and heterochromatin compartment change over the life cycle, when expression of the genes near the contact points varies? Can similar structures be observed in rings?

Suppl. Fig. 3E: In schizonts, usually var exon 2 transcripts dominate, whereas coding mRNAs are highly expressed in rings. Do the observed transcripts reflect exon 2 transcripts or coding transcripts? Do you see any differences in var expression in the ring stages?

The effect of AP2-P KD on MORC expression (Figure 3A) is interesting and highly relevant. According to Subudhi et al. 2023 AP2-P ChIP data, there is an AP2-P peak at the TSS of MORC (in rings). Is this evident in the data presented here? This may suggest that the loss of boundary integrity observed after AP2-P depletion is a consequence of the downregulation of MORC. How does the genomic distribution of MORC change upon AP2-P KD in the double transgenic parasite line?

I find the KD MicroC graphs in Figure 3 difficult to judge – to me, the KD data appear more noisy in general. Can the data shown in Fig 3B and C be quantified, for example by comparing the normalized contacts at the boundary to normalized contacts of telomeres, centromeres and central var clusters, which are expected not to be affected if the role of AP2-P KD in boundary formation is specific (similar to 3F)? If I understand correctly, in table 6 the insulation scores are reported for subtelomeric and non-subtelomeric heterochromatic genes etc. Does the category “non-genic subtelomeric” represent the identified subtelomeric

fold structure contact points? Is there a significant difference in the insulation scores or contacts of these different chromosomal locations between KD and WT?

Fig 5A: What is plotted on the Y-axis here? If I understand correctly, the boundary is a point of decreased interaction with the surrounding genome. So what exactly are the genes "outside the boundaries" (Fig 5C) - genes (how) distant from boundaries? The experiments were conducted in four technical replicates. To get an idea of the biological variability, how well did AP2-P KD (no glucosamine) and NF54 MicroC data correlate regarding (1) the boundaries between telomeric and subtelomeric heterochromatin, (2) contacts between virulence gene clusters, and (3) euchromatic contact points?

In the introduction (page 5), the authors suggest that MicroC is useful to address the question of enhancers-promoter pairing. Enhancers remain poorly defined in *P. falciparum*. Does the MicroC data provide any evidence for the existence of enhancers, for example within genes?

Minor comments:

The MORC mutation is in the literature and protein databases commonly spelled "microrchidia" not "microorchidia", although I agree that the latter would be biologically more appropriate.

Please report the correlation coefficients for the four NF54 MicroC replicates.

Fig 1 C-F please include a scale for the blue red part of the graphs (log2 scaled observed/expected interaction frequency)

Page 7, last sentence: it is unclear what findings were confirmed by the ChIPseq results of AP2-P and MORC.

Fig 1 G: please defined the "non-virulent HP1 domains" further, for example by providing an excel sheet of coordinates and associated genes.

Table 6: correct the word "euchromatinatic"

Please also provide p-values for Suppl. Figure 3H.

Page 10, Motifs: please note that the AP2i "GTGCA" motif is also part of the SPE2 motif and is not only (weakly) evident in the predicted AP2-P binding sites, but also in one of the MORC motifs. As mentioned above, it would make sense to investigate motifs in heterochromatic and euchromatic boundaries separately.

Page 12, Table 7: there are no inter-chromosomal interactions listed in the table, only intra-chromosomal.

Figure 6 – colors are missing

Reviewer #3 (Remarks to the Author):

The manuscript titled "Micro-C reveals MORC/ApiAP2-mediated links between distant, functionally related genes in the human malaria parasite" by Singh et al. offers valuable new insights into the three-dimensional organization of the *Plasmodium falciparum* genome. Through a combination of cutting-edge genomic techniques, including high-resolution Micro-C, ChIP-seq, RNA-seq, and other genomic assays, the authors present a comprehensive and nuanced depiction of the genome's structural and functional dynamics during specific stages of parasite infection.

The clarity of the presentation, coupled with the outstanding quality of the writing, allows the intricate findings to be easily accessible to the reader. The authors provide compelling evidence, backed by high-resolution Micro-C maps, to support their conclusions on genome folding and chromatin landscape. The data are thoroughly analyzed and convincingly discussed, with the clear and easily understandable figures.

In sum, this manuscript offers significant contributions to our understanding of *P. falciparum* biology, shedding light on novel forms of coordination between chromatin factors (like AP2-P; MORC and HP1), genome architecture and gene expression as well as offering an appealing hypothesis on a mechanism bringing var genes in close proximity for their known recombination events. I have no reservations in recommending this work for publication in *Nature Microbiology*.

However, I would like to suggest a few minor revisions to further improve the manuscript. First, when discussing Hi-C maps and similar 3C-based data, I recommend consistently using the term "contact" instead of "interaction," as seen in a few instances within the text and figures. Additionally, in the second paragraph of the Results section (page 7), the term "compacts" would be preferable to "condenses" when describing DNA organization.

Lastly, I noted a minor inconsistency in the PCR band sizes on agarose gels, as depicted in Figures S2D, E, and S3G, which may be due to a labeling issue in the DNA ladder lane. It would also be beneficial to include P-values (even if non-significant) in Figure S3H for consistency with similar graphs, such as Figure 3F.

Version 1:

Reviewer comments:

Reviewer #1

(Remarks to the Author)

The changes made by the authors addressed all my concerns and further strengthened this excellent manuscript.

Björn Kafsack

Reviewer #2

(Remarks to the Author)

The authors have adequately addressed most of my comments. Particularly, the addition of further replicates and the inclusion of a second time point for MicroC in wt parasites, as well as the addition of several statistical comparisons have significantly strengthened the key conclusions presented in the manuscript,

I have a few remaining questions:

Page 7 Line 167; page 9 line 212: While the authors have corrected the graphical depiction of the left end of Chr. 14 in ED Fig 2C, they should still also clarify that the var gene-free chromosomal ends they are looking at are special in more than the aspect of not having a ups-B var gene; that they do not have canonical telomeric sequences and the parasites used for MicroC evidently carry deletions in the relevant boundary areas between non-coding and virulence gene regions on the right chromosome end, as seen in ED Fig 2C. I think this is very important information when concluding that the data point to a role of fold structures and var gene biology.

In ED Fig 2C right panel, the right chromosome end of Chr. 14 is shown in the wrong orientation. Consequently, the annotated area (3,191-3,291 kb) is reverse. Please correct this depiction (it should be the mirror image).

Line 227: There is a difference of about 2-3 hours in the transcriptional age of controls and AP2-P KD according to ED Fig 3D, which would be worth pointing out. This is surprising considering the down-regulation of schizont stage genes in the absence of AP2-P. What is the (transcriptional) explanation of this "accelerated" transcriptome profile?

Please add the scale to all ChIPseq traces or mention the range in the legends (Fig. 1, 2, 3, 5, 6 and ED Figures). According to Fig. 4, ChIPseq data are presented on a log₂ scale from 1-5, which I guess was chosen to limit the depicted ChIP peaks to signals enriched more than 2-fold.

Line 310: A boundary insulation score is mentioned with reference to Fig. 5D. I am unsure which (numerical?) score is referred to here.

Line 368: The notion of an anchor point upstream of the single active var gene Pf3D7_1240900 is interesting, but I can not easily follow how the authors arrive at this conclusion from ED Fig. 6H. There is only one cyan diamond depicting an anchor point at the start of the heterochromatin boundary in early stages, and this is missing/shifted in late stages. However, the area in front of the green active var gene appears similar in both, early and late stage parasites and no anchor point is depicted. Thus, I am wondering whether there is any statistical evidence for this claim.

The correlation of MicroC data and var gene expression with heterochromatin boundaries is very interesting and relevant. The authors should perform a HP1 (or H3K9me3) ChIPqPCR experiment in their parasite line used for MicroC to answer the question raised in line 373, whether the active var gene locus is heterochromatic in early/late stages and whether this correlates with the proposed boundary anchors.

Decision Letter:

Our ref: NMICROBIOL-24092815A

2nd April 2025

Dear Jessica,

Thank you for submitting your revised manuscript "Micro-C reveals MORC/ApiAP2-mediated links between distant, functionally related genes in the human malaria parasite" (NMICROBIOL-24092815A). It has now been seen by two of the original referees and their comments are below. The reviewers find that the paper has improved in revision, and therefore we'll be happy in principle to publish it in Nature Microbiology, pending minor revisions to satisfy the referees' final requests and to comply with our editorial and formatting guidelines. Please note that we editorially overrule the request by referee #2 for an HP1 (or H3K9me3) ChIPqPCR experiment in the parasite line used for MicroC. However, we will need you to address all other comments through text changes.

Thank you again for your interest in Nature Microbiology. Please do not hesitate to contact me if you have any questions.

Best wishes,

Reviewer #1 (Remarks to the Author):

The changes made by the authors addressed all my concerns and further strengthened this excellent manuscript.

Björn Kafsack

Reviewer #2 (Remarks to the Author):

The authors have adequately addressed most of my comments. Particularly, the addition of further replicates and the inclusion of a second time point for MicroC in wt parasites, as well as the addition of several statistical comparisons have significantly strengthened the key conclusions presented in the manuscript,

I have a few remaining questions:

Page 7 Line 167; page 9 line 212: While the authors have corrected the graphical depiction of the left end of Chr. 14 in ED Fig 2C, they should still also clarify that the var gene-free chromosomal ends they are looking at are special in more than the aspect of not having a ups-B var gene; that they do not have canonical telomeric sequences and the parasites used for MicroC evidently carry deletions in the relevant boundary areas between non-coding and virulence gene regions on the right chromosome end, as seen in ED Fig 2C. I think this is very important information when concluding that the data point to a role of fold structures and var gene biology.

In ED Fig 2C right panel, the right chromosome end of Chr. 14 is shown in the wrong orientation. Consequently, the annotated area (3,191-3,291 kb) is reverse. Please correct this depiction (it should be the mirror image).

Line 227: There is a difference of about 2-3 hours in the transcriptional age of controls and AP2-P KD according to ED Fig 3D, which would be worth pointing out. This is surprising considering the down-regulation of schizont stage genes in the absence of AP2-P. What is the (transcriptional) explanation of this "accelerated" transcriptome profile?

Please add the scale to all ChIPseq traces or mention the range in the legends (Fig. 1, 2, 3, 5, 6 and ED Figures). According to Fig. 4, ChIPseq data are presented on a log₂ scale from 1-5, which I guess was chosen to limit the depicted ChIP peaks to signals enriched more than 2-fold.

Line 310: A boundary insulation score is mentioned with reference to Fig. 5D. I am unsure which (numerical?) score is referred to here.

Line 368: The notion of an anchor point upstream of the single active var gene Pf3D7_1240900 is interesting, but I can not easily follow how the authors arrive at this conclusion from ED Fig. 6H. There is only one cyan diamond depicting an anchor point at the start of the heterochromatin boundary in early stages, and this is missing/shifted in late stages. However, the area in front of the green active var gene appears similar in both, early and late stage parasites and no anchor point is depicted. Thus, I am wondering whether there is any statistical evidence for this claim.

The correlation of MicroC data and var gene expression with heterochromatin boundaries is very interesting and relevant. The authors should perform a HP1 (or H3K9me3) ChIPqPCR experiment in their parasite line used for MicroC to answer the question raised in line 373, whether the active var gene locus is heterochromatic in early/late stages and whether this correlates with the proposed boundary anchors.

Version 2:

Decision Letter:

16th May 2025

Dear Jessica,

I am pleased to accept your Article "High resolution map of the Plasmodium falciparum genome reveals MORC/ApiAP2-mediated links between distant, functionally related genes" for publication in Nature Microbiology. Thank you for having chosen to submit your work to us and many congratulations.

Authors may need to take specific actions to achieve [compliance](https://www.springernature.com/gp/open-research/funding/policy-compliance-faqs) with funder and institutional open access mandates. If your research is supported by a funder that requires immediate open access (e.g. according to [Plan S principles](https://www.springernature.com/gp/open-research/plan-s-compliance)) then you should select the gold OA route, and we will direct you to the compliant route where possible. For authors selecting the subscription publication route, the journal's standard licensing terms will need to be accepted, including [self-archiving policies](https://www.nature.com/nature-portfolio/editorial-policies/self-archiving-and-license-to-publish). Those licensing terms will supersede any other terms that the author or any third party may assert apply to any version of the manuscript.

We welcome the submission of potential cover material (including a short caption of around 40 words) related to your manuscript; suggestions should be sent to Nature Microbiology as electronic files (the image should be 300 dpi at 210 x 297 mm in either TIFF or JPEG format). Please note that such pictures should be selected more for their aesthetic appeal than for their scientific content, and that colour images work better than black and white or grayscale images. Please do not try to design a

cover with the Nature Microbiology logo etc., and please do not submit composites of images related to your work. I am sure you will understand that we cannot make any promise as to whether any of your suggestions might be selected for the cover of the journal.

Congratulations once again and I look forward to seeing the article published.

With kind regards,

P.S. Click on the following link if you would like to recommend Nature Microbiology to your librarian
<http://www.nature.com/subscriptions/recommend.html#forms>

** Visit the Springer Nature Editorial and Publishing website at http://editorial-jobs.springernature.com?utm_source=ejP_NMicro_email&utm_medium=ejP_NMicro_email&utm_campaign=ejP_NMicro for more information about our career opportunities. If you have any questions please click [here](mailto:editorial.publishing.jobs@springernature.com). **
